# Elongator is a microtubule polymerase selective for polyglutamylated tubulin

Vicente J Planelles-Herrero [ID][1][✉], Mariya Genova[2,3,4], Lara K Krüger [ID][1,4], Alice Bittleston [ID][1,4], Kerrie E McNally [ID][1], Tomos E Morgan[1], Gianluca Degliesposti[1], Maria M Magiera [ID][2,3], Carsten Janke [ID][2,3] & Emmanuel Derivery [ID][1][✉]

## Abstract

**Elongator is a tRNA-modifying complex that regulates protein translation. Recently, a moonlighting function of Elongator has been identified in regulating the polarization of the microtubule cytoskeleton during asymmetric cell division. Elongator induces symmetry breaking of the anaphase midzone by selectively stabilizing microtubules on one side of the spindle, contributing to the downstream polarized segregation of cell-fate determinants, and therefore to cell fate determination. Here, we investigate how Elongator controls microtubule dynamics. Elongator binds both to the tip of microtubules and to free GTP-tubulin heterodimers using two different subcomplexes, Elp123 and Elp456, respectively. We show that these activities must be coupled for Elongator to decrease the tubulin critical concentration for microtubule elongation. As a consequence, Elongator increases the growth speed and decreases the catastrophe rate of microtubules. Surprisingly, the Elp456 subcomplex binds to tubulin tails and has strong selectivity towards polyglutamylated tubulin. Hence, microtubules assembled by Elongator become selectively enriched with polyglutamylated tubulin, as observed in vitro, in mouse and *Drosophila* cell lines, as well as in vivo in *Drosophila* Sensory Organ Precursor cells. Therefore, Elongator rewrites the tubulin code of growing microtubules, placing it at the core of cytoskeletal dynamics and polarization during asymmetric cell division.**

**Keywords** Microtubules; Cytoskeleton; Elongator; Tubulin Modifications; Central Spindle
**Subject Categories** Cell Adhesion, Polarity & Cytoskeleton; RNA Biology

## Introduction

Microtubules are dynamic cytoskeletal polymers found in every eukaryotic cell, where they are essential for cell division, morphogenesis, cell motility and intracellular transport. The structure, properties and dynamics of microtubules structures are tightly regulated by a plethora of proteins, including microtubule-associated proteins (MAPs), motor proteins and tubulin-modifying enzymes. Together, all these factors control the geometry of the microtubule landscape, leading to the formation of structures with highly distinctive shapes and behaviours, such as long and stable axonal microtubules, the dynamic mitotic spindle, or axonemal microtubules that mediate ciliary beating.

Microtubules are asymmetric hollow tubes built from heterodimers of α- and β-tubulin that are incorporated at both ends of the polymer: the slowly growing minus-end, and the fast growing plus-end. In cells, microtubule dynamics are primarily controlled by regulating plus-end dynamics, while the minus-end is often anchored or protected from depolymerisation (Howard and Hyman, 2003; Dammermann et al, 2003; Desai and Mitchison, 1997). A particularly important group of proteins controlling microtubule dynamics is the family of the so-called microtubule polymerases, which specifically recognize the growing-end of microtubules and increase their growth rate and/or decrease their catastrophe frequency (Geyer et al, 2018; Chen and Hancock, 2015; Brouhard et al, 2008; Arpag et al, 2020). Arguably the best characterized member of this family is XMAP215, a TOG (Tumour Overexpressed Gene)-domain containing protein. Through a series of tandem-linked TOG domains, XMAP215 polymerizes microtubules by simultaneously binding to the microtubule end and to free αβ-tubulin heterodimers, thus facilitating the integration of αβ-tubulin heterodimers into the growing microtubule (Al-Bassam et al, 2007; Brouhard et al, 2008). On the other hand, CLASP, which contains only a single TOG domain, seems to stabilize microtubules using a different mechanism (Lawrence et al, 2023; Majumdar et al, 2018; Lawrence et al, 2018; Arpag et al, 2020). Finally, motor proteins have also been shown to modulate microtubule dynamics (Chen and Hancock, 2015; Arpag et al, 2020). For example, kinesin-5, a tetrameric member of the kinesin family, enhances microtubule polymerization by stabilizing tubulin-tubulin interactions at the growing ends of microtubules (Chen and Hancock, 2015); whilst other kinesins, such as Kinesin-13, can actively depolymerise microtubules (Ems-McClung and Walczak, 2010).

Microtubules can also be modified by post-translational modifications (PTMs) of tubulin, further modulating their

[1]Cell Biology Division, MRC Laboratory of Molecular Biology, Francis Crick Avenue, Cambridge, UK. [2]Institut Curie, Université PSL, CNRS UMR3348, Orsay, France. [3]Université Paris-Saclay, CNRS UMR3348, Orsay, France. [4]These authors contributed equally: Mariya Genova, Lara K Krüger, Alice Bittleston. ✉E-mail: vicente@mrc-lmb.cam.ac.uk; derivery@mrc-lmb.cam.ac.uk

properties (Janke and Magiera, 2020; Roll-Mecak, 2020). The C-terminal tails of both α- and β-tubulin are hotspots of modifications, for instance polyglutamylation, a modification adding secondary peptide chains of glutamates attached to the C-termini of tubulins. Polyglutamylation is catalysed by the members of the tubulin tyrosine ligase-like (TTLL) protein family, and it promotes the binding of several MAPs, or the action of microtubule-severing enzymes such as spastin and katanin (Lacroix et al, 2010; Valenstein and Roll-Mecak, 2016; Shin et al, 2019; Kuo et al, 2019; Bonnet et al, 2001; Boucher et al, 1994; Chen and Roll-Mecak, 2023). Polyglutamylation has thus the potential to modulate microtubule dynamics and ultrastructure. Since most, albeit not all, enzymes catalysing post-translational modifications preferentially modify microtubules rather than soluble tubulin dimers (Shida et al, 2010; Regnard et al, 1998; Kumar and Flavin, 1981; Chen and Roll-Mecak, 2023), it is currently thought that PTMs mostly change the behaviour of pre-existing microtubules.

The Elongator complex is a conserved molecular machine regulating protein translation. In particular, Elongator selectively modulates protein translation rates by modifying the wobble uridines at position 34 ($U_{34}$) of a subset of tRNAs (Dauden et al, 2019; Lin et al, 2019). Structurally, Elongator is composed of six subunits, Elp1-6, which are each present in two copies and arranged in two discrete, stable sub-complexes (Dauden et al, 2017; Glatt et al, 2012; Jaciuk et al, 2023; Setiaputra et al, 2017; Abbassi et al, 2024): an Elp123 dimer and an Elp456 hexamer (Fig. 1A). In the context of its tRNA modification function, tRNAs bind to the larger, catalytically active Elp123 subcomplex (Lin et al, 2019; Dauden et al, 2019; Jaciuk et al, 2023; Abbassi et al, 2024). The function of the Elp456 sub-complex is less understood, but it is thought that its binding to the Elp123-tRNA complex releases the modified tRNA through a competition mechanism (Jaciuk et al, 2023; Abbassi et al, 2024).

Surprisingly, Elongator has recently been shown to control symmetry breaking of the anaphase spindle midzone during the asymmetric cell division of *Drosophila* Sensory Organ Precursor (SOP) cells (Planelles-Herrero et al, 2022). This asymmetric central spindle in turns polarizes the segregation of signalling endosomes containing cell-fate determinants towards only one daughter cell, therefore contributing to asymmetric cell fate determination. Unexpectedly, Elongator's activity of modulating microtubule dynamics was found to be independent on its effect on protein translation (Planelles-Herrero et al, 2022). Rather, Elongator directly binds to microtubules and modulates their dynamics, specifically by increasing their growth rate and their lifetime by decreasing their catastrophe frequency (Planelles-Herrero et al, 2022). Since Elongator is asymmetrically localized on one side of the anaphase spindle midzone, this induces preferential microtubule stabilization on one side of the spindle and thus symmetry breaking. However, how Elongator modulates microtubule dynamics at the molecular level was unknown.

Here, we reveal how Elongator stabilizes microtubules. We show that Elongator can specifically recognise and track the growing ends of microtubules. By simultaneously binding to microtubule tips, via Elp123, and to αβ-tubulin heterodimers, via Elp456, Elongator increases the local concentration of tubulin dimers at growing ends, thereby increasing the growth rate and decreasing the catastrophe rate of microtubules. Strikingly, we show that Elp456 preferentially binds polyglutamylated tubulin heterodimers, while Elp123 binds to

microtubules regardless of their PTMs. Thus, in the presence of Elongator, microtubules not only grow faster, but also get selectively and specifically enriched in polyglutamylated tubulin. Accordingly, we found that Elongator depletion leads to reduced levels of polyglutamylation at the mitotic spindle in fly S2 cells, mouse fibroblasts and fly SOP cells in vivo. These results highlight an unexpected function for Elongator in remodelling the landscape of microtubule modifications, whilst itself not being a tubulin-modifying enzyme. Our work thus uncovers a novel molecular mechanism of how microtubule PTM diversity can be achieved in cells.

# Results

## Elongator is a microtubule end-tracking protein complex

To investigate the molecular mechanism by which Elongator binds and stabilizes microtubules, we capitalized on our previously described procedure for purifying the *Drosophila* Elongator complex from cultured S2 cells (Planelles-Herrero et al, 2022) to derive an Alexa488-labelled SNAP-Elongator (Figs. 1 and EV1A). We then investigated the localization of this complex on dynamic microtubules using an established assay in which cycles of growth/catastrophe of dynamic microtubules from stable seeds are imaged by Total Internal Reflection Microscopy (TIRFM) (Gell et al, 2010) (Figs. 1B–E and EV1B,C). Under conditions in which Elongator and tubulin were at physiological concentrations (17 μM tubulin, 25 nM Alexa 488-SNAP-Elongator (Planelles-Herrero et al, 2022)) strong binding of Elongator to both the plus- (Fig. 1C,D) and minus-ends (Fig. 1E) of growing microtubules could be detected in ~10% of the analysed growth events. The end-localization of Elongator was lost when the microtubule underwent catastrophe (Figs. 1E and EV1B,C, orange arrows), showing that Elongator specifically recognizes the growing ends of microtubules. Moreover, and consistent with other end-tracking proteins (Brouhard et al, 2008; Lopez and Valentine, 2016; Reid et al, 2019), Elongator could also be observed diffusing on the GMPCPP seeds (Fig. EV1B,C), in conditions where Elongator tracks the growing end of microtubules (that is, when free tubulin is present). Strikingly, in these conditions, Elongator could be observed moving from the seed to the growing end of a microtubule (Fig. EV1C, yellow arrows), highlighting its preference for GTP-like structures. Consistently, the GDP-lattice of the microtubule was mostly devoid of Elongator signal (Fig. 1C,D).

The growing end of microtubules (also known as the microtubule "tip" or "cap") is formed by newly incorporated GTP-αβ-tubulin heterodimers. Once incorporated, GTP is hydrolysed to GDP, creating a characteristic GDP microtubule shaft with a "GTP-cap" located at the growing end of the polymer. To better understand how Elongator binds to the end of microtubules, we examined the binding of Elongator to GTP-mimicking microtubules. First, we used the GTP analogue GMPCPP, which mimics key features of the GTP-state of microtubule ends (Estévez-Gallego et al, 2020; Alushin et al, 2014), to grow microtubules. As we previously reported (Planelles-Herrero et al, 2022), Elongator binds and decorates GMPCPP-stabilized microtubules along their entire length (Fig. 1F). To confirm that tip-recognition involved recognition of the tubulin nucleotide state, we used $BeF_3^-$ microtubule seeds, which are known to more

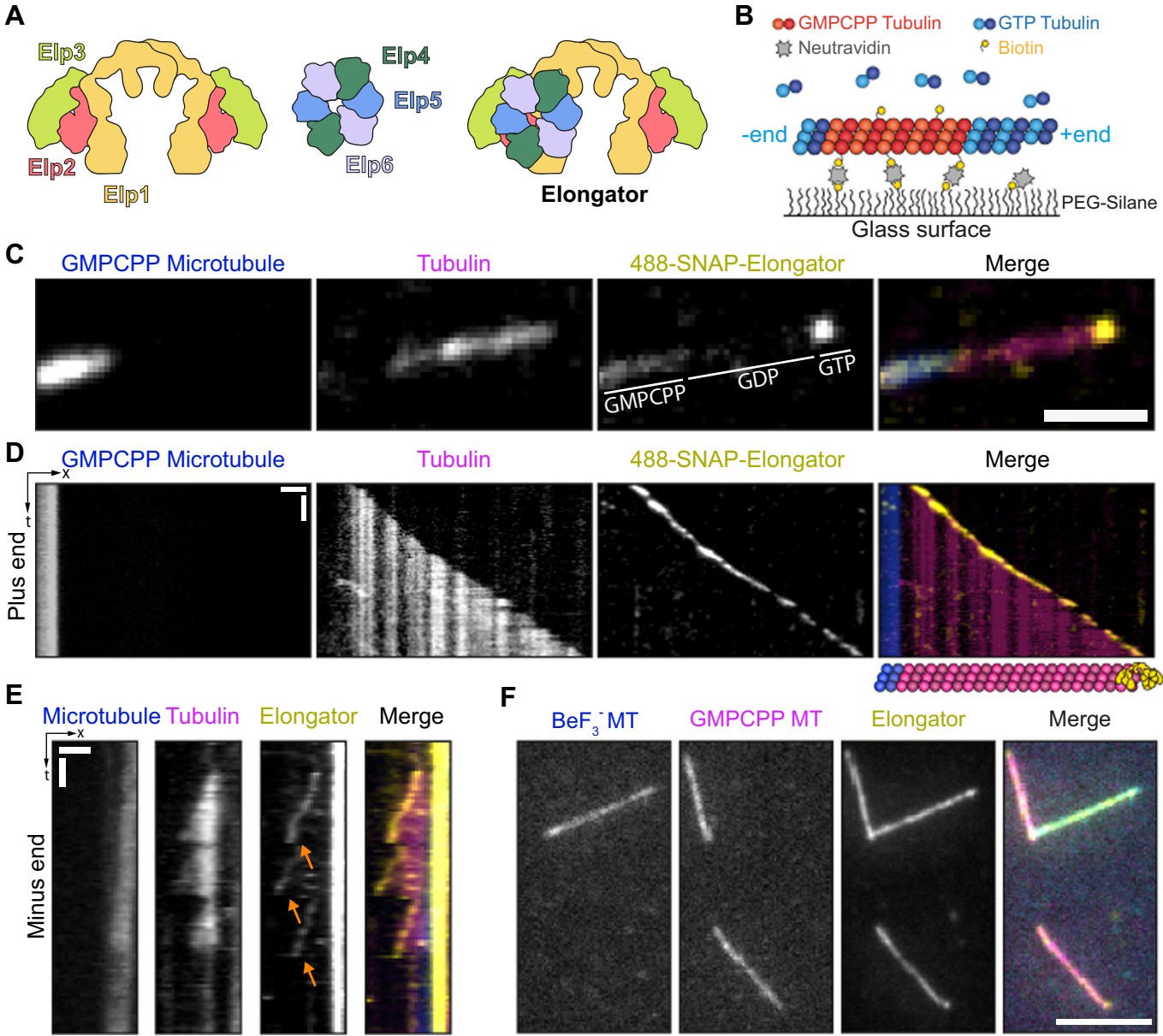

**Figure 1.  Elongator complex tracks the growing ends of microtubules.**

(**A**) Elongator contains one copy of two distinct stable sub-complexes: Elp123 (left) and the Elp456 ring (right). Note that although Elp123 contains two potential binding sites for Elp456, multiple assays confirmed that only one copy is preferentially bound (Setiaputra et al, 2017; Dauden et al, 2017; Jaciuk et al, 2023), making Elongator an asymmetric complex. (**B**) Assay design: Biotinylated, rhodamine-labelled GMPCPP-stabilized seeds (magenta) are anchored via NeutrAvidin to PLL-PEG-Silane-Biotin. Free tubulin (17 μM 10% HiLyte 647-labelled, cyan) and 25 nM Alexa 488-SNAP-Elongator (yellow) is added and microtubule polymerization is observed by TIRFM. (**C**) Representative microtubule showing Alexa488 SNAP-Elongator binding to the GMPCPP and GTP moieties of a microtubule. Note that Elongator signal is absent in the GDP region of the microtubule. (**D, E**) Representative kymographs showing Alexa488 SNAP-Elongator complex tracking the plus (**D**) and minus (**E**) ends of growing microtubules. Note that Elongator detaches from the microtubule end when microtubules undergo catastrophe (orange arrows). Also note Elongator accumulating at the transition point between the GMPCPP seed and the GDP lattice (see also Fig. EV1B). (**F**) Elongator decorates both GDP·BeF₃⁻ and GMPCPP stabilized microtubules. Scale bars = 2 μm (**C**), 2 min/2 μm (**D, E**), 5 μm (**F**). Source data are available online for this figure.

precisely mimic the GTP-state of the microtubules than GMPCPP (Estévez-Gallego et al, 2020). Elongator was consistently found to bind BeF₃⁻ stabilized microtubules with similar apparent affinities to GMPCPP stabilized microtubules (Fig. 1F). This establishes that Elongator specifically recognizes the GTP (or GTP-like) state of the growing ends of microtubules, both at the minus and at the plus ends. Note that an increased Elongator signal can also be observed at transition points between the GMPCPP seed and the GDP lattice (Figs. 1E and EV1B), a feature seen in end-binding (EB) proteins (Reid et al, 2019), suggesting that the binding to GTP-cap could be facilitated by the open, tapered microtubule ends.

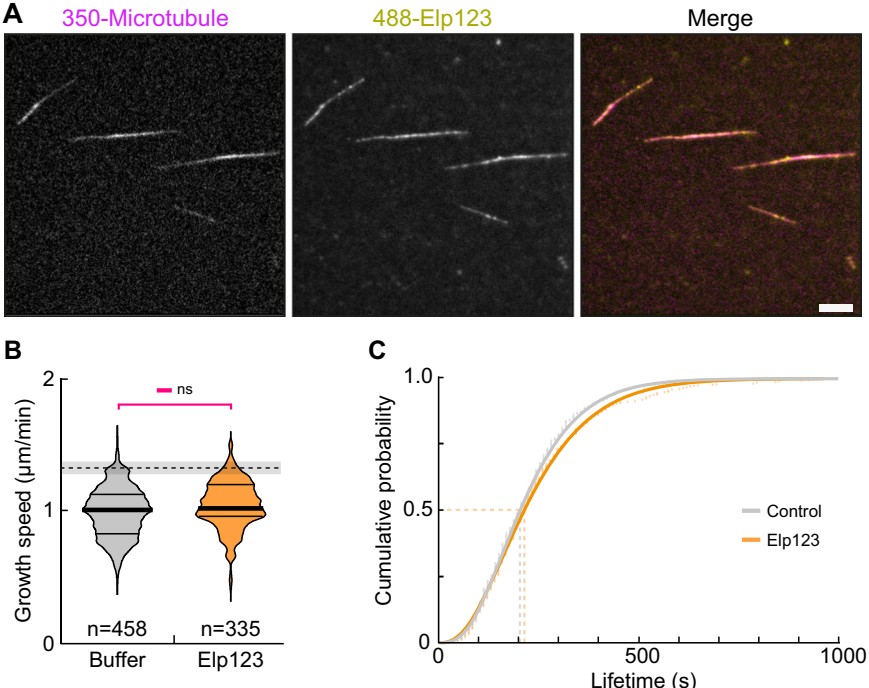

**Figure 2. Elp123 binds but does not stabilize microtubules.**

(A) AMCA-labelled GMPCPP stabilized microtubules incubated with 25 nM Elp123 sub-complex labelled with AlexaFluor488 dye and observed by TIRFM. Elp123 directly binds to GMPCPP microtubules. (B) Microtubule seeds were polymerized in the presence of 16 μM GTP-tubulin (10% HiLyte 647 labelled) in the absence or presence of 25 nM (His)$_6$-PC-SNAP-Elp123, and microtubule growth rate at the plus end was measured by TIRFM. The control was performed using the same buffer as Elongator kept from the final purification step (see Appendix Fig. S1A). *P* value from a two-sided Kruskal–Wallis test for non-parametric samples is indicated; *n*, number of individual growth events quantified from at least three independent experiments. Dashed line represents an increase of ~1.4-fold in the speed of microtubule growth, which happens in the presence of the full Elongator complex (Planelles-Herrero et al, 2022). Thick line, median; thin lines, quartiles. Similar results were observed at the minus end (see text). (C) Cumulative microtubule lifetime distribution of microtubules grown at the plus end in the absence and presence of Elp123. Elp123 does not stabilize microtubules. Mean lifetime estimate ± error (lifetime at half cumulative distribution, dashed lines): 202.61 ± 7.10 s (Control) and 211.76 ± 8.14 s (Elp123). Lines, gamma distribution fits. Scale bar = 5 μm. Source data are available online for this figure.

## The Elp123 subcomplex binds to, but does not stabilise, microtubules

We next investigated how Elongator binds to microtubules. An isolated Elp2 subunit produced in yeast has been shown to weakly interact with microtubules at high concentration (Dong et al, 2015), and we previously reported that neither Elp456 hexamers nor a partial sub-complex of Elp23 and interacting protein Hsc70-4 exhibit appreciable microtubule binding (Planelles-Herrero et al, 2022). Since the full Elongator complex, consisting of two copies of Elp123, which homo-dimerize via Elp1, and one Elp456 hexameric ring (Fig. 1A) binds to microtubules at very low concentrations (Fig. 1), we speculated that the full, dimeric Elp123 sub-complex might be needed for this activity.

To purify the Elp123 subcomplex, we used the biG-Bac expression system to simultaneously overexpress *Drosophila* Elp1, Elp2 and Elp3 in Sf9 insect cells (Weissmann et al, 2016) (Appendix Fig. S1A–C). We tagged Elp3 since it is the only subunit in fly Elp123 that can accommodate a SNAP-tag whilst being compatible with the assembly of the full complex (Planelles-Herrero et al, 2022). After two affinity purification steps, the gel filtration profile shows that a highly pure Elp123 elutes at the expected elution volume (~600 kDa, Appendix Fig. S1B), showing that *Drosophila* Elp123 dimerizes as expected (Dauden et al, 2017; Setiaputra et al, 2017; Jaciuk et al, 2023).

Strikingly we found that Alexa 488-SNAP-Elp123 directly binds to microtubules using TIRFM (Fig. 2A; Appendix Fig. S1F, left panel) and microtubule pelleting assays (Appendix Fig. S1D), confirming that the Elp123 subcomplex is sufficient for microtubule binding. Yet, Elp123 does not stabilize microtubules (Fig. 2B,C). Indeed, using the same microtubule dynamics assays depicted before (Fig. 1B), no effect was detected on microtubule plus-end growth speed (Fig. 2B, mean ± SEM: 1.002 ± 0.009 μm/min for the buffer control, versus 1.043 ± 0.009 μm/min for Elp123, *n* = 335) or lifetime (Fig. 2C, 202.61 ± 7.10 s versus 211.76 ± 8.14 s). This is in sharp contrast to the effects of the full Elongator complex at the plus end in similar conditions, which shows a 1.4-fold increase of the growth speed (dashed line in Fig. 2C) and a 1.8-fold increase of microtubule lifetime (Planelles-Herrero et al, 2022). Similarly, we also could not detect any effect of Elp123 at the minus ends (control: 0.348 ± 0.008 μm/min, *n* = 115 ; Elp123: 0.343 ± 0.010 μm/min, *n* = 138, Appendix Fig. S2A,B).

Altogether, our data show that, although Elp123 directly interacts with microtubules (Fig. 2A), this is not sufficient to modulate their dynamics, in particular microtubule stabilization. Thus, an as-of-yet uncharacterized mechanism must be responsible for this effect. Since the Elp456 sub-complex was absent in this experiment, we examined the role of Elp456 in microtubule stabilization.

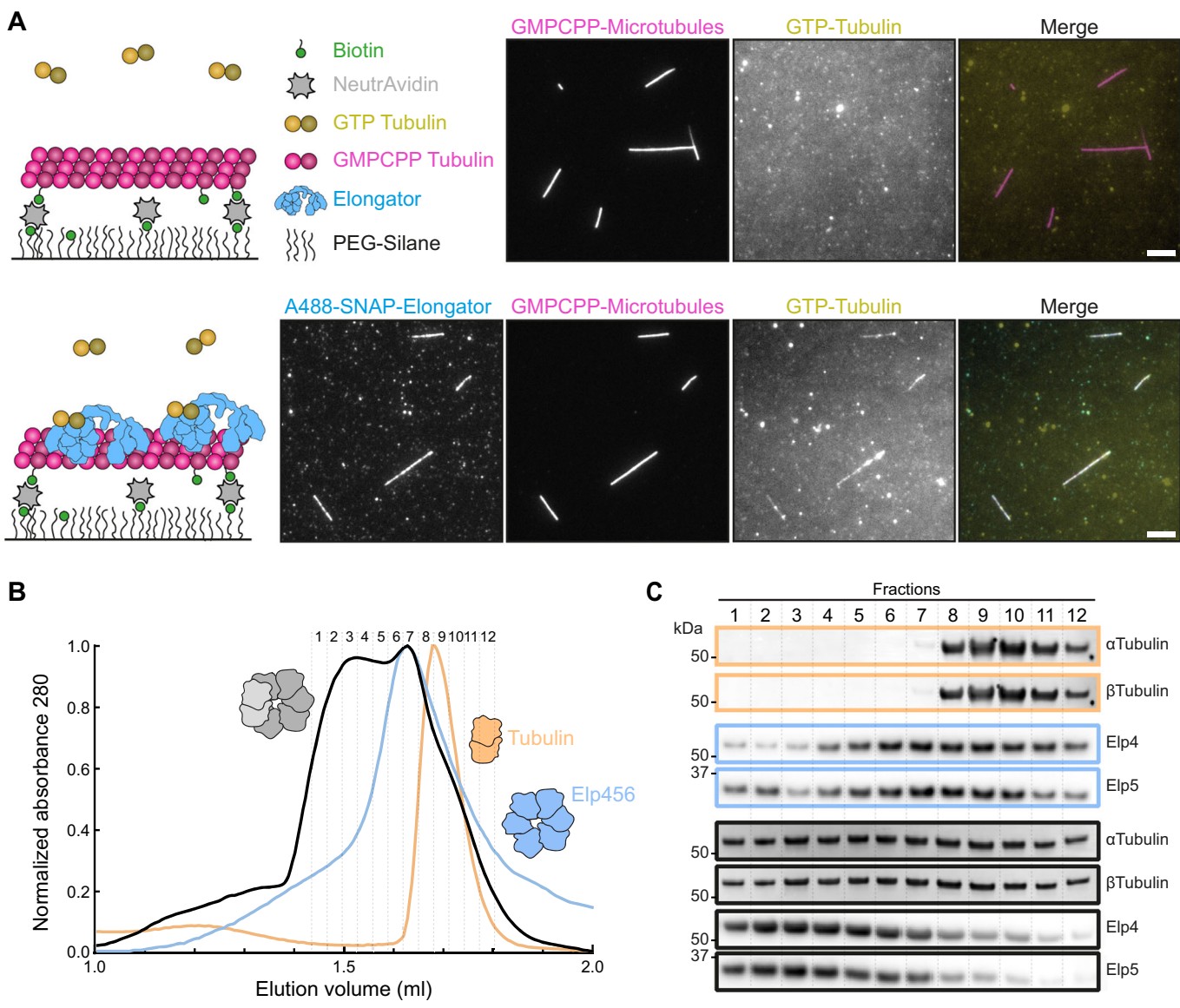

**Figure 3. Elongator complex binds to tubulin through Elp456.**

(A) While bound to GMPCPP-stabilized microtubules, the Elongator complex can recruit additional GTP-tubulin heterodimers. Top row: preformed rhodamine-labelled, GMPCPP-stabilized microtubules incubated with HiLyte647-labelled GTP-tubulin (16.7 μM GTP-tubulin, 10% HiLyte647-tubulin). Bottom row: preformed Rhodamine-labelled, GMPCPP-stabilized microtubules incubated first with AlexaFluor488-SNAP-Elongator complex, then with HiLyte647-labelled GTP-tubulin as above. (B) Elp456 binds to recombinant *Drosophila* α1β1-tubulin heterodimers. Size-exclusion (Superose 6) elution profiles of tubulin (orange), Elp456 (blue) and a mixture containing both proteins in a 1.2:1 ratio (tubulin:Elp456, black). The Elp456-tubulin complex elutes earlier than Elp456, indicating a higher molecular weight. (C) All elution profiles shown in (B) were analysed by western blot with the indicated antibodies, confirming a shift in the protein elution patterns. Scale bars = 5 μm. Source data are available online for this figure.

## Elp456 binds to αβ-tubulin heterodimers

We noticed in our reconstitution of microtubule dynamics in the presence of the full Elongator complex that HiLyte 647-αβ-tubulin signal could be seen diffusing together with Alexa488-SNAP-Elongator on microtubules (Fig. EV1C, magenta arrows). This suggested that Elongator might be able to interact simultaneously with both microtubules and soluble αβ-tubulin heterodimers, reminiscent of the behaviour of TOG-domain containing proteins (Brouhard et al, 2008; Lawrence et al, 2018). To verify this, we added the different proteins into our reconstitution assay sequentially,

as opposed to simultaneously. We first incubated GMPCPP-stabilized microtubules with Alexa 488-SNAP-Elongator. Then, an excess of Hilyte 647-αβ-tubulin was added and the chamber was imaged by TIRFM (Fig. 3A). We found that Hilyte 647-αβ-tubulin signal could be observed decorating microtubules, but only in the presence of Elongator (Fig. 3A). This establishes that Elongator can interact with both microtubules and tubulin.

However, Elongator does not contain any TOG-domain, or other sequence motives predicted to interact with unpolymerized, αβ-tubulin heterodimers. We thus wondered if Elp456, whose role in Elongator function is still not fully understood (Jaciuk et al,

2023; Gaik et al, 2022a, 2022b), could bind to tubulin using as-of-yet uncharacterized surfaces. For this, we purified the *Drosophila* Elp456 subcomplex from *E. coli* (Appendix Fig. S3A,B) and investigated its interactions with tubulin. Importantly, using size-exclusion chromatography, we found that *Drosophila* Elp456 binds to recombinant *Drosophila* α1β1-tubulin heterodimers (Fig. 3B,C, see also Appendix Fig. S3F for controls) in solution. We confirmed this result by immunoprecipitation using the Protein C (PC) affinity tag present on our recombinant tubulin (Appendix Fig. S3C,D). To determine if the binding of αβ-tubulin heterodimers was specific to Elp456, we assessed Elp123 ability to bind tubulin under similar conditions. Critically, although Elp123 did bind to microtubules (Fig. 2A; Appendix Fig. S1D), it did not bind to αβ-tubulin heterodimers (Appendix Fig. S3E,G).

All together, these results reveal a clear separation of function for the two Elongator sub-complexes: Elp123 binds to microtubules (Fig. 2A), but not to free (GTP-) tubulin (Appendix Fig. S3E,G), and, conversely, Elp456 binds to free tubulin (Fig. 3B) but not microtubules (Planelles-Herrero et al, 2022).

## Elongator binding to both microtubules and tubulin is needed for microtubule stabilization

We reasoned that the effect of Elongator on microtubule stabilization must require both the microtubule- and the αβ-tubulin-binding moieties, since, unlike the full Elongator complex, neither Elp123 (Fig. 2) nor Elp456 (Fig. 4B) on their own showed any effect on microtubule dynamics. We sought to test this hypothesis by investigating the effects on microtubule dynamics of a reconstituted full Elongator, which was generated by mixing the Elp123 and Elp456 subcomplexes together after purification (Fig. 4).

We first confirmed that the full Elongator complex can be reconstituted on GMPCPP microtubules by sequentially adding Elp123 then Elp456 subcomplexes, and, more importantly, that the reconstituted complex is functional as assessed by its ability to capture αβ-tubulin heterodimer once bound to the microtubule lattice (Fig. 4A). For this, we sequentially added in the same chamber fluorescently labelled AMCA-GMPCPP-microtubules, 488-SNAP-Elp123, mScarlet-Elp456 followed by Alexa647-αβ-tubulin. Strikingly, tubulin heterodimers could be recruited onto microtubules by the reconstituted full Elongator (i.e. Elp123+Elp456, see Figs. 4A and EV2 for drop out controls), which therefore mimics the behaviour observed with the native full Elongator complex purified from *Drosophila* cells (Fig. 3A). As expected, in the absence of Elp456, tubulin was not recruited to microtubules (Fig. EV2A), in line with the fact that Elp123 alone does not bind to tubulin (Appendix Fig. S3E,G). Note that at the high concentrations used in this experiment, a very weak mScarlet-Elp456 signal could be detected on microtubules, but this was markedly less seen than in the presence of Elp123 (9.3-fold less fluorescence intensity, Fig. EV2A, third versus first row, Fig. EV2B for quantification). Together, these results confirm that the full Elongator complex can bind simultaneously to both microtubules and αβ-tubulin heterodimers via Elp123 and Elp456, respectively.

We next investigated the effects of the reconstituted Elongator complex on microtubules dynamics. Remarkably, when both Elp123 and Elp456 were added to the assay, microtubules grew significantly faster compared to the buffer alone or the two individual sub-complexes (Fig. 4B,D). With the reconstituted Elongator complex,

microtubule growth speed increases ~1.35-fold at the plus end in the presence of Elp123 and an excess of Elp456 ($1.002 \pm 0.009\,\mu m/min$ to $1.380 \pm 0.013\,\mu m/min$), which is remarkably similar to the ~1.4-fold increase we reported for the full Elongator complex purified from *Drosophila* cells (Planelles-Herrero et al, 2022). A similar effect was observed at the minus-ends, again matching the ~1.2-fold increase we reported for the full Elongator complex (Planelles-Herrero et al, 2022) (Appendix Fig. S2A, control: $0.348 \pm 0.008\,\mu m/min$, $n = 115$; Elp456: $0.329 \pm 0.011\,\mu m/min$, $n = 72$; Elp123: $0.343 \pm 0.010\,\mu m/min$, $n = 138$; Elp123+Elp456: $0.408 \pm 0.011\,\mu m/min$, $n = 176$; Elp123+2xElp456: $0.472 \pm 0.011\,\mu m/min$, $n = 133$). Furthermore, the microtubule lifetime was increased in the presence of both sub-complexes (Fig. 4C; Appendix Fig. S2A,B), again recapitulating the effect observed with the full Elongator complex (Planelles-Herrero et al, 2022). Altogether, these results suggest that the effect of Elongator on microtubule dynamics require both its microtubule and tubulin heterodimer binding activities via its Elp123 and Elp456 subcomplexes, respectively.

## Coordinated action between subcomplexes is critical for microtubule stabilization

We showed that Elp123 binds to microtubules (Fig. 2; Appendix Fig. S1), Elp456 binds to αβ-tubulin heterodimers (Fig. 3B; Appendix Fig. S3C,D) and that both activities are required to reconstitute the activity of the full complex (Fig. 4). Since Elongator is recruited to the growing ends of microtubules (Fig. 1), this suggests a mechanism whereby Elongator would increase microtubule growth speed by bringing additional free αβ-tubulin heterodimers close to the tip of the microtubules. This would increase the local concentration of tubulin at the growing ends, thereby increasing the growth speed of microtubules and decreasing their catastrophe rate, which could explain Elongator's effects on microtubule dynamics (Fig. 4E, left). To test this hypothesis, we performed microtubule polymerization experiments close to the critical tubulin concentration required for microtubule elongation from seeds. In our experimental conditions, little to no microtubule polymerization could be observed at a concentration of GTP-αβ-tubulin heterodimers of $6\,\mu M$ (Fig. EV3A,B), in agreement with previously calculated critical concentrations for porcine brain tubulin (Walker et al, 1988). Importantly, slow microtubule growth could be observed when $25\,nM$ SNAP-Elongator was added (Fig. EV3A,B), consistent with Elongator increasing the local concentration of tubulin beyond the critical concentration. Elongator would thus stabilize microtubules through a mechanism reminiscent of microtubule polymerases such as XMAP215 and CLASPs (Al-Bassam et al, 2007; Brouhard et al, 2008; Lawrence et al, 2018). Importantly, we showed that the Elp456-αβ-tubulin pre-formed complex could be recruited to Elp123 already bound to microtubules (Fig. EV2C), further supporting this mechanism of action. We thus decided to probe the importance of the coupling between the activities of Elp123 and Elp456 for the effect of the full complex on microtubules (Appendix Fig. S4).

We sought to produce a mutant of Elp456 displaying weaker affinity towards Elp123, whilst maintaining full αβ-tubulin binding capacity, uncoupling the activities of the two Elongator sub-complexes (Fig. 5E, right). We first performed cross-linking mass spectrometry on the full Elongator complex to identify residues at the interface between *Drosophila* Elp123 and Elp456 (see "Methods"). This approach yielded several fragments containing

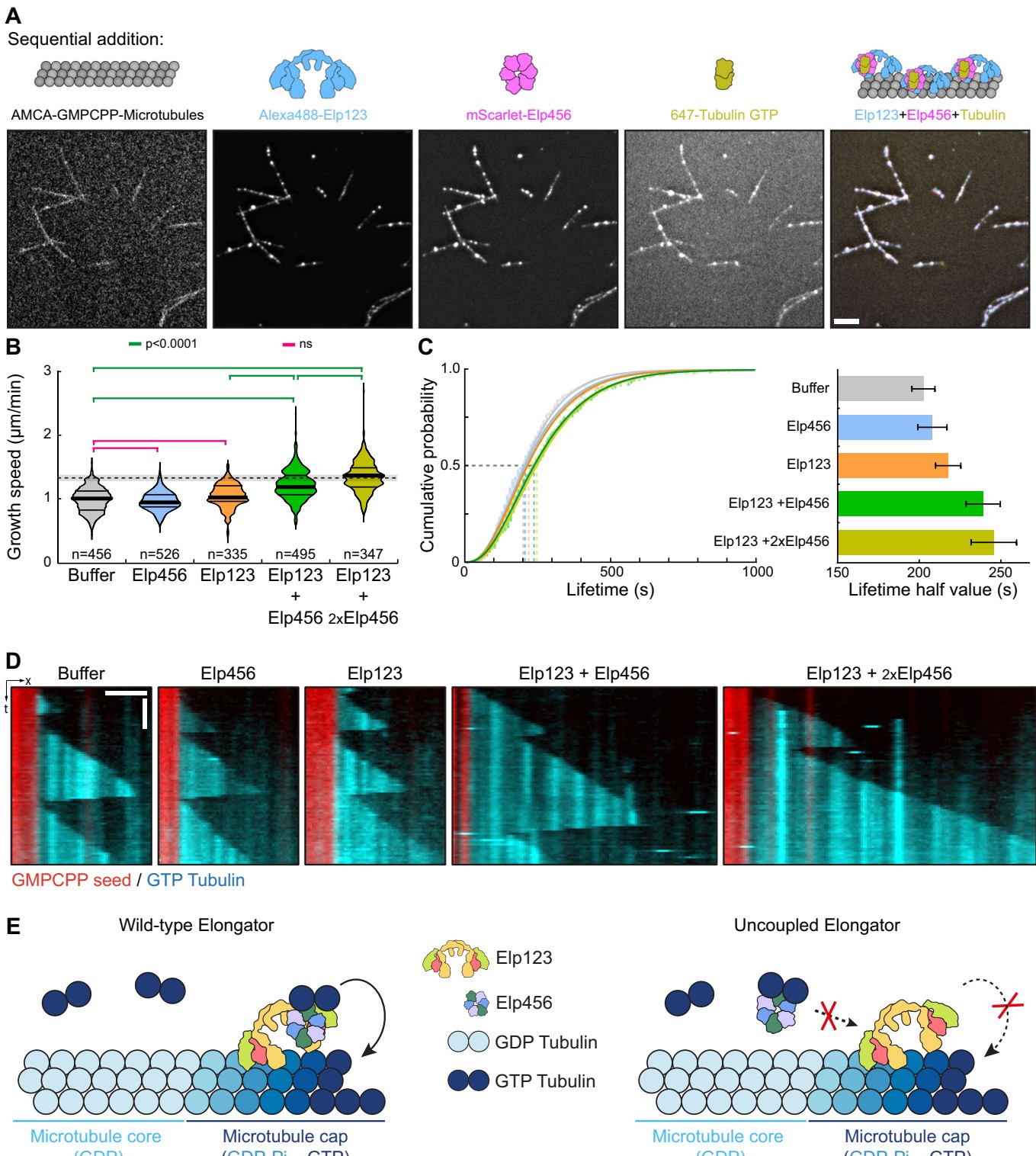

cross-linked lysine residues (Appendix Fig. S5A–D and Appendix Table S1). We then cross-referenced these results with the localization of various Elongator disease-related mutations (Krassowski et al, 2018) (Appendix Fig. S5E), conservation analysis (Appendix Fig. S5A–C), and AlphaFold2 Multimer (Jumper et al, 2021; Evans et al, 2022) modelling (see "Methods") to infer the surfaces of Elp456 involved in Elp123 binding (Appendix Fig. S5D).

This allowed the rational design of a mutant harbouring 3-point mutations in each subunit (18-point mutations per Elp456 hexameric ring: Elp4 K364E, R397E, E410R; Elp5 K151E, K179E, T181A; Elp6 K119E, S200W, K228E; Appendix Fig. S5A–C,E). Importantly, we made sure to preserve the solvent-exposed surface of Elp456 (i.e., opposite to the surface that binds to Elp123) to assure that this this mutant could retain full binding to αβ-tubulin

**Figure 4.   Binding to both microtubules and free tubulin is essential for microtubule stabilization.**

(A) Sequential reconstitution of a full Elongator complex onto microtubules using independently-purified Elp123 and Elp456 subcomplexes. Indicated components were sequentially added starting with AMCA-labelled GMPCPP-microtubules, followed by AlexaFluor488-SNAP-Elp123, mScarlet-Elp456 and finally HiLyte647-tubulin. Note that all three proteins can be observed on microtubules. See Fig. EV2A for drop out controls. For visualisation purposes, images were processed with a Wavelet "à trous" denoising filter. (B, C) Effect of the indicated conditions on the growth speed (B) and lifetime (C) of microtubules at the plus end imaged by TIRFM in the presence of 16 µM GTP-tubulin (10% HiLyte 647 labelled). Elongator concentrations are: 100 nM Elp456 ("Elp456"), 50 nM (His)$_6$-PC-SNAP-Elp123 ("Elp123"), 50 nM Elp456 + 50 nM (His)$_6$-PC-SNAP-Elp123 ("Elp123+Elp456"), and 100 nM Elp456 + 50 nM (His)$_6$-PC-SNAP-Elp123 ("Elp123+2xElp456"). *n*, number of microtubule-growing events analysed. Similar results were observed at the minus end (see text and Appendix Fig. S2A,B). (B) *P* values for a two-tailed Kruskal–Wallis test followed by Dunn's multiple comparison test are indicated (exact ns *P* values: Buffer vs Elp456 0.0612, Buffer vs Elp123 0.0573). Dashed line represents an increase of ~1.4 in the speed of microtubule growth (Planelles-Herrero et al, 2022). Thick line, median; thin lines, quartiles. (C) Microtubule lifetime estimate ± error from the bootstrapped mean lifetimes (see methods) are indicated in the right panel. Number of microtubule-growing events analysed as in (B). (D) Representative kymographs of the indicated conditions. (E) Proposed mechanism of Elongator's mode of action on microtubules. Elongator binds to the microtubule tips via Elp123 and to αβ-tubulin heterodimers via Elp456. This increases the local tubulin concentration at the microtubule tip (Fig. EV3A,B), thereby increasing the polymerization speed and decreasing the catastrophe rate. Note that as expected, if the interaction between Elp123 and Elp456 is perturbed (Appendix Fig. S4), the effect on microtubule dynamics is reduced. Scale bars = 5 µm (A), 2 min/2 µm (D). Source data are available online for this figure.

heterodimers. Hereafter, we refer to this mutant as Elp456 "solo" due to its predicted weaker binding to Elp123.

Elp456 *solo* could be readily purified from bacteria using the same protocol as for *wild-type* Elp456 (see "Methods"). We then characterized the biochemical properties of Elp456 *solo* (Appendix Figs. S4 and S5), namely its ability to bind Elp123 and to αβ-tubulin heterodimers. First, we confirmed that the binding to recombinant *Drosophila* tubulin is not compromised in the *solo* mutant ($K_d$: 1.6 ± 0.5 µM vs *wild-type* 1.2 ± 0.8 µM, Appendix Fig. S5F). Critically, however, the binding of Elp456 *solo* to Elp123 was significantly reduced compared to the *wild-type* (>4-fold reduction, from $K_d$ of 15.5 ± 3.7 nM to 67.1 ± 9.4 nM, Appendix Fig. S4A). Importantly, the measured $K_d$s are in the range of protein concentrations used in our in vitro experiments (i.e., 50 nM), thus we do expect a difference of occupancy in our assays. Indeed, at the concentrations used (i.e., 50 nM of Elp123 and 100 nM of Elp456), a ~ 35% reduction in the total amount of Elp456 bound to Elp123 is expected (see "Methods")). We then verified this prediction by using the Elp456 *solo* mutant in microtubule dynamics assays (Appendix Fig. S4B–D) and a range of concentrations, including data points below and above the calculated $K_d$. Strikingly, the reconstituted Elongator containing Elp456 *solo* had an intermediate effect between the buffer alone and reconstituted Elongator containing *wild type* Elp456, both for the microtubule growth speed (Appendix Fig. S4B) and the lifetime (Appendix Fig. S4C,D) of microtubules. We observed a dose-dependent effect when increasing amounts of *wild type* or *solo* Elp456 were added (Appendix Fig. S4B–D). At concentrations below the $K_d$, the *solo* mutant, together with Elp123, does not display any effect on the microtubules. At higher concentrations, however, the weaker affinity is compensated and the *solo* mutant increases both the speed of growth (Appendix Fig. S4B) and the lifetime (Appendix Fig. S4C,D) of microtubules. Importantly, the effect of the *solo* mutant is always weaker than that observed for wild-type Elp456, although this difference is smaller at very high concentrations (i.e. 200 nM), that is, greatly above the $K_d$. Similar results were observed for the minus ends (Appendix Fig. S2C,D). These intermediate, rather than total, effects are consistent with the relative affinity of Elp456 *solo* towards Elp123 (Appendix Fig. S4A). Note that a recently published structure of the full *S. cerevisiae* and *M. musculus* Elongator complexes (Jaciuk et al, 2023) confirmed that most of the residues mutated in *D. melanogaster* Elp456 solo are indeed expected to be involved in the binding to Elp123, validating our predictions. For example, the Elp6 K199E and S200W

mutations lie at the interface between Elp3 and Elp6 (Appendix Fig. S5E, inset), and the introduction of a negative charge and a bulky tryptophan, respectively, are expected to greatly perturb this interface.

Altogether, our results establish that a tight coupling between both sub-complexes is required for Elongator's ability to stabilize microtubules, whereby the Elp456-αβ-tubulin complex binds to Elp123 and thereby increases the local concentration of tubulin in the vicinity of the microtubule tip (Fig. 4E).

## Elongator can discriminate between microtubules and tRNA

The best-characterized function of Elongator in eukaryotic cells involves its interaction with tRNAs, a process critical for maintaining optimal translation rates and protein homeostasis (Abbassi et al, 2020). Since both tRNAs and tubulin have negatively charged surfaces, it is conceivable that they may compete for binding to the same sites on Elp123 and/or Elp456. We thus extended the characterization of the effect of the Elongator complex on microtubules in the presence of saturating and physiological concentrations of tRNA (100 µM), as well as salt (50 mM KCl). Importantly, in these conditions, Elongator still binds to GMPCPP-stabilized microtubules (Fig. EV3D) and tracks the growing end of dynamic microtubules (Fig. EV3E). Critically, these experiments were performed in the presence of 2.5 mM MgATP, a reported optimal concentration to favour the interaction between Elp456 and tRNAs (Jaciuk et al, 2023; Glatt et al, 2012). Thus, Elongator's ability to directly bind to microtubules is not affected by physiological concentrations of tRNA, at least in vitro.

To confirm Elongator's ability to bind to microtubules in physiological conditions, we used total cell extracts of *Drosophila* S2 cells expressing fluorescently labelled, SNAP Elongator at endogenous levels (see methods). Compared to control, untransfected S2 cell extracts, SNAP-Elongator extracts lead to GMPCPP-stabilized microtubules fully decorated with fluorescent SNAP-ligand (Fig. EV3F). This demonstrates that even in presence of physiological concentrations of tRNA and all other endogenous microtubule-associated proteins present in S2 cells, Elongator has a strong-enough affinity for microtubules not to be outcompeted. Conversely, we also verified Elongator's ability to bind to tubulin dimers in cells. Using S2 cells stably expressing PC-tagged α-tubulin 84B, the main *Drosophila* α-tubulin isotype (Kalfayan and Wensink, 1982), Elp2 and Elp4 subunits could be detected after

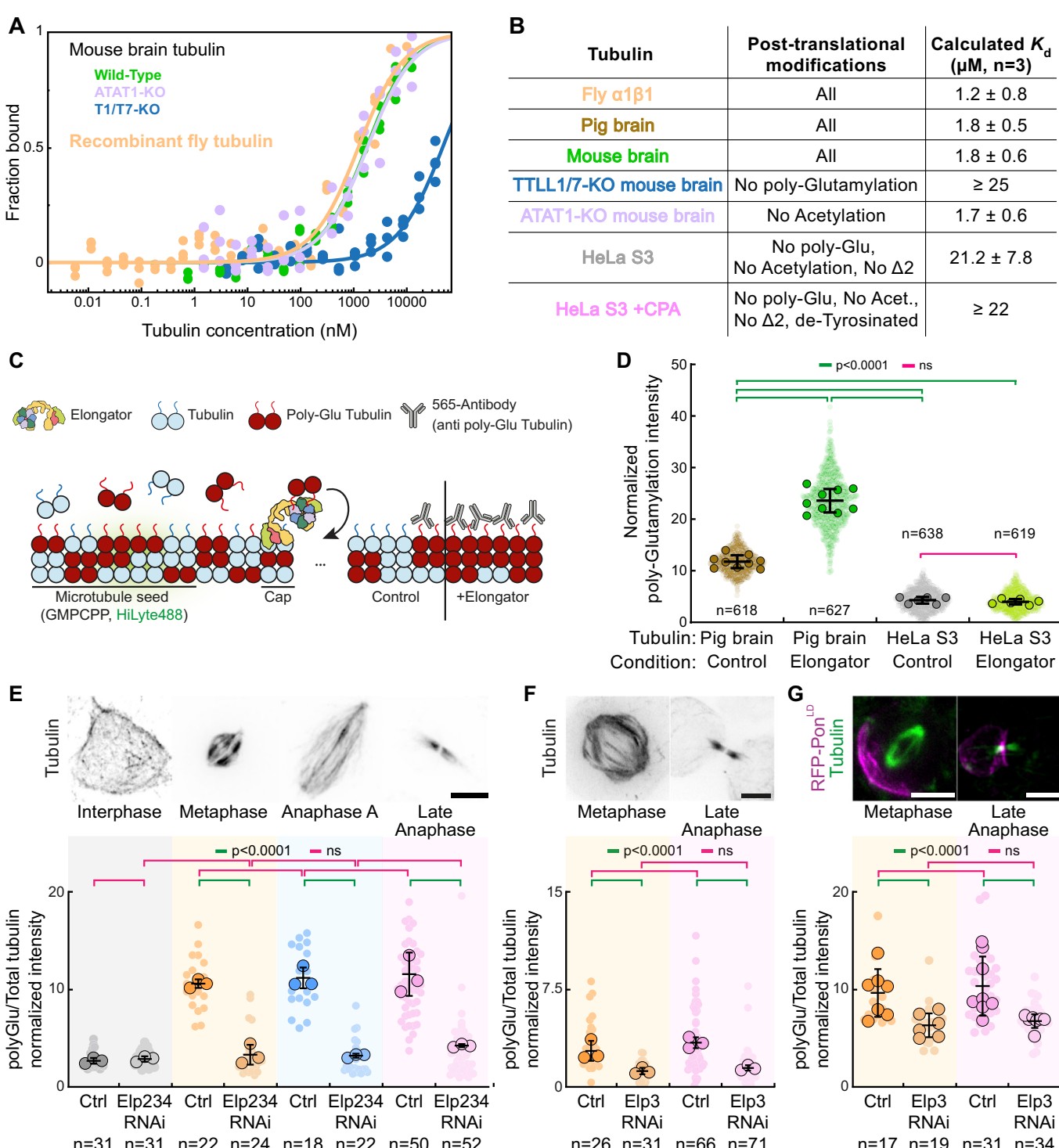

PC-α-tubulin immunoprecipitation (Fig. EV4A), suggesting that both the Elp123 and Elp456 subcomplexes co-precipitate with tubulin dimers. Since high salt concentrations have been reported to disrupt the assembly of the full Elongator complex in vitro (Setiaputra et al, 2017), we supplemented the cell lysate with 300 mM KCl, as well as nocodazole to depolymerize microtubules (Fig. EV4A). In both cases, only Elp4 could be detected, confirming that only the Elp456 subcomplex interacts with αβ-tubulin heterodimers. We then confirmed in vitro (Fig. EV3C) that GFP-

Elp456 binds to αβ-tubulin heterodimers with similar affinity in high tRNA and MgADP buffers ($K_d = 2.9 \pm 0.8$ μM) compared to BRB80 ($K_d = 1.2 \pm 0.8$ μM). Importantly, we confirmed that our GFP-Elp456 subcomplex is functional and could bind to tRNA in these conditions with similar affinity (Fig. EV4B, $K_d = 19.7 \pm 4.0$ nM) as previously reported in buffers containing 2.5 mM MgATP (Jaciuk et al, 2023).

Finally, we found that Elongator stabilizes microtubules in 100 μM tRNA, 50 mM KCl and 2.5 mM MgADP

**Figure 5. Elongator is a tubulin polymerase selective for polyglutamylated monomers.**

(A, B) Elp456 binds preferentially to polyglutamylated tubulin. (A) Affinity measurement between eGFP-Elp456 and indicated αβ-tubulin heterodimers using microscale thermophoresis (see methods). Estimated dissociation constant ($K_d$) values are indicated (mean ± s.d.; $n = 3$). See also Fig. EV5A. (B) Estimated dissociation constants for the indicated tubulins. Elp456 binds with similar affinity to recombinant *Drosophila* α1β1-tubulin, pig brain tubulin, mouse brain tubulin and mouse brain tubulin lacking K40 acetylation. However, the binding to tubulin purified from TTLL1/7-double KO mouse brains (lacking polyglutamylation) and HeLa S3 cells (lacking polyglutamylation, K40 acetylation and Δ2 deletion (Souphron et al, 2019; Barisic et al, 2015)) is similar (≥ 20 µM) and ~18 times lower than other tubulins. Note that removal of the last tyrosine does not seem to have an effect in Elp456 binding (CPA treated tubulin purified from HeLa S3 cells, see methods). (C) Experimental assay to detect polyglutamylation in polymerized microtubules in vitro. See methods for details about the quantification and signal normalization. (D) Microtubules polymerized in the presence of Elongator are significantly enriched in polyglutamylated tubulin when using a 50:50 pig brain tubulin:HeLa S3 tubulin mix. Correspondingly, when using HeLa S3 tubulin, which is not polyglutamylated (Barisic et al, 2015; Souphron et al, 2019), no effect is observed. Error bars are mean ± standard deviation for quantified intensities. $n$, microtubules quantified (transparent dots); Full dots represent the average of all microtubules from a field of view from three independent experiments. $P$ values for an Ordinary one-way ANOVA test followed by two-tailed Turkey multiple comparison test are indicated (exact ns $P$ values: 0.9854. (E–G) Elongator depletion (E: Elp2, Elp3 and Elp4; F, G: Elp3) results in decreased polyglutamylation levels at the spindle. mean ± standard deviation. $P$ values for an Ordinary one-way ANOVA test on independent experiments ($N = 3$) followed by two-tailed Turkey multiple comparison test are indicated. (E) *Drosophila* S2 cells depleted for Elongator (see also depletion controls in Fig. EV5C) have significantly decrease levels of polyglutamylated tubulin at the mitotic spindle. Exact ns $P$ values, left to right: 0.99999944, 0.99916415, 0.99461841, 0.99999993, >0.99999999, 0.78773591). Similar results could be detected in mouse NIH/3T3 cells (F, exact ns $P$ values: Control Metaphase vs Control Late Anaphase 0.4076, RNAi Metaphase vs RNAi Late Anaphase 0.9171), and in vivo during asymmetric cell division of SOP cells during *Drosophila* development (G exact ns $P$ values: Control Metaphase vs Control Late Anaphase 0.9225, RNAi Metaphase vs RNAi Late Anaphase 0.9841). Scale bars = 5 µm. Source data are available online for this figure.

(Fig. EV4E,F), consistent with the fact that Elongator binds to both microtubules (Fig. EV3D–F) and tubulin (Figs. EV3C and EV4A) in these conditions. Specifically, we found that microtubule growth at the plus-end was increased (Fig. EV4E) compared to the control in these conditions (1.024 ± 0.012 µm/min to 1.450 ± 0.013 µm/min) and similar to our previous measurements without tRNA (1.380 ± 0.013 µm/min). Similarly, microtubule stability is also increased by Elongator in these conditions (Fig. EV4F), confirming that the presence of saturating concentrations of tRNA does not prevent Elongator's action on microtubules. Consistently, similar effects were observed at the minus-ends: control: 0.366 ± 0.017 µm/min, $n = 22$; Elongator: 0.477 ± 0.019 µm/min, $n = 58$; see also Appendix Fig. S2E,F).

On the other hand, however, we found that microtubule binding by Elongator prevents the tRNA-Elongator interaction (Fig. EV4C). Indeed, using TIRFM we observed that microtubules decorated by Elongator could not recruit fluorescently labelled tRNA, suggesting that the conformation adopted by Elongator on the microtubules is not compatible with tRNA binding. Importantly, Elongator alone (i.e., in the absence of microtubules) was able to bind to fluorescent-tRNA (Fig. EV4D). Since tRNA binding induces a conformational change on Elp123 (Dauden et al, 2019), particularly on the Elp1 subunit, it is conceivable that binding to microtubules prevents this rearrangement, and thus efficient tRNA binding.

Together, our data suggests that Elongator binds and stabilizes microtubules in physiological conditions containing its other highly abundant substrate, tRNA. However, upon binding to microtubules, Elongator loses its ability to interact with tRNA, functionally separating its microtubule stabilizing and tRNA modifying activities. Understanding the regulatory mechanisms that switch Elongator between microtubule binding and tRNA modification is therefore a critical question for future studies.

## Elongator can discriminate tubulin with specific post-translational modifications

We next investigated how Elongator can recognise and recruit free αβ-tubulin dimers at the molecular level, since Elongator does not contain any predicted tubulin- or microtubule- binding surfaces. We first tested αβ-tubulin binding using human Elp456 to verify that Elongator's effects on microtubules are not *Drosophila* specific. For

this, we purified recombinant human Elp456 (Appendix Fig. S6A) and used it in our reconstitution assays together with *Drosophila* Elp123. Strikingly, the reconstituted chimeric fly Elp123 /human Elp456 recapitulates the effect measured with fly Elp456 both for the microtubule growth speed (Appendix Fig. S6A) and lifetime (Appendix Fig. S6C,D). This strongly suggests that microtubule stabilization is not exclusive of the *Drosophila* Elongator complex.

We next characterized the binding of Elp456 to tubulins from different sources to identify the specific tubulin surfaces involved in this interaction. We first tested the binding of Elp456 to recombinant *Drosophila* α1β1-tubulin, pig brain tubulin and wild-type mouse brain tubulin, which are highly modified (polyglutamylated, acetylated and partially detyrosinated) and found that Elp456 binds with similar affinity to all of them ($K_d = 1.2 ± 0.8$ µM, $1.8 ± 0.5$ µM and $1.8 ± 0.6$ µM, respectively, Figs. 5A,B and EV5A,B). We next tested the binding of Elp456 to tubulin purified from HeLa S3 cells, which is well established to be almost entirely free of polyglutamylation, acetylation and de-tyrosination (i.e. non-modified) (Barisic et al, 2015; Souphron et al, 2019). Strikingly, we detected an ~18-fold reduction in the affinity of Elp456 towards HeLa S3 tubulin ($K_d$: $21.2 ± 7.8$ µM, $n = 3$) compared to the highly modified tubulins (Figs. 5B and EV5A,B). To pinpoint the specific PTM controlling the interaction between tubulin and Elp456, we used tubulin purified from brains of mice lacking enzymes responsible for tubulin modifications (see "Methods"). Remarkably, tubulin purified from TTLL1/7-double KO mouse brain, lacking polyglutamylation as previously established (Souphron et al, 2019), displays a much weaker affinity towards Elp456 ($K_d ≥ 25$ µM, $n = 3$, Fig. 5A,B) than tubulin from wild-type mice ($1.8 ± 0.6$ µM, $n = 3$). However, using ATAT1-KO mouse brain tubulin, lacking detectable α-tubulin acetylation (Souphron et al, 2019), we showed that Elp456 binding is independent of α-tubulin acetylation ($K_d = 1.7 ± 0.6$ µM, Fig. 5A,B). Similarly, Elp456 binding was independent of the tyrosination state of the α-tubulin C-terminal tail, since HeLa S3 tubulin (fully tyrosinated) and its de-tyrosinated version (obtained using carboxypeptidase A to remove the C-terminal tyrosine) display almost identical binding curves ($K_d ≥ 22$ µM, $n = 3$, Fig. EV5A). This suggests that the Elp456-tubulin interaction specifically requires polyglutamylated αβ-tubulin C-terminal tails (Fig. 5B). Note that the absence of post-translational modifications in the

C-terminal tail greatly reduces, but not completely abolishes, the interaction with Elp456 (Figs. 5B and EV5A, HeLa S3 and HeLa S3 + CPA), which could explain the weak binding of Elp456 towards microtubules we observed in TIRF microscopy (Fig. EV2A,B). These results suggest that the body of the tubulin molecule likely also plays a role in the interaction, as otherwise full binding of Elp456 to microtubules would be expected if only the C-terminal tails participated in the interaction, independently of the polymerization state of tubulin.

To gain insights into the surfaces of Elp456 involved in the binding, we used crosslinking coupled with mass spectrometry (XL-MS). For this, we used a synthetic peptide containing 10 linear glutamate resides ($Glu_{10}$), mimicking a polyglutamylated branch of a tubulin tail, in order to minimise the overwhelming heterogeneity present in tubulin tails. Several crosslinks were detected between Elp456 and the $Glu_{10}$ peptide using two different crosslinkers, EDC (1-ethyl-3-(3-dimethylaminopropyl)carbodiimide hydrochloride) and SDA (succinimidyl 4,4'-azipentanoate). Importantly, mapping the crosslinked residues on the Alphafold2 model of fly Elp456 (Appendix Fig. S7A) revealed that they cluster together on the same surface of the Elp456 ring, away from the centre of the molecule where tRNA binding was previously mapped (Glatt et al, 2012), explaining the absence of competition between tubulin and tRNA (Appendix Fig. S7 and Appendix Table S2). Interestingly, the crosslinked residues are preferentially located at the "edges" of the Elp456 ring, close to the interface with the Elp123 subcomplex (Appendix Fig. S7A, side views), yet not directly participating in the interaction (Appendix Fig. S7B).

## Elongator is a polyglutamylation-selective tubulin polymerase

Whilst Elp456 specifically binds to polyglutamylated tubulin, the interaction of Elp123 with the microtubule lattice is independent of PTMs. Indeed, we observed that Elp123 is still able to bind to microtubules assembled from HeLa tubulin or pig brain tubulin partially digested with subtilisin to remove the C-terminal tail once polymerised (Appendix Fig. S1E,F). Therefore, whilst binding of Elp123 to microtubules is independent of their post-translational modifications, the binding of Elp456 is not. Given the molecular mode of action of Elongator (Figs. 1–5), this surprising property could endow Elongator with the unique ability to selectively enrich newly polymerised microtubules with polyglutamylated tubulin without requiring an enzymatic tubulin-modifying activity. In other words, by binding to microtubules irrespective of their PTMs, but selectively elongating them with polyglutamylated tubulin subunits, Elongator could change the tubulin code of *dynamic* microtubules. This stands in contrast to the mode of action of the modifying enzymes such as members of the TTLL family, which preferentially modify long-lived, stable microtubules (Garnham et al, 2015a; van Dijk et al, 2007; Mukai et al, 2005).

To test this hypothesis, we established an assay to quantitatively measure the incorporation of polyglutamylated tubulin in polymerized microtubules (Fig. 5C, see also "Methods"). Briefly, microtubules were elongated from stable, HiLyte488-labelled GMPCPP seeds in the presence or absence of Elongator, from a non-fluorescent source of heterodimers composed of a 50:50 mixture of HeLa S3 tubulin, which is weakly polyglutamylated, and

pig brain tubulin, which is heavily polyglutamylated. After polymerization, the microtubules were stabilized with Taxol, incubated with fluorescently labelled antibodies against polyglutamylated tubulin and imaged by TIRFM. This allowed us to measure the amount of polyglutamylation within the microtubules with and without Elongator (Fig. 5C). To normalize for sample-to-sample differences in fluorescence, the polyglutamylated tubulin antibody signal was normalized against the HiLyte488 signal from the microtubule seeds (see methods). Note that we use a 50:50 HeLa S3/pig brain tubulin to decrease the ratio of polyglutamylated tubulin in the initial pool and thereby facilitate the detection of any enrichment of polyglutamylated tubulin in the resulting microtubules.

Strikingly, we could detect a clear enrichment in polyglutamylated tubulin in microtubules assembled in the presence of Elongator compared to the control (normalized intensity = $23.58 \pm 4.93$, $n = 627$ versus $11.70 \pm 2.13$, $n = 618$ for the control, Figs. 5D and EV5E). As expected, microtubules grow longer and at higher density in the presence of Elongator (Fig. EV5E,F), as expected in the presence of a microtubule stabilizing activity, threfore confirming that indeed the Elongator complex participated in the elongation of microtubules. Critically, in control conditions in which only HeLa S3 tubulin was available, microtubule density was similar in the presence and absence of Elongator (Fig. EV5G,H), which is in line with our findings that Elongator displays low affinity towards non-polyglutamylated tubulin (Figs. 5B and EV5A). Furthermore, microtubules polymerised with HeLa S3 tubulin alone showed low polyglutamylation-signal in both the absence ($4.35 \pm 1.93$, $n = 638$) and presence ($4.08 \pm 1.8$, $n = 619$) of Elongator (Fig. 5D), confirming the low abundance of polyglutamylated tubulin in the sample. Note that the higher density of microtubules in HeLa S3 tubulin conditions reflects an intrinsically higher polymerization rate of HeLa S3 tubulin compared to pig brain tubulin, independent of Elongator (Fig. EV5G,H). This confirms that Elongator does preferentially use polyglutamylated tubulin as a substrate and, therefore, Elongator does not stabilize microtubules when the available tubulin pool is not polyglutamylated. Altogether, these results highlight an unexpected property of the Elongator complex, namely that it functions as a microtubule polymerase selective for polyglutamylated tubulin in vitro, which directly stems from its molecular mode of binding towards globular tubulin and microtubules.

Finally, we confirmed that Elongator controls the polyglutamylation levels of microtubules in vivo. Indeed, Elongator depletion results in the reduction of polyglutamylation levels at the mitotic spindle in cultured cells of different species (*Drosophila* S2, mouse NIH/3T3 fibroblasts), but also in *Drosophila* Sensory Organ Precursor (SOP) cells in developing flies (Fig. 5E–G, see Fig. EV5C,D for characterization of respective RNAi depletion efficiencies). Interestingly, the levels of polyglutamylation of stable interphase microtubules are not affected by the depletion (Fig. 5E), suggesting that Elongator's activity is more relevant in highly dynamic microtubule structures such as spindles (the turnover rate of spindle microtubules is ~9 times higher than that of interphase microtubules (Hyman and Karsenti, 1996)). Taken together, our data confirm that the activity of the Elongator complex on microtubules regulates their polyglutamylation levels, serving as a mechanism for the in vivo regulation of microtubule polyglutamylation at the mitotic spindle.

## Discussion

### Convergent evolution of the mode of action of microtubule polymerases

A key finding of our study is that Elongator directly binds to both the growing microtubule ends and to free αβ-tubulin heterodimers, and that the coupling between these two activities allows Elongator to modulate microtubule dynamics. Indeed, we found that: (i) Elongator specifically recognises the growing ends of microtubules, both at the plus and at the minus ends (Figs. 1C,E and EV1B,C); (ii) the Elp123 sub-complex binds directly to microtubules (Fig. 2, Appendix Fig. S1D,F), but this binding is not sufficient to drive microtubule stabilization (Fig. 2B,C); (iii) Elp456 binds to soluble αβ-tubulin heterodimers (Fig. 3B,C; Appendix Fig. S3C,D,F) but not microtubules; (iv) when Elp123 binds to microtubules, it can recruit Elp456, which in turn brings tubulin (Fig. 3A, Fig. 4A, Fig. EV2); (v) both Elp123 and Elp456 are needed to stabilize microtubules (Fig. 4B,D; Appendix Fig. S2A-B); and (vi) weakening the interaction between both sub-complexes results in a loss of microtubule stabilization (Appendix Figs. S4 and S2C,D). This dual binding of Elongator to microtubule ends and αβ-tubulin heterodimers explains molecularly how Elongator enhances the growth speed and thereby the persistence of microtubules by increasing the local concentration of tubulin at both ends (Figs. 4E and EV3A,B). Importantly, and whilst we detect Elongator end-binding in ~10% of the analysed growth events, our results suggest that Elongator affects most, if not all, the microtubules present in our assays, as growth speed (Fig. 4B; Appendix Fig. S2A,C) and polyglutamylation levels (Fig. 5D) distributions are unimodal, not bimodal. Thus, the reduced detection likely reflects technical limitations in our ability to detect Elongator at microtubule ends, rather than restricted Elongator activity.

It is worth noting that we detected a higher effect on microtubule stability when using an excess of Elp456 over Elp123 to reconstitute Elongator (Fig. 4B–D; Appendix Fig. S4B–D; Appendix Fig. S2A–D). However, when using the full Elongator complex purified from *Drosophila* cells, no dose dependence was observed (Planelles-Herrero et al, 2022). Since the purified full Elongator complex has a defined Elp123:Elp456 ratio, this hints at a mechanism in which Elp456 is the limiting factor. One plausible hypothesis is that Elp456 acts as a "shuttle" to bring tubulin dimers to Elp123, located at the growing end of microtubules (Fig. 4E). In this scenario, a higher availability of Elp456 would bring more tubulin to the microtubules, further increasing their speed of growth and lifetime. Additionally, it has been described that Elongator can recruit an extra Elp456 in low-salt conditions or when additional Elp456 is added (Setiaputra et al, 2017; Jaciuk et al, 2023). Together, our data suggests a mechanism in which the effect of Elongator on microtubules can be fine-tuned by controlling the amount of available Elp456, providing cells with an additional way to control the effect that the Elongator complex has on microtubules.

This mechanism of action relying on increasing the local concentration of free tubulin at microtubule tips is reminiscent of other microtubule polymerases, like CLASP and XMAP215. Like Elongator, CLASP (in the presence of EB1) and XMAP215 have indeed been shown to track microtubule ends and to recruit free tubulin once bound, thereby enhancing the incorporation of tubulin dimers to the growing ends of microtubules (Brouhard

et al, 2008; Arpag et al, 2020; Al-Bassam et al, 2007; Lawrence et al, 2018; Majumdar et al, 2018). But there is a major difference, namely that Elongator does not contain any sequence motive or domains predicted to interact with αβ-tubulin heterodimers (i.e. TOG domains) or with end-binding proteins (i.e. SxIP motifs). It is striking that a complex devoid of TOG-domains like Elongator has evolved to converge towards a similar mode of action through a completely orthogonal way, in this case using two sub-complexes to achieve the same result as having two separate TOG-domains. Presumably related to this, Elongator seems to have a distinctive mode of binding and recognising microtubule ends, combining features of both EB proteins, like nucleotide-dependent binding (Fig. 1); but also TOG-containing proteins, such as lattice-diffusion and dual microtubule- and tubulin-binding (Figs. 3, 4, and EV1B,C). This different mode of action might explain another key difference between Elongator and XMAP215, namely that while both increase the microtubule polymerisation rate, Elongator decreases the catastrophe rate, on the contrary to XMAP215, which increases it (Farmer et al, 2021). It is tempting to speculate that cells evolved multiple microtubule stabilizers with different properties because despite the relatively high physiological concentrations of tubulin in cells, the frequency of microtubule catastrophe is still relatively high at these concentrations. Proteins like CLASP, XMAP215 and Elongator that locally increase the concentration of tubulin is probably a cost-efficient way to ensure robust microtubule elongation.

### An orthogonal way of writing the tubulin code

Whilst characterizing the mechanism by which Elongator binds to microtubules and tubulin heterodimers, we found a very unexpected behaviour, namely that Elongator can discriminate between tubulin heterodimers carrying different PTMs. On one hand, the Elp456 subcomplex displays ~18-fold reduced affinity towards tubulin lacking polyglutamylation, while changes in acetylation and tyrosination had little effects (Figs. 5A,B and EV5A). On the other hand, the Elp123 subcomplex binds to polymerized microtubules independently of their PTM status (Appendix Fig. S1E,F). This made the intriguing prediction that microtubules polymerized by Elongator should become selectively enriched in polyglutamylated tubulin, which we could confirm experimentally in vitro and in vivo, in multiple species and in both isolated cells and tissues (Fig. 5). Importantly, this may explain why the effect of Elongator on microtubule speed and lifetime, whilst significant, is small when compared to the much more dramatic effects of effectors like XMAP215. Indeed, polyglutamylation has been reported to decrease both microtubule growth speed and stability (Chen and Roll-Mecak, 2023). Since the microtubules polymerized by Elongator are enriched in polyglutamylation (Fig. 5), the intrinsic properties of the microtubules are therefore partially "masking" the full magnitude of the effect of Elongator on microtubules. This could represent an evolutionary adaptation to maintain microtubule dynamics within a certain range while simultaneously increasing polyglutamylation levels, potentially to achieve specific downstream effects since polyglutamylation regulates the binding and activity of various microtubule-associated proteins (Bodakuntla et al, 2021; Valenstein and Roll-Mecak, 2016; Shin et al, 2019).

Overall, the Elongator complex showed an unprecedented behaviour for both tubulin polymerases and tubulin PTM

regulators. Indeed, it is well established that tubulin polyglutamylases, which are members of the tubulin-tyrosine ligase-like (TTLL) family, prefer modifying polymerized microtubules over free tubulin (Regnard et al, 1998; Garnham et al, 2015b). Hence, the current paradigm postulates that microtubules can only get modified *after* their assembly. To achieve spatial modification of microtubules with this paradigm, TTLL proteins must thus be targeted to the desired microtubules, for instance by other microtubule-associated proteins. This has been demonstrated for CEP41 (*Centrosomal Protein of 41 kDa*), CCSAP (*Cilia and Spindle-Associated Protein*) and PGs1 (*Tubulin Polyglutamylase complex subunit 1*) directly recruiting and/or activating TTLL enzymes to specific structures in the cells (Regnard et al, 2003; Lee et al, 2012; Bompard et al, 2018).

Our discovery that Elongator assembles dynamic microtubules specifically from polyglutamated tubulin now shifts this paradigm and offers an orthogonal way to modify microtubules. Indeed, since it can bind to microtubules regardless of their PTM composition, and that its activity biases microtubule towards polyglutamylated tubulin, Elongator effectively changes the PTM composition of microtubules as they are being polymerized. In other words, whilst most tubulin-modifying enzymes change the tubulin code of *stable* microtubules, Elongator changes the code of *dynamic* ones. This property could be important for very dynamic, short-lived microtubule structures, such as the mitotic spindle. Indeed, interpolar microtubules at the mitotic spindle are enriched in polyglutamylated tubulin (Wehenkel and Janke, 2014; Lacroix et al, 2010) (Fig. 5E). Since Elongator is recruited to the mitotic spindle (Planelles-Herrero et al, 2022) this allows Elongator and other factors acting in a similar way to actively remodel the PTM landscape of the newly synthesized microtubules (Fig. 5E–G). In addition, we previously reported that Elongator induces cytoskeleton symmetry breaking during asymmetric cell division, suggesting that Elongator activity can be modulated by cortical polarity cues (Planelles-Herrero et al, 2022). It is thus tempting to speculate that Elongator could imprint polarity onto the landscape of PTMs of the microtubule network downstream of polarity cues. This could explain the markedly nonhomogeneous landscape of microtubule PTMs in polarized cells such as neurons (Tas et al, 2017; Janke and Magiera, 2020).

# Methods

### Reagents and tools table

| Reagent/resource | Reference or source | Identifier or catalog number |
| --- | --- | --- |
| **Experimental models** | | |
| Sf9 cells | ThermoFisher | 11496015 |
| *Drosophila* S2 cells | Gibco | R69007 |
| DH10EmBacY cells | Geneva Biotech | N/A |
| BL21 (DE3) Rosetta2 | Invitrogen | 71397 |
| Flp-In NiH/3T3 cells | Invitrogen | R76107 |
| Mouse WT line | Genova et al, 2023 | N/A |
| Mouse *Ttll1⁻/⁻/Ttll7⁻/⁻* line | Genova et al, 2023 | N/A |

| Reagent/resource | Reference or source | Identifier or catalog number |
| --- | --- | --- |
| Mouse *Atat1⁻/⁻* WT line | Genova et al, 2023 | N/A |
| HeLa S3 cells | ATCC | CCL-2.2 |
| **Recombinant DNA** | | |
| pBig1a PC-SNAP-Elp123 | This study | N/A |
| pMT His-PC-SNAP-Elp3 | Planelles-Herrero et al, 2022 | N/A |
| pGEX Elp456 (*fly*) | Planelles-Herrero et al, 2022 | N/A |
| pGEX Elp456 (*human*) | This study | N/A |
| pGEX eGFP-Elp456 (*fly*) | This study | N/A |
| pGEX eGFP-Elp456 (*human*) | This study | N/A |
| pGEX Elp456 ΔpolyN | This study | N/A |
| pGEX Elp456 solo | This study | N/A |
| pMT His-PC-αTubulin 84B | Wagstaff et al, 2023 | N/A |
| **Antibodies** | | |
| ATTO-647N-anti-K40 acetylated α-tubulin | Derivery et al, 2015 | N/A |
| anti-α-tubulin | Developmental Studies Hybridoma Bank | Clone 12G10 |
| anti-β-tubulin | Developmental Studies Hybridoma Bank | Clone E7 |
| Anti-polyGlu tubulin | AdipoGen Life Sciences | AG-20B-0020-C100 |
| Anti-Elp3 (*fly*) | Planelles-Herrero et al, 2022 | N/A |
| Anti-Elp3 (*mouse*) | Abcam | AB190907 |
| Anti-Elp4 (*fly*) | Planelles-Herrero et al, 2022 | N/A |
| Anti-Elp5 (*fly*) | Planelles-Herrero et al, 2022 | N/A |
| **Oligonucleotides and other sequence-based reagents** | | |
| dsRNA Elp2 | VDRC | 105393 |
| dsRNA Elp3 | VDRC | NIG-15433R-3 |
| dsRNA Elp4 | VDRC | 22460 |
| dsRNA GFP | Derivery et al, 2015 | N/A |
| Elp3 ON-TARGETplus siRNA smartpool | Dharmacon | L-045781-01-0005 |
| **Chemicals, enzymes and other reagents** | | |
| QIAprep Spin Miniprep Kit | Qiagen | 27106 |
| Enhanced Gibson Assembly | Rabe and Cepko, 2020 | N/A |
| Sf-900-II SFM | ThermoFisher | 10902088 |
| FuGENE® HD | Promega | E2311 |
| Insect-Xpress | Lonza | 181562 |
| Penicillin-Streptomycin | Gibco | 15140122 |
| Amphotericin B | Gibco | 15290018 |
| ExpiFectamine™ Sf | Gibco | A38915 |
| Puromycin | ThermoFisher | A1113803 |
| DMEM-Glutamax | Gibco | 31966047 |
| Donor Bovine Serum | Gibco | 16030074 |
| Protein C affinity resin | Planelles-Herrero et al, 2022 | N/A |

| Reagent/resource | Reference or source | Identifier or catalog number |
|---|---|---|
| HiTrap Heparin HP | Cytiva | 17040601 |
| Superose 6 10/300 column | Cytiva | 29091596 |
| Amicon Ultra-4 30kD MWCO | Millipore | UFC9030 |
| SNAP-Surface AlexaFluor 488 | NEB | S9129S |
| cOmplete™ Protease Inhibitor Cocktail Tablets | Roche | 11697498001 |
| Glutathione Sepharose 4B resin | Cytiva | 17075601 |
| Porcine tubulin | Cytoskeleton | T240 |
| AMCA-Porcine tubulin | Cytoskeleton | TL440M |
| Rhodamine-Porcine tubulin | Cytoskeleton | TL590M |
| HiLyte647-Porcine tubulin | Cytoskeleton | TL670M |
| GMPPCP | Jena Bioscience | NU-405S |
| Carboxypeptidase A | Sigma | C9268 |
| Subtilisin | Sigma | P5380 |
| mPEG-Silane 30 kDa | Creative PEGWorks | PSB-2014 |
| PEG-Silane-Biotin 3.4 kDa | Laysan Bio | Biotin-PEG-SIL-3400–500 mg |
| DSBU | Bruker | 1881355 |
| Sulfo-SDA | ThermoFisher | 26173 |
| Pierce™ Premium Grade EDC | Fisher Scientific | 15235763 |
| HyperSep SpinTip P-20 | | |
| **Software** | | |
| FiJi | https://imagej.net/software/fiji/ | |
| MetaMorph v7.10.1.161 | https://www.moleculardevices.com/ | |
| LUMICKS BlueLake | https://lumicks.com/ | |
| Matlab 2020b | https://www.mathworks.com/products/matlab.html | |
| PyMOL v.2.3.3 | https://pymol.org | |
| MSConvert | https://proteowizard.sourceforge.io/ | |
| MeroX | https://www.stavrox.com/ | |
| GraphPad Prism 10.2.3 | https://www.graphpad.com | |
| Unicorn 5.31 | https://www.cytivalifesciences.com/ | |

## Plasmids

For baculoviral expression of Elp123, genes encoding for Elp1 (CG10535), Elp2 (CG11887) and Elp3 (CG15433) were codon optimized for *Drosophila melanogaster*, synthetized by Twist Bioscience (San Francisco, CA) and sub-cloned into pACEBac1 vectors either untagged (Elp1 and Elp2) or containing a His$_6$-PC-SNAP N-terminal tag (Elp3). The final pBig1A vector containing the three subunits (referred here as Elp123) was assembled using an enhanced Gibson Assembly mix (Rabe and Cepko, 2020; Weissmann et al, 2016). Empty pBig plasmids were gifts from Dr Andrew Carter, MRC-LMB, Cambridge, UK. The Elp123 pBig1A vector was then transformed into DH10EmBacY cells (Geneva Biotech) and plated onto agar plates containing 50 μg/ml kanamycin, 10 μg/ml tetracycline, 7 μg/ml gentamycin, 40 μg/ml Isopropyl β-d-1-thiogalactopyranoside (IPTG) and 100 μg/ml Blue-Gal. White colonies, in which the vector has been integrated with the baculovirus genome, were grown overnight in 4 ml of LB supplemented with 50 μg/ml kanamycin, 10 μg/ml tetracycline, 7 μg/ml gentamycin. Bacmid DNA was prepared using QIAprep Spin Miniprep Kit (27106, Qiagen) buffers according to the MultiBac protocol.

The plasmid used for Elongator purification in D.mel-2 cells was previously described in (Planelles-Herrero et al, 2022).

For co-expression of Elp4 (CG6907), Elp5 (CG2034) and Elp6 (CG9829) in bacteria, a modified pGEX vector was used as previously described (Planelles-Herrero et al, 2022). Briefly, Elp4 was first cloned into a modified pGEX vector containing an N-terminal GST-tag followed by a TEV protease cleavage sequence. Then a synthetic fragment comprising both Elp5 and Elp6 ORFs, each flanked by a Ribosome Binding Sequences (RBS) and a stop codon, was cloned in 3' of Elp4. This generates a single transcription unit (Promoter-GST-TEV-Elp4-STOP-RBS-Elp5-STOP-RBS-Elp6-STOP-Terminator) expressing all three subunits. This vector is referred to as pGEX-Elp456 in this study. This strategy was adapted to express all variants of Elp456 used in this study, including mScarlet-Elp456 (Planelles-Herrero et al, 2022), eGFP-Elp456, as well as Elp456 *solo* mutant. The human and *solo* versions of Elp4 (K364E, R397E and E410R), Elp5 (K151E, K179E and T181A) and Elp6 (K199E, S200W and K228E) were synthetized and cloned using the same strategy as for the wild-type.

For recombinant *Drosophila* tubulin expression and purification, a codon-optimised gene coding for tubulin α1 84B (GenBank entry NM_057424) carrying a tandem N-terminal His$_6$-tag and a Protein C epitope tag (PC tag, EDQVDPRLIDGKG) was custom-synthesised (Twist Bioscience) and cloned into a pMT Puro vector (Planelles-Herrero et al, 2022).

## Insect cell transfection

For baculoviral expression of Elp123, Sf9 cells were seeded at $5 \times 10^5$ cells/ml in a six-well plate in a total volume of 2 ml of Sf-900-II SFM media (10902088, ThermoFisher Scientific). Bacmids (see above) were transfected using FuGENE® HD using manufacturer's protocol (E2311, Promega). After 9 days, the supernatant was recovered and used to infect a 50 ml culture of Sf9 cells at $2 \times 10^6$ cells/ml. After 72 h, the virus was harvested by pelleting the cells at 300×*g* for 10 min. This virus was stored at 4 °C in the dark and used at a 1:100 dilution for large-scale cell infection for protein production.

For holo-Elongator production in insect cells, we followed the protocol we previously established (Planelles-Herrero et al, 2022). Briefly, D.mel-2 cells (CRL-1963, ATCC) were grown at 25 °C in Insect-Xpress medium (181562, Lonza) supplemented with 1% Pen/Strep (15140122, Gibco) and 0.25 μg/ml Amphotericin B (15290018, Gibco). Cells were transfected with a pMT Puro His-PC-SNAP-Elp3 vector using ExpiFectamine™ Sf (A38915, Gibco)

using manufacturer's instruction, followed by selection in 5 µg/ml Puromycin (A1113803, ThermoFisher). A similar procedure was used to derive stable cells lines expressing PC-tagged α tubulin using the plasmid described above.

For RNAi-treated S2 cell extracts (Fig. 5E), cells were cultured and incubated with 5 µg of each dsRNA for 4 days as previously described (Goshima and Vale, 2003). We previously validated the dsRNA sequences Elp2, 3 and 4 (Planelles-Herrero et al, 2022), which correspond to the sequences found in the VDRC #105393 (Elp2), NIG-15433R-3 (Elp3) and VDRC #22460 (Elp4) fly stocks. As a control, we used dsRNA against GFP.

## Mouse fibroblasts cells culture and transfection

Flp-In NiH/3T3 cells (Invitrogen) were cultured in DMEM-Glutamax (Gibco) supplemented with 10% Donor Bovine Serum (Gibco) and Pen/Strep 100 units/ml (Gibco) at 37 °C with 5% $CO_2$. Elp3 Gene silencing using ON-TARGETplus siRNA smartpool (Dharmacon, ref L-045781-01-0005) was achieved by reverse transfection using Lipofectamine RNAi max according to the manufacturer's instructions in six-well dishes (day 1). The following day (day 2) the cells were transferred into a 10 cm-dish, and another 24 h later, a reverse forward transfection was performed (day 3). On day 4, cells were washed with PBS and fresh medium was added. Finally, on day 5, the cells were used for respective experiments.

## Mouse lines

Animal care and use for this study were performed in accordance with the recommendations of the European Community (2010/63/UE) for the care and use of laboratory animals. Experimental procedures were specifically approved by the ethics committee of the Institut Curie CEEA-IC #118 (authorisation no. 04395.03 given by National Authority) in compliance with the international guidelines. Adult males and females (2–8 months) were used in this study. All mouse lines used in this study (wild-type, $Ttll1^{-/-}/Ttll7^{-/-}$ and $Atat1^{-/-}$) were described before (Kalebic et al, 2013; Genova et al, 2023).

## Protein purification

For purification of Elp123, Sf9 cells co-expressing Elp1, Elp2 and His-PC-SNAP-Elp3 were pelleted, resuspended in lysis buffer (0.05 M K-HEPES, 0.1 M K-Acetate, 0.002 M $MgCl_2$, 0.01 M $CaCl_2$, 5% glycerol, 120 µg/ml Benzamidine, 20 µg/ml Chymostatin, 20 µg/ml Antipain, 0.5 µg/ml Leupeptin, 240 µg/ml Pefabloc and 1 mM PMSF, pH 8.0). Cells were lysed by extrusion using a Teflon Dounce homogenizer and clarified by centrifugation at 30,000×g for 40 min at 4 °C using a Ja 25.50 rotor (Beckman). The clarified lysate was incubated with 2 ml of pre-equilibrated Protein C affinity resin (Roche) for 2 h at 4 °C. Then, the resin was packed in an empty column (Bio-rad), washed with 20 ml of wash buffer (0.05 M K-HEPES, 0.1 M K-Acetate, 0.002 M $MgCl_2$, pH 8.0) supplemented with 0.001 M $CaCl_2$, followed by 50 ml of wash buffer without $CaCl_2$. The Elp123 sub-complex was then eluted with elution buffer (wash buffer supplemented with 0.01 M EGTA pH 8.0) and analysed by SDS-PAGE. The Elp123 containing fractions were further purified using a Heparin column (HiTrap Heparin HP,

Cytiva) equilibrated in Elution buffer and eluted using a linear gradient between Heparin A (0.05 M K-HEPES, 0.1 M K-Acetate, 0.002 M $MgCl_2$, 0.01 M EGTA, pH 8.0) and Heparin B (0.05 M K-HEPES, 1 M K-Acetate, 0.002 M $MgCl_2$, 0.01 M EGTA, pH 8.0) buffers. The elution was analysed by SDS-PAGE, and the fractions of interest were pooled and injected in Superose 6 10/300 column (Cytiva) equilibrated and eluted in Elongator buffer (0.02 M K-HEPES, 0.15 M K-Acetate, 0.001 M DTT, pH 8.0). The peak fractions were collected, concentrated (Amicon Ultra-4 30kD MWCO, Millipore) and labelled using SNAP-Surface AlexaFluor 488 (NEB) using manufacturer's instructions. The labelled protein was then desalted into Elongator buffer to remove non-bound dye, concentrated, flash-frozen in liquid $N_2$ and stored in small aliquots at −80 °C.

Holo-Elongator was purified from D.mel-2 cells transfected with His-PC-SNAP-Elp3 using a previously established protocol (Planelles-Herrero et al, 2022). Briefly, the transfected D.mel-2 cells were grown to litre-scale over several weeks, then the expression was induced by the addition of 0.6 mM $CuSO_4$. After 4 days, the cells were pelleted, lysed, and purified using a PC-resin as described for Elp123 except that all buffers contain 20% glycerol. After PC-resin elution, Elongator-containing fractions were concentrated and further purified by sucrose gradient by layering the protein on top of a manually prepared 10–30% discontinuous sucrose gradient. After ultracentrifugation for 1 h at 258,488×g at 4 °C using a TLS-55 rotor (Beckman), the fractions were analysed by SDS-PAGE and the positive fractions were concentrated, labelled with SNAP-Surface Alexa Fluor 488, and desalted using the same procedure described for Elp123.

All variants of the Elp456 sub-complex (wild type *Drosophila* and human, mScarlet tagged, eGFP tagged and *solo* mutant) were purified using the same protocol. BL21 (DE3) Rosetta2 (Invitrogen) *Escherichia coli* were transformed with the respective vector and grown at 37 °C in 2xYT medium to $OD_{600}$ = 0.8. Protein expression was induced by the addition of 0.5 mM IPTG at 20 °C for at least 16 h. The cells were collected by centrifugation, resuspended in lysis buffer (0.05 M K-HEPES pH 7.5, 0.1 M K-Acetate, 5% glycerol, 0.001 M DTT, 0.01 M $MgCl_2$, 1% Triton X-100 and cOmplete™ Protease Inhibitor Cocktail Tablets (Roche) and lysed by sonication. The lysate was then clarified (using a JA 25.50 Beckman rotor at 40,000×g for 30 min, 4 °C) and the soluble fraction was incubated with 2 ml of Glutathione Sepharose 4B resin for 2 h at 4 °C under constant agitation. The resin was washed with 4 times 20 ml of wash buffer (0.05 M K-HEPES pH 7.5, 0.1 M K-Acetate, 5% glycerol, 0.001 M DTT, 0.01 M $MgCl_2$) and the protein was eluted by incubating the resin with rTEV protease (overnight, 4 °C) to remove the N-terminal GST-tag in Elp4 (releasing the protein from the resin). The next morning, the eluted protein was recovered and further purified by size-exclusion chromatography on a Superdex 200 16/60 column (Cytiva) in Elongator buffer. Finally, the peak fractions containing the three proteins (Elp4, Elp5 and Elp6 at seemingly equimolar quantities) were concentrated, flash-frozen in liquid $N_2$ and stored in small aliquots at −80 °C.

Unlabelled porcine tubulin, AMCA-, rhodamine- and HiLyte647-labelled porcine tubulin were purchased from Cytoskeleton (T240, TL440M, TL590M and TL670M, respectively). All tubulins were reconstituted at 5 mg/ml in BRB80 buffer (80 mM K-Pipes, 1 mM $MgCl_2$, 0.5 mM EGTA, pH 6.9) supplemented with either 1 mM GTP (Roche), 1 mM GMPPCP (NU-405S, Jena

Bioscience) or 1 mM BeF$_4$ (prepared by mixing 50 mM BeSO$_4$ and NaF 100 mM to make 10 mM BeF$_3$), flash frozen and kept in liquid N$_2$. Microtubules seeds stocks were prepared at 5 mg/ml tubulin concentration (20% fluorescent-tubulin, 20% biotinylated tubulin) in BRB80 buffer supplemented with either 1 mM GMPCPP or 1 mM BeF$_4$, aliquoted in 1 µl aliquots and stored in liquid N$_2$. To polymerize the seeds, one aliquot of microtubule seed stock is incubated at 37 °C for 30 min, sedimented on a table-top centrifuge by centrifugation (8 min at 14,000× $g$), and resuspended in BRB80 buffer supplemented with 1 mM nucleotide.

For recombinant *Drosophila* tubulin purification, D.mel-2 cells were transfected and grown using the same strategy as for His-PC-SNAP-Elp3. To purify the αβ-tubulin heterodimers, the cells were lysed in lysis buffer (80 mM K-PIPES pH 6.9, 10 mM CaCl$_2$, 10 µM Na-GTP, 10 µM MgCl$_2$, 0.12 mg/mL benzamidine, 20 µg/mL chymostatin, 20 µg/mL antipain, 0.5 µg/mL leupeptin, 0.24 mM Pefabloc SC, 0.5 mM PMSF) and lysed by using a Dounce homogeniser. The lysate was rocked for 1 h at 4 °C to ensure complete microtubule depolymerisation, then clarified by centrifugation at 66,000× $g$ for 30 min using a JA 25.50 rotor (Beckman). The supernatant was incubated with 2 mL of pre-equilibrated Protein C affinity resin (Roche) for 3 h at 4 °C. After incubation, the resin was packed into an empty column (Bio-Rad), washed with 50 mL of tubulin wash buffer (80 mM K-PIPES pH 6.9, 10 µM Na-GTP, 10 µM MgCl$_2$, 1 mM CaCl$_2$), 50 mL of tubulin ATP-buffer (wash buffer + 10 mM MgCl$_2$ and 10 mM Na-ATP), 50 mL of low-salt buffer (wash buffer + 50 mM KCl), 50 ml of high-salt buffer (wash buffer + 300 mM KCl), 50 mL of Tween buffer (wash buffer + 0.1% Tween-20 and 10% glycerol) and finally 50 mL of tubulin wash buffer without CaCl$_2$. Tubulin was eluted with tubulin buffer (80 mM K-PIPES, 10 µM Na-GTP, 10 µM MgCl$_2$, 5 mM EGTA, pH 6.9). The protein-containing fractions were pooled and further purified on a Superdex 200 column (GE Healthcare), equilibrated, and eluted in tubulin buffer. The peak fractions were pooled and concentrated to 10 mg/ml, flash-frozen in liquid N$_2$ and stored in liquid N$_2$. The resulting αβ-tubulin heterodimers are formed by isotype-pure tubulin α1-84B (Uniprot TBA1_DROME) and the co-purified β tubulin (tubulin β1-56D, Uniprot TBB1_DROME (Wagstaff et al, 2023). We previously characterised this tubulin by CryoEM which shows that purified recombinant tubulin purified this way is functional and assembles in 13-protofilament microtubules in the presence of GTP (Wagstaff et al, 2023).

Monovalent streptavidin (mSA) was purified from BL21 (DE3) Rosetta2 *Escherichia coli* (Invitrogen) following a similar protocol as described for Elp456 above and adapting the buffers as follows: lysis buffer (20 mM HEPES, 150 mM KCl, 5% glycerol, 1% Triton, 1 mg/ml lysozyme, 4 µg/ml DNaseI, 5 mM MgCl$_2$, 15 mM imidazole, 1 tablet cOmplete™ EDTA-free Protease Inhibitor Cocktail, pH 7.5); wash buffer (20 mM HEPES, 150 mM KCl, 5% glycerol, 15 mM imidazole, pH 7.5); elution buffer (20 mM HEPES, 150 mM KCl, 5% glycerol, 250 mM imidazole, pH 7.5). Ni-NTA resin (Qiagen) was used instead of Glutathione Sepharose 4B resin).

## Tubulin purification from murine brains

Tubulin was purified from mouse brains (see Mouse lines) via cycles of temperature-dependent microtubule polymerisation and depolymerisation as previously described (Souphron et al, 2019;

Bodakuntla et al, 2020). Briefly, the animals were sacrificed and the brains extracted immediately and added in ice-cold lysis buffer (BRB80 (80 mM K-PIPES pH 6.8, 1 mM K-EGTA, 1 mM MgCl$_2$), 1 mM β-mercaptoethanol, 1 mM PMSF and 1× protease inhibitor cocktail: 20 µg/ml leupeptin, 20 µg/ml aprotinin and 20 µg/ml 4-(2aminoethyl)-benzenesulfonyl fluoride; Sigma-Aldrich) in a ratio of 2 ml buffer per 1 g of tissue. The brains were homogenised using an Ultra-Turrax® blender and the lysates cleared by centrifugation at 112,000×$g$ in TLA-55 fixed-angle rotor (Beckman Coulter) for 30 min at 4 °C (i.e. cold centrifugation). To this first supernatant (SN1) final concentration of 1 mM GTP and 1/3 volume pre-warmed 100% glycerol were added, mixed and incubated for 20 min at 30 °C to allow microtubule polymerisation. Polymerised microtubules were pelleted for 30 min at 112,000×$g$, 30 °C, (i.e. warm centrifugation) and the supernatant SN2 discarded. The microtubule-containing pellet was resuspended in cold BRB80 (depolymerisation 1; 1/10 of the initial SN1-volume) and incubated on ice with occasional pipetting up-and-down over 20 min to mechanically assist microtubule depolymerization. Solubilised tubulin was cleared via cold centrifugation, the supernatant SN3 was transferred to a fresh tube and adjusted to final concentrations of 1 mM GTP and 0.5 M PIPES, complemented with 1/3 volume pre-heated 100% glycerol. The microtubule polymerization step was repeated by incubation of the solution for 20 min at 30 °C and the resulting microtubules were pelleted via warm centrifugation. Note that the microtubules yielded in this step by polymerisation in presence of the high molarity buffer are largely free of associated proteins. The supernatant SN4 containing these associated proteins was discarded and the pellet was resuspended in cold BRB80 (depolymerisation 2; 1/40 of the initial SN1-volume), depolymerised on ice for 20 min as before, and cleared via cold centrifugation. The resulting SN5 was adjusted to 1 mM GTP, supplemented with 1/3 volume of pre-heated 100% glycerol, and incubated for 20 min at 30 °C, after which the microtubules were pelleted by warm centrifugation and resuspended in ice-cold BRB80 (depolymerisation 3; 1/ 40 of the initial SN1-volume). After 15 min on ice, the soluble tubulin was cleared by a final cold centrifugation and the tubulin yield was estimated with a NanoDrop ND-1000 spectrophotometer (Thermo Scientific; absorbance at 280 nm; MW = 110 kDa; ε = 115,000 M − 1/cm$^{-1}$). Samples were aliquoted, snap-frozen in liquid N2 and stored at −80 °C.

The characterization of these tubulin from knockout mice, in particular their content in tubulin polyglutamylation and acetylation was previously published (Genova et al, 2023).

## Purification of tyrosinated and de-tyrosinated tubulin from HeLa cells

Tubulin was purified from HeLa S3 cells using cycles of polymerisation and depolymerisation similar to the approach used for brain tubulin (Souphron et al, 2019). Briefly, HeLa S3 cells were grown in suspension in 1-L spinner bottles for 7 days, collected, and the cell pellet (approx. 10 ml for 4 spinner bottles) was lysed in the same volume of ice-cold lysis buffer (same composition as for the brain tubulin prep, see above) using a French press. The subsequent steps were the same as described above for brain tubulin, except that during the depolymerisation step 1, the microtubule pellet was resuspended in 1/60 of the SN1 volume, for depolymerisation 2 in 1/100 of the SN1 volume, and for

depolymerisation 3 in 1/300 of SN1 volume. Tubulin purified from HeLa cells is highly tyrosinated (Barisic et al, 2015). To obtain detyrosinated tubulin, the solubilized tubulin in SN3 was incubated with 1/300 volume of Carboxypeptidase A (Sigma C9268) at 30 °C for 5 min before the incubation in high molarity PIPES buffer, and the rest of the purification procedure was followed as described above.

## SDS-PAGE and western blot

SDS-PAGE was performed using NuPAGE 4–12% Bis-Tris gels (Life Technologies) according to the manufacturer's instructions. Instant Blue (Sigma) was used for total protein staining of gels.

For western blot, gels were transferred on nitrocellulose membranes using iBLOT (Life Technologies). The membranes were first stained with Ponceau (Sigma) to assess the quality of the transfer, then washed in TBS and blocked in TBS supplemented with 5% semi-skimmed milk for 30 min at RT. For fluorescent western blot with Atto-647N-labelled anti-α K40 acetylated tubulin antibodies(Derivery et al, 2015), membranes were incubated overnight at 4 °C with 1 µg/ml antibody in TBS supplemented with 0.2% BSA. The membranes were then imaged using a Typhoon scanner.

## Microtubule co-sedimentation assay

All proteins were pre-clarified at 20,000×g at 4 °C for 10 min before use in biochemical assays. Reconstituted porcine tubulin or HeLa tubulin was diluted to 50 µM in BRB80 buffer supplemented with 1 mM GTP. To initiate microtubule polymerization, polymerization buffer (BRB80 supplemented with 60% glycerol) was added to a final concentration of 5% glycerol. After 20 min incubation at 37 °C in a water bath, the microtubule samples were diluted to 10 µM in BRB80 buffer supplemented with 40 µM Taxol.

To prepare subtilisin-digested porcine tubulin, freshly prepared Taxol-stabilized microtubules were incubated with 0.05 mg/ml subtilisin (P5380, Sigma) for 40 min at 37 °C in a water bath. The reaction was stopped by the addition of 2 mM PMSF (93482, Sigma).

At this point, all microtubule samples were cleared by spinning for 15 min at 20,000×g at Room Temperature to remove non-polymerized tubulin. The supernatant was then discarded and the microtubules resuspended in BRB80 buffer supplemented with 40 µM Taxol.

To test the binding of Elp123 to microtubules, 50 µL solutions were prepared with 5 µM microtubules and 160 nM Elp123, and the samples were incubated at Room Temperature for 30 min. After incubation, the samples were transferred into 20 mm polycarbonate ultracentrifuge tubes (Beckman) pre-filled with 150 µl of cushion solution (BRB80 supplemented with 60% glycerol and 40 µM taxol). The samples were centrifuged at 100,000×g for 30 min at 25 °C using a TLA100 rotor. After centrifugation, the top 20 µl were recovered and saved for SDS-PAGE/Western blot analysis, designated as "supernatant". The remaining 10 µl atop the glycerol cushion were removed and discarded, and the interfaces were washed three times with 100 µl of BRB80 buffer. The glycerol cushion solutions were then removed and the pellets gently washed with 3×100 µl of BRB80 buffer. Finally, the pellets were resuspended in 50 µl of BRB80 buffer, saved and designated as "pellet" for further analysis.

## Microtubule dynamics assays

All proteins were pre-clarified at 20,000×g at 4 °C for 10 min before use in microscopy assays. Glass 22 × 22 mm coverslips (NEXTERION, Schott) were incubated at least for 48 h at room temperature with gentle agitation in a 1:10 (w/w) mix of mPEG-Silane (30 kDa, PSB-2014, Creative PEGWorks) and PEG-Silane-Biotin (3.4 kDa, Laysan Bio) at a final concentration of 1 mg/ml in 96% (v/v) ethanol and 0.2% (v/v) HCl. The day of the experiment, the coverslips were washed with ethanol and ultrapure water, dried with a nitrogen gas gun, and assembled into an array of flow cells on mPEG-Silane passivated slides using double-sided tape (Adhesive Research AR-90880 precisely cut with a Graphtec CE6000 cutting plotter). The chamber was first perfused with BRB80 buffer, and then washed with 5% Pluronic-F127 and BRB80. Neutravidin (25 µg/ml) was then added to the chamber and incubated for 5 min, then washed out with 10 chamber volumes of BRB80. Biotin-stabilized microtubule seeds (20% with rhodamine-tubulin, 20% biotin-tubulin) were injected in imaging buffer (BRB80 enriched with 0.1 mg/ml K-Casein, 0.1 mg/ml BSA, 40 µM DTT, 64 mM D-glucose, 160 µg/ml glucose oxidase, 20 µg/ml catalase and 0.2% methylcellulose) and let for 5 min to bind to the neutravidin, followed by a final wash with imaging buffer. Dynamic microtubules were elongated from the seeds by injecting in the chamber a solution of 10% labelled HiLyte647-GTP Tubulin and different Elongator sub-complexes (or an equivalent volume of Elongator buffer as a control) and microtubule dynamics were monitored by TIRFM. Note that an aliquot of the last buffer exchange step during the purification of each protein was always kept to serve as a true negative control in all tubulin dynamics experiments. The microscope stage was kept at 37 °C with an objective heater in addition to the heating chamber containing the microscope.

## Microtubule-decoration experiments

To further increase the signal-to-noise ratio in our TIRFM images for microtubule-decoration experiments (Figs. 1E, 2A, 3A, 4A and EV2) we modified the passivation of the glass coverslips as follows: glass coverslips were initially incubated with a 1 mg/ml solution of mPEG-Silane. After assembly, the chamber was first perfused with BRB80, then incubated with a solution of 0.1 mg/ml PLL-g-PEG (Susos) to further passivate the surface. After three minutes, the chamber was washed with 10 volumes of BRB80 then passivated with 1 mg/ml K-Casein. After three more minutes, GMPCPP-stabilized microtubule seeds diluted in imaging buffer were introduced in the chamber and left for 5 min to settle. Then, the chamber was washed with imaging buffer and the proteins of interested were sequentially added to the chamber. Concentrations used are: Fig. 2A, 100 nM 488-SNAP-Elp123; Fig. 3A, 100 nM 488-SNAP-Elongator, 15 µM 647-GTP-Tubulin; Figs. 4A and EV2, 100 nM 488-SNAP-Elp123, 500 nM mScarlet-Elp456, 15 µM 647-GTP-Tubulin.

## Microtubule polyglutamylation detection experiments

All proteins were centrifuged for 10 min at 20,000×g at 4 °C before use in biochemical assays. Glass coverslips and experimental chambers were prepared following the same protocol as for

microtubule dynamics experiments. After addition of biotin-stabilized microtubule seeds (20% with HiLyte488-tubulin, 20% biotin-tubulin), the chamber was washed with imaging buffer and then 15 µM of tubulin (either 50:50 pig brain tubulin:HeLa S3 tubulin, or 100% HeLa S3 tubulin) together with either 50 nM Elongator complex or Elongator buffer were injected in the chamber and incubated at 37 °C. After 10 min, a mix containing 50 µM taxol and 30 nM anti poly glutamylated alpha tubulin (GT335, AdipoGen Life Sciences) labelled with ATTO565 in imaging buffer was injected, and after 10 min, the chamber was finally washed with imaging buffer containing 50 µM taxol. The microscope stage was kept at 37 °C.

To quantify the polyglutamylation state of polymerized microtubules, we measured the intensity of hundreds of microtubules per condition (from two or three independent experiments) in both the ATTO565-fluorescence (polyglutamylated tubulin) and HiLyte488 (microtubule seeds) fluorescence channels. For this, we traced a 2 pixel-wide line along the length of each microtubule and measured the underlying signal using the linescan function in FiJi (Schindelin et al, 2012). For each microtubule, the 565-fluorescence intensity was averaged along the length of the microtubule and the background signal value was subtracted (background was measured at the edges of the linescan devoid of microtubule). Then, the calculated signal of all microtubules in a field of view (FOV) was averaged, and the result was normalized against the average 488-fluorescence intensity from the microtubule seeds of that FOV, generating an averaged value per FOV. For each condition, we then averaged this mean normalized 565-fluorescence microtubule intensity across multiple FOVs ($n = 6$–9) and compared them (non-transparent dots Fig. 5D, individual values for each microtubule are also displayed as transparent dots). Note that analysing this dataset by a two-way ANOVA confirmed that the vast majority of the variation comes from comparing different conditions (86.25%), and not from different FOVs from the same condition (0.21%), whose differences are not statistically significant (i.e. FOV1 from Control vs FOV2 from Control, p_value = 0.9945). In addition, note also that the HiLyte488 fluorescence signal from the seeds is very homogeneous between FOVs and samples, high-lighting the homogeneity of the samples, microscope conditions and TIRF field (Control: $178.62 \pm 11.19$ a.u. from 354 seeds, Elongator: $174.42 \pm 13.49$ a.u. from 375 seeds, HeLa S3 control: $175.31 \pm 17.91$ a.u. from 281 seeds, and HeLa S3 Elongator: $179.84 \pm 10.86$ a.u. from 288 seeds). HiLyte488 fluorescence differences are not significant between FOVs of the same conditions, and between all conditions (ordinary one-way ANOVA followed by a Turkey test for multiple comparisons). Number of analysed microtubules, FOV and independent experiments are: Control ($N = 3$ independent experiments, $n = 9$ FOV, $n = 618$ microtubules), Elongator ($N = 3$ independent experiments, $n = 9$ FOV, $n = 627$ microtubules), HeLa S3 Control ($N = 2$ independent experiments, $n = 6$ FOV, $n = 638$ microtubules) and HeLa S3 Elongator ($N = 2$ independent experiments, $n = 6$ FOV, $n = 619$ microtubules).

To quantify the total polyGlutamylated tubulin signal in a field of view (Fig. EV5E–H), the total signal from the antibody channel was measured in FiJi (Schindelin et al, 2012) and the background was subtracted (background signal was extrapolated from the mean average per pixel of the signal from a region of interested devoid of signal multiplied by the total amount of pixels in the field of view).

The resulting number was divided by the amount of microtubule seeds observed.

## Tubulin binding experiments

The binding of Elongator subcomplexes to tubulin was first assessed using gel filtration experiments. Pig brain tubulin was diluted to 50 µM in ice-cold BRB80 and kept on ice for 30 min to favour microtubule depolymerization. Then, Elp456 was added to create a final solution of 30 µM Elp456 and 35 µM tubulin. The proteins were incubated on ice for 30 min, then injected in a superpose 6 increase 3.2/300 column equilibrated and eluted in binding buffer (40 mM K-Pipes, 1 mM MgCl$_2$, 1 mM EGTA, 20 µM GTP, 20 µM MgCl$_2$, pH 6.9) (Fig. 3B; Appendix Fig. S3F,G). The elution profile was analysed by SDS-PAGE and western blot (Fig. 3C; Appendix Fig. S3F,G).

The binding of Elp456 and Elp123 to tubulin was also analysed by protein immunoprecipitation (Appendix Fig. S3). For Elp123, 30 µM pig brain tubulin was incubated alone or in the presence of 1 µM PC-Elp123 in binding buffer supplemented with 1 mM CaCl$_2$ (required for PC-binding to the epitope) for 30 min on ice. Then, the solution was diluted tenfold in binding buffer and 20 µL of PC-resin (pre-equilibrated in binding buffer) was added to the mixture. The proteins were left to interact with each other and bind to the resin for 2 h in constant rotation at 4 °C. Then, the sample was centrifuged for 10 min at 20,000×$g$ at 4 °C, the resin was washed thrice with binding buffer (CaCl$_2$ was removed in the last wash step to facilitate elution) and finally the PC-bound proteins were released by incubating the resin with binding buffer supplemented with 5 mM EGTA for 30 min (twice). The results were analysed by SDS-PAGE, 488-fluorescence (using a typhoon scanner) and western-blot anti αTubulin. To assess the binding of Elp456 to tubulin, the same protocol was followed except that non-tagged Elp456 and recombinant *Drosophila* PC-Tubulin (both at 30 µM) were used.

## Concentrated cytoplasmic extracts of *Drosophila* S2 cells

To generate highly concentrated cytoplasmic extracts of S2 cells, wild-type or stably expressing *Drosophila* S2 cells were grown in 25 cm dishes. SNAP-Elp3 expression was induced by the addition of 0.6 mM CuSO$_4$ for 3 days, so it is incorporated into the endogenous Elongator complex (note that this is the same method we previously used to purify the semi-endogenous Elongator complex (Planelles-Herrero et al, 2022)). Cells were then treated with SNAP-Cell® Oregon Green® (NEB) following manufacturer's instructions. Briefly, 2 ml of wild-type and SNAP-Elp3 cells at ~80% confluency were seeded in six-well plates and left to adhere to the surface. Then, 1 ml of supernatant was removed and 1 ml of a solution of 10 µM dye in Insect Xpress medium was added to each well. After incubating for 30 min at 25 °C, the cells were resuspended in 10 ml of fresh Insect Xpress medium and pelleted for 5 min at 300×$g$. This was repeated three times to ensure complete removal of dye not incorporated in cells. In the final step, the cells were resuspended in 1 ml and plated again in 6 wells plates and incubated for 30 min at 25 °C. Finally, the cells were resuspended, pelleted and resuspended again in 500 µl of lysis buffer (20 mM HEPES, 150 mM KCl, 40 mM DTT, 1% Triton-X100, pH 7.5). The cells were lysed using a dounce and rocked for 30 min at 4 °C in the dark. Finally, the cells were pelleted at 20,000×$g$ for 10 min at 4 °C, and the supernatant was used for imaging immediately after.

## Immunoprecipitation from *Drosophila* S2 cell extracts

For immunoprecipitation of recombinant Tubulin, S2 cells stably expressing PC-tagged tubulin α1 84B were lysed in lysis buffer (50 mM K-HEPES pH, 50/300 mM KCl, 1% Triton X-100, 1 mM CaCl$_2$, 5 mM MgCl$_2$, 20% glycerol, 0.12 mg/ml Benzamidine, 20 µg/ml chymostatin, 20 µg/ml antipain, 0.5 µg/ml leupeptin, 0.24 mM Pefabloc SC and 0.5 mM PMSF, pH 7.5) then incubated with 20 µl of agarose beads crosslinked with 4 mg/ml of anti-PC antibody (HPC4). After 2 h incubation at 4 °C with rocking, the beads were washed three times in lysis buffer and the samples analysed by SDS-PAGE and Western blot. Antibodies used: *Drosophila* Elp2 and Elp4 (validated in Planelles-Herrero et al, 2022), Oregon Green 514-labelled mouse anti-β-tubulin (E7, Developmental studies hybridoma bank, see (Planelles-Herrero et al, 2022), for characterization of the labelled antibody).

## *Drosophila* stocks

Fly handling was done according to standard procedures. To generate the desired genotypes and remove balancers, all experiments were performed on F1 of crosses at 25 °C. Transgenes used in this study included UAS-mRFP-PonLD (Bellaïche et al, 2001).), Neur>Gal4 (Emery et al, 2005) and UAS>Elp3$^{RNAi}$ (NIG-15433R-3). Note that we previously validated the specificity of this UAS>Elp3$^{RNAi}$ in these cells and conditions using RNAi-resistant rescue constructs (Planelles-Herrero et al, 2022). For immunofluorescence of dividing SOPS, larvae were then shifted to 16 °C until puparium and were shifted to 29 °C 16 h prior to dissection.

Detailed genotypes:
Figure 5F: w$^{1118}$;;Neur>Gal4, UAS>mRFP-PonLD/+ (29 °C)
w$^{1118}$;;Neur>Gal4, UAS>mRFP-PonLD/UAS>Elp3$^{RNAi}$ (29 °C).

## Immunofluorescence

For immunofluorescence of S2 cells, $3.5 \times 10^6$ cells/well were plated on µ-Slide 4 Well Glass Bottom slides (ibidi #80427) pre-treated with 0.05 mg/ml concanavalin A. Cells were left to adhere for 2 h at 25 °C, then washed with PBS and fixed with ice-cold methanol for 3 min at −20 °C. Then, cells were washed with 3× PBS and blocked with PSB-2% bovine serum albumin (BSA, Fisher, BP1605) for 30 min at room temperature. Finally, cells were processed for immunofluorescence using anti-polyglutamylated (GT335, Adipo-Gen, labelled with Alexa 647) and anti-α-tubulin (12G10, abcam, labelled with OregonGreen 514).

For immunofluorescence of 3T3 cells, mitotic cells were isolated using the mitotic shake-off method. For this, cells growing in 10-cm dishes at ~75% confluency were gently washed with PBS, then 250 µl PBS was added to the 10-cm dish, and each side of the dish was tapped 10–15 times. The liquid was then collected and transferred onto Fibronectin-coated µ-Slide 4 Well Glass Bottom (ibidi #80427) slides. The slides were subsequently incubated for 20 min at 37 °C to allow attachment of the mitotic cells. Cells were then fixed with 4% paraformaldehyde (PFA) in PBS for 20 min at room temperature, then washed in PBS and subsequently permeabilized with 0.1% Triton X-100 for 5 min. Cells were then washed in PBS and incubated with 1% bovine serum albumin (BSA, Fisher, BP1605) in PBS for 10 min. Cells were then with anti-polyglutamylated (GT335, AdipoGen, labelled with Alexa 647) and anti-α-tubulin (12G10, abcam, labelled with OregonGreen 514).

For *Drosophila* notum immunofluorescence (Fig. 5G), dissected nota were fixed in PEM buffer (0.1 M PIPES, 1 mM EGTA, pH 6.9) enriched 4% PFA (Electron Microscopy Science) for 20 min before permeabilization in fixation buffer with 0.2% Triton X100. Nota were then processed for immunofluorescence using anti-polyglutamylated (GT335, AdipoGen, labelled with Alexa 647) and anti-α-tubulin (12G10, Abcam, labelled with OregonGreen 514) as described (Planelles-Herrero et al, 2022) and mounted in Prolong Gold antifade reagent (Molecular Probes). Imaging was performed on the Spinning Disk confocal microscope described below.

## Microscale thermophoresis

To measure the Elp456 binding affinity to different tubulins or to Elp123, microscale thermophoresis (MST) assays were performed on a Monolith X (Nanotemper Technologies) using standard capillaries (MO-K022) at 25 °C. In all, 16 or 24 twofold dilutions series of tubulin or Elp123 were performed in binding buffer supplemented with 0.05% (v/v) Tween-20. eGFP-Elp456 (WT or mutated) was used at a constant concentration of 50 nM. For the measurements in high tRNA conditions, eGFP-Elp456 was diluted in a buffer containing additional 100 µM tRNA and incubated on ice for 10 min to ensure complex formation. Similarly, tubulin serial dilutions were performed in a buffer containing 100 µM tRNA. Thermophoresis was measured using automatic LED power and irradiating the capillaries for 20 s at medium laser power (50%). The dissociation constant values were obtained by analysing the change in thermophoresis using the MO.Affinity (v3.0.5) software provided by the manufacturer.

## Microscopy

Most imaging was performed using a custom Spinning disk/Total Internal Reflection Fluorescence (TIRF) microscope composed of a Nikon Ti stand equipped with perfect focus, a fast piezo z-stage (ASI) and a Plan Apochromat lambda 100× NA 1.45 objective. TIRF illumination was achieved with an azimuthal TIRF illuminator (iLas2, Roper France) modified to have an extended field of view to match the size of the camera (Cairn). Images were recorded with a Photometrics Prime 95B back-illuminated sCMOS camera run in pseudo-global shutter mode and synchronized with the rotation of the azimuthal illumination. TIRF angle was set independently for all channels so that the depth of the TIRF field was identical for all channels. Confocal illumination is achieved using a Yokogawa CSU-X1 spinning disk head and a Photometrics 95B back-illuminated sCMOS camera operated in global shutter mode and synchronized with the spinning disk rotation. Excitation was performed using 405 nm (100 mW OBIS LX), 488 nm (150 mW OBIS LX), 561 nm (100 mW OBIS LS) and 637 nm (140 mW OBIS LX) lasers fibered within a Cairn laser launch. To minimize bleedthrough, single band emission filters were used (Chroma 525/50 for Alexa 488/GFP/Alexa514; Chroma 595/50 for mScarlet/rhodamine/mRFP/mCherry and Chroma 655LP for HiLyte647/Alexa647/ATTO647N) and acquisition of each channel was performed sequentially using a fast filter wheel (Cairn Optospin). To enable fast acquisition, the entire setup is synchronized at the hardware level by a Zynq-7020 Field Programmable Gate Array (FPGA) stand-alone card (National Instrument sbrio 9637) running custom code. Sample temperature

was maintained at 25 °C (or 37 °C for microtubule dynamics) using a heating enclosure (MicroscopeHeaters.com, Brighton, UK). Acquisition was controlled by Metamorph software (v7.10.1.161).

Dual IRM (Interference reflection microscopy) and 488-TIRF imaging were performed in a LUMICKS C-Trap Edge setup. The chambers were prepared using the same protocol as for TIRFM. For IRM, 10 frames of 100 ms were collected and averaged, and the background was subtracted using LUMICKS BlueLake software. Sample temperature was maintained at 25 °C.

## Image processing

Images were processed using Fiji (Schindelin et al, 2012) (ImageJ version: 1.53f) and Matlab 2020b (Mathworks). Figures were assembled in Adobe Illustrator 2021. All lookup tables applied to images in this paper come from the collection from James Manton (https://github.com/jdmanton/ImageJ_LUTs).

Spatial drift during acquisition for TIRF movies was corrected using a custom GPU-accelerated registration code based on cross correlation between successive frames. Drift was measured on one channel and applied to all the channels in multichannel acquisitions. Code is available on our github page (https://github.com/deriverylab/GPU_registration).

Microtubule polarity was established by measuring growth rates, with fast-growing extensions indicating plus ends and slow-growing extensions indicating minus ends. Microtubule dynamics were analyzed manually on kymographs using the ImageJ plugin 'Kymo-ToolBox' developed by Fabrice Corderlières (Zala et al, 2013) after homogenous background subtraction. Microtubule growth speeds were determined from the slope of kymographs to account for transient growth irregularities. Microtubule lifetime were computed for each microtubule as the elapsed time between the onset of growth and the last time frame before a catastrophe event. Following the method established by Gardner and colleagues (Gardner et al, 2011), we plotted the cumulative distribution of microtubule lifetimes and fitted it to a gamma distribution in Matlab using the gamfit function. As an estimate of the mean lifetime, we computed the lifetime at the half cumulative distribution from the fitted data (i.e. when the cumulative distribution is equal to 0.5). We then used boostrapping to estimate the error of this mean lifetime value, as we did previously (Planelles-Herrero et al, 2022). Briefly, we drew 10,000 bootstrap data samples from the observed lifetime distribution and computed the lifetime at the half cumulative distribution as above for each of them. We then computed the interquartile range of the distribution of the boostrapped mean lifetimes (i.e. the difference between the 75th and the 25th percentiles of the boostrapped mean lifetime distribution), and used it as an estimate of the error. Note that throughout the paper, we plotted results for microtubule growth rate and lifetime only at the plus end in figure panels for simplicity. This is because since Elongator has similar effects on the dynamics of both the plus and the minus ends (Planelles-Herrero et al, 2022), plus end dynamics measurements are sufficient to evaluate the effects of Elongator. We nevertheless provide minus end dynamics measurements in the text, and in Appendix Fig. S2.

## AlphaFold2 modelling of Drosophila Elp456

The *Drosophila melanogaster* Elp456 model was generated using AlphaFold2 Multimer (Jumper et al, 2021; Evans et al, 2022) as implemented in Colabfold (Mirdita et al, 2022). *Drosophila melanogaster* sequences were provided and identical results were obtained with and without template search in the PDB for the known structures of the Elp456 ring from different species (Glatt et al, 2012; Gaik et al, 2022a, 2022b; Jaciuk et al, 2023), resulting in both cases in high confidence models. In detail, the core of the six subunits as well as the interfaces were predicted with high confidence and minimal variability between models, whereas the differences were more important in the flexible regions, as expected. The model with the highest overall confidence was retained as the model shown in Appendix Figs. S5D and S7.

## Cross-linking coupled to mass spectrometry (XL-MS)

For cross-linking coupled to mass spectrometry of the full Elongator complex, 75 μl of His-PC-Elongator at 4.2 mg/ml was cross-linked with the N-hydroxysuccinimide (NHS) ester disuccinimidyl dibutyric urea (DSBU). Cross-linking reaction was allowed for 45 min at room temperature with a 150× excess of crosslinker. The reaction was quenched by the addition of $NH_4HCO_3$ to a final concentration of 50 mM, and incubating for 15 min. Note that this experiment was repeated twice using identical conditions ($n = 2$). Cross-linked samples were precipitated with methanol/chloroform (Wessel and Flügge, 1984), resuspended in 8 M urea, reduced with 10 mM DTT, and alkylated with 50 mM iodoacetamide. Following alkylation, proteins were diluted with 50 mM $NH_4HCO_3$ to a final concentration of 2 M urea and digested with trypsin (Promega), at an enzyme-to-substrate ratio of 1:20, overnight at 37 °C or sequentially with trypsin and Glu-C (Promega) at an enzyme-to-substrate ratio of 1:20 and 1:50 at 37 °C and 25 °C, respectively. The samples were acidified with formic acid to a final concentration of 2% (v/v) then split into two equal amounts for peptide fractionation by peptide size exclusion and reverse phase C18 high pH chromatography (C18-Hi-pH). For peptide size exclusion, a Superdex Peptide 3.2/300 column (GE Healthcare) with 30% (v/v) acetonitrile/0.1% (v/v) TFA as mobile phase and a flow rate of 50 μl/min was used, and fractions collected every two minutes over the elution volume of 1.0–1.7 ml. C18-Hi-pH fractionation was carried out on an Acquity UPLC CSH C18 1.7 μm, 1.0 × 100 mm column (Waters) over a gradient of acetonitrile 2–40% (v/v) and ammonium hydrogen bicarbonate 100 mM.

The fractions were lyophilized and resuspended in 2% (v/v) acetonitrile and 2% (v/v) formic acid and analysed by nano-scale capillary LC-MS/MS using an Ultimate U3000 HPLC (Thermo Fisher Dionex, USA) to deliver a flow of approximately 300 nl/min. A C18 Acclaim PepMap100 5 μm, 100 μm × 20 mm nanoViper (Thermo Fisher Dionex, USA), trapped the peptides before separation on a C18 Acclaim PepMap100 3 μm, 75 μm × 250 mm nanoViper (Thermo Fisher Dionex, USA). Peptides were eluted with a gradient of acetonitrile. The analytical column outlet was directly interfaced via a nano-flow electrospray ionization source, with a hybrid quadrupole orbitrap mass spectrometer (Orbitrap Q-Exactive HF-X, Thermo Scientific). Mass-Spectrometry data were acquired in data-dependent mode. High-resolution full scans ($R = 120,000$, $m/z$ 350–2000) were recorded in the Orbitrap followed by higher energy collision dissociation (HCD, stepped collision energy $30 \pm 3$) of the 10 most intense MS peaks. MS/MS scans ($R = 45,000$) were acquired with a dynamic exclusion window of 20 s being applied.

For data analysis, Xcalibur raw files were converted to MGF format by MSConvert (Proteowizard) and put into MeroX (Kessner et al, 2008; Götze et al, 2012). Searches were performed against an ad hoc protein database containing the sequences of the complexes and randomized decoy sequences generated by the software. The following parameters were set for the searches: a maximum number of missed cleavages of three; targeted residues K, S, Y and T; minimum peptide length of five amino acids; variable modifications: carbamidomethyl-Cys (mass shift 57.02146 Da), Met-oxidation (mass shift 15.99491 Da); DSBU modification fragments: 85.05276 Da and 111.03203 Da (precision: 5 ppm MS1 and 10 ppm MS2); false discovery rate cut-off: 5%. Finally, each fragmentation spectrum was manually inspected and validated.

For cross-linking coupled mass spectrometry of the Elp456+Glu$_{10}$ peptide, cross linking reactions were carried out with sulfo-SDA (succinimidyl 4,4'-azipentanoate) and EDC (1-ethyl-3-(3-dimethyla-minopropyl) carbodiimide hydrochloride, protein was cross linked with SDA (2 mM) and EDC (5 mM). SDA and complex were mixed and incubated on ice for 5 min before being cross linked for 10 s with 365 nm UV radiation from a home build UV LED setup. EDC and complex were incubated for 30 min on ice. Both reactions were quenched with the addition of ammonium bicarbonate to a final concentration of 50 mM. The quenched solution was reduced with 5 mM DTT and alkylated with 20 mM idoacetamide. SP3 protocol as described in (Hughes et al, 2014; Batth et al, 2019) was used to clean-up and buffer exchange the reduced and alkylated protein. Shortly, proteins are washed with ethanol using magnetic beads for protein capture and binding. The proteins were resuspended in 100 mM $NH_4HCO_3$ and were digested with trypsin (Promega, UK) at an enzyme-to-substrate ratio of 1:20, and protease max 0.1% (Promega, UK). Digestion was carried out overnight at 37 °C. Clean-up of peptide digests was carried out with HyperSep SpinTip P-20 (ThermoScientific, USA) C18 columns, using 60% Acetonitrile as the elution solvent. Peptides were then evaporated to dryness via Speed Vac Plus (Savant, USA).

Dried peptides were resuspended in 30% acetonitrile and were fractionated via size exclusion chromatography using a Superdex 30 Increase 3.2/300 column (GE Heathcare, Sweden) at a flow rate of 20 µL/min using 30% (v/v) ACN 0.1% (v/v) TFA as a mobile phase. Fractions are taken every 5 min, and the 2nd to 7th fractions containing cross linked peptides are collected. Dried peptides were suspended in 3% (v/v) Acetonitrile and 0.1% (v/v) formic acid and analysed by nano-scale capillary LC-MS/MS using an Ultimate U3000 HPLC (ThermoScientific, USA) to deliver a flow of 300 nl/min. Peptides were trapped on a C18 Acclaim PepMap100 5 µm, 0.3 µm × 5 mm cartridge (ThermoScientific, USA) before separation on Aurora Ultimate C18, 1.7 µm, 75 µm × 25 cm (Ionopticks, Melbourne). Peptides were eluted on optimised gradients of 90 min and interfaced via an EasySpray ionisation source to a tribrid quadrupole Orbitrap mass spectrometer (Orbitrap Eclipse, ThermoScientific, USA) equipped with FAIMS. MS data were acquired in data-dependent mode with a Top-25 method, high-resolution scans full mass scans were carried out ($R = 120{,}000$, $m/z$ 400–1550) followed by higher energy collision dissociation (HCD) with stepped collision energy range 21, 30, 34% normalised collision energy. The tandem mass spectra were recorded ($R = 60{,}000$, isolation window $m/z$ 1, dynamic exclusion 50 s). Mass spectrometry measurements were cycled for 3 s durations between FAIMS CV -45, and −60 V.

Cross linking data analysis: Xcalibur raw files were converted to MGF files using ProteoWizard (Chambers et al, 2012) and cross links were analysed by XiSearch (Mendes et al, 2019). Search conditions used 3 maximum missed cleavages with a minimum peptide length of 5. Variable modifications used were carbmido-methylation of cysteine (57.02146 Da) and Methionine oxidation (15.99491 Da). False discovery rate was set to 5%.

## Statistics and reproducibility

Unless stated otherwise, measurements are given in mean ± SEM. Statistical analyses were performed using GraphPad Prism 8 with an alpha of 0.05. Normality of variables was verified with Kolmogorov–Smirnov tests. Homoscedasticity of variables was always verified when conducting parametric tests. Post-hoc tests are indicated in their respective figure legends. No statistical method was used to predetermine sample size. The experiments were not randomized. All microtubule dynamics experiments were blindly analysed. No data were excluded from the analyses.

All representative results shown for western-blots, protein purifications, microtubule pelleting assays and microtubule TIRFM experiments (including microtubule binding and microtubule dynamics assay) were performed at least three times independently with equivalent results. All cell lines used were tested for mycoplasma contamination.

## Data availability

The mass spectrometry proteomics data have been deposited to the ProteomeXchange Consortium via the PRIDE partner repository with the dataset identifier PXD058480. Cell and in vivo data used in this study are available via the BioImage Archive repository BioStudies with accession numbers S-BIAD1471 (S2 cell data), S-BIAD1472 (NIH/3T3 data) and S-BIAD1473 (*Drosophila* SOP data). Numerical data used to generate every panel of this study can be found in the Source Data. All other data supporting the findings of this study are available from the corresponding author on reasonable request.

The source data of this paper are collected in the following database record: biostudies:S-SCDT-10_1038-S44318-024-00358-0.

## Peer review information

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

## Acknowledgements

This work was supported by the Medical Research Council, as part of United Kingdom Research and Innovation (UK Research and Innovation; grant no. MC_UP_1201/13 to ED) and the Human Frontier Science Program (Career Development Award grant no. CDA00034/2017 to ED). VJP-H and LKK were supported by an EMBO postdoctoral fellowships (ALTF 577-2018 and ALTF 876-2021, respectively). KEM is supported by the Wellcome Trust through a Sir Henry Wellcome Postdoctoral Fellowship (220480/Z/20/Z). CJ was supported by the Agence Nationale de la Recherche ANR-10-IDEX-0001-02, LabEx Cell(n)Scale ANR-11-LBX-0038, Institut Curie, the French National Research Agency (ANR) awards ANR-17-CE13-0021 and ANR-20-CE13-0011, and the Fondation pour la Recherche Medicale (FRM) grant DEQ20170336756. MMM is supported by the Fondation Vaincre Alzheimer FR-16055p and the France Alzheimer grant 2023. MG is supported by Institut Curie 3-i PhD Program (IC-3i). We are grateful to J Grimmett, T Darling and I Clayson for maintenance of the LMB scientific computing infrastructure. We thank the LMB Biophysics facility for access and maintenance of the MST and C-trap equipment, and specially C. Batters for his help with combined IRM-TIRF imaging. We thank the electronics workshop of the LMB, in particular M Kyte for his help with the Field Programmable Gate Array hardware required for precise synchronization of our microscope. We also thank C Lau and S Chaaban for help with a local implementation of AlphaFold2. We thank S Bullock for critical reading of the manuscript.

## Author contributions

**Vicente J Planelles-Herrero**: Conceptualization; Data curation; Formal analysis; Funding acquisition; Validation; Investigation; Methodology; Writing—original draft; Writing—review and editing. **Mariya Genova**: Investigation; Methodology; Writing—review and editing; Purified mouse and HeLa S3 tubulin. **Lara K Krüger**: Investigation; Methodology; Performed fibroblasts experiments, with help of VJP-H. **Alice Bittleston**: Formal analysis; Methodology; Writing—review and editing; Performed fly crosses and dissections, with help of VJP-H for staining and quantifications. Optimized glass coverslip passivation for TIRFM. **Kerrie E McNally**: Methodology; Writing—review and editing; Optimized BiGBac vectors and glass coverslip passivation for TIRFM. **Tomos E Morgan**: Conceptualization; Investigation; Methodology; Writing—review and editing; Performed and analysed XL-MS. **Gianluca Degliesposti**: Investigation; Methodology; Performed and analysed XL-MS. **Maria M Magiera**: Resources; Methodology; Writing—review and editing; Purified mouse and HeLa S3 tubulin. **Carsten Janke**: Conceptualization; Resources; Methodology; Writing—original draft; Writing—review and editing. **Emmanuel Derivery**: Conceptualization; Resources; Software; Supervision; Funding acquisition; Visualization; Writing—original draft; Project administration; Writing—review and editing.

Source data underlying figure panels in this paper may have individual authorship assigned. Where available, figure panel/source data authorship is listed in the following database record: biostudies:S-SCDT-10_1038-S44318-024-00358-0.

## Disclosure and competing interests statement

The authors declare no competing interests.

# Expanded View Figures

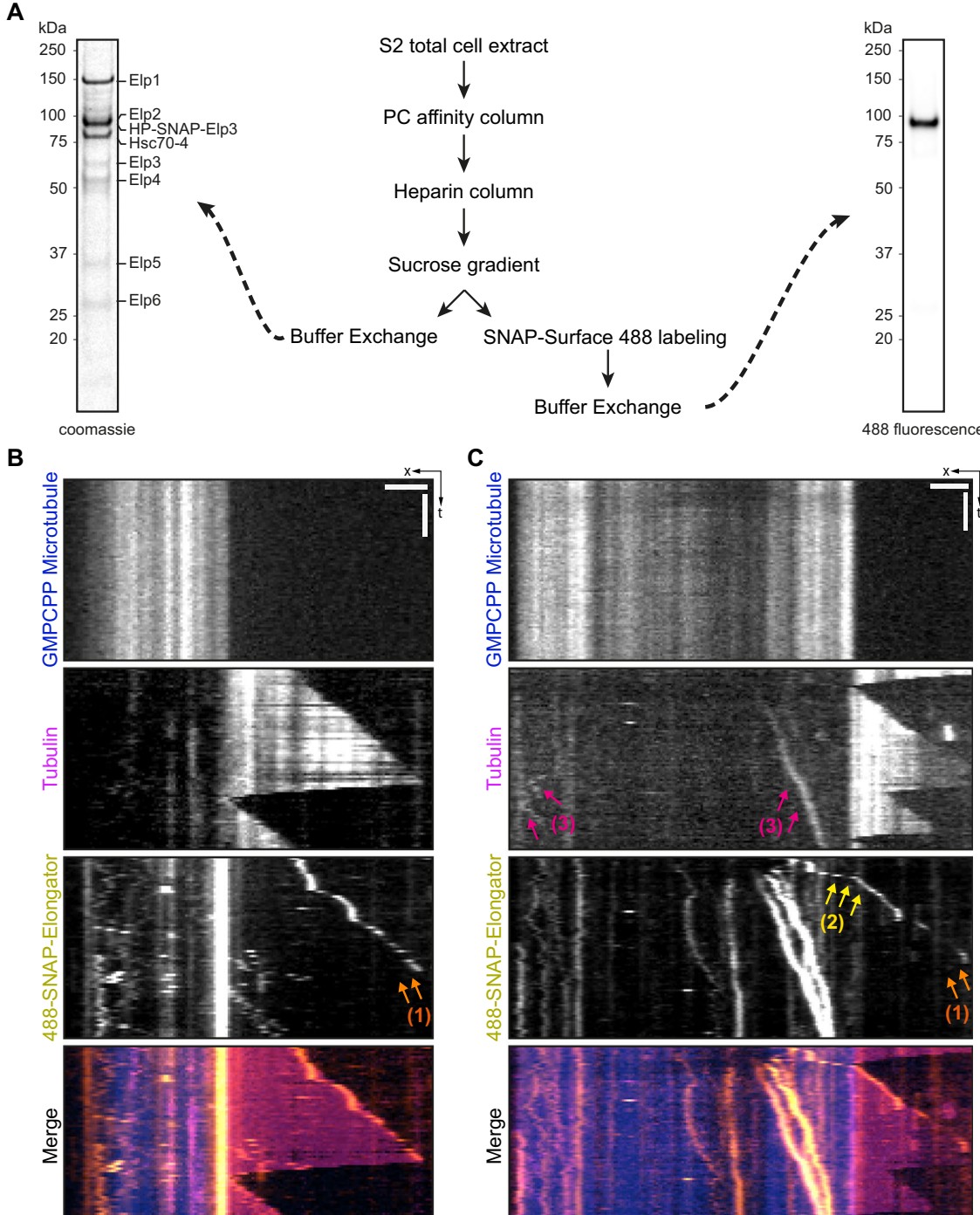

**Figure EV1. Elongator purification and characterization.**

(A) Purification and fluorescent labelling of Elongator complex from *Drosophila* S2 cells (see methods). (B, C) Biotinylated, rhodamine-labelled GMPCPP-stabilized seeds (red) are anchored via NeutrAvidin to PLL-PEG-Silane-Biotin. Free tubulin (10% HiLyte 647-labelled, cyan) and Alexa488-SNAP-Elongator (yellow) is added and imaged by TIRFM. Several behaviors of the Elongator complex can be observed: (1) Elongator detaches from the microtubule ends when microtubules undergo catastrophe (orange arrows); (2) Elongator can "jump" from the GMPCPP seed to the dynamic microtubule, tracking the growing end (yellow arrows); (3) Tubulin signal can be observed diffusing together with Elongator (magenta arrows). Note that 488-Elongator signal can also be observed diffusing on the microtubules. Scale bars = 2 min/2 µm. Source data are available online for this figure.

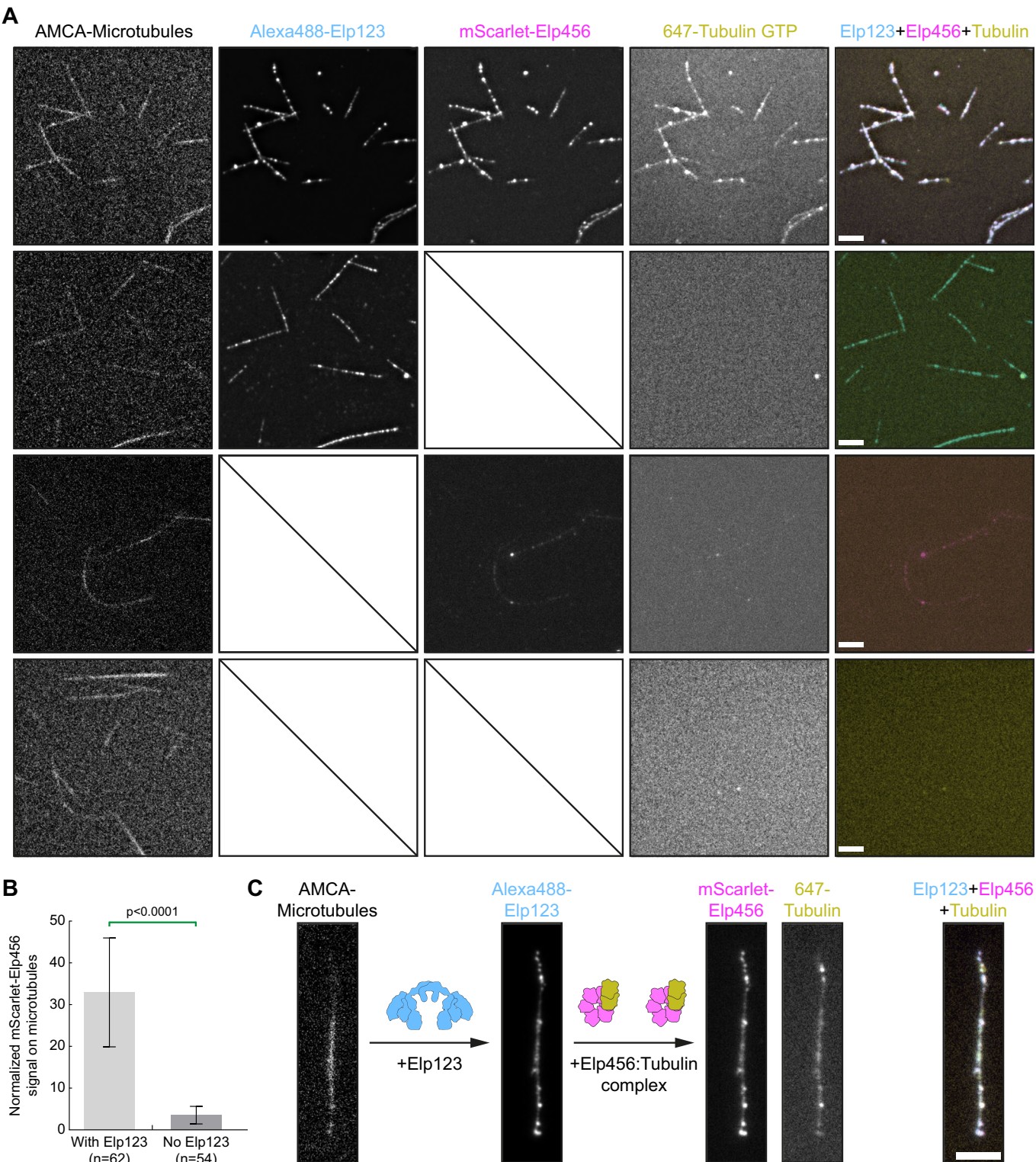

**Figure EV2. Elongator-tubulin complex reconstitution on microtubules.**

(A) Controls for Fig. 4A. Note that the first row is duplicated from Fig. 4A shown here for convenience. In the absence of Elp456, no tubulin signal can be detected on microtubules (second row). Similarly, in the absence of both Elp123 and Elp456, no tubulin signal is detected on microtubules (bottom row). Note that in the absence of Elp123, a weak mScarlet-Elp456 signal can be observed on microtubules (third row). (B) This signal is however significantly weaker than when Elp123 is also present (*P* < 0.0001 two-tailed, unpaired t-test). Error bars are mean ± standard deviation for quantified intensities. *n*, quantified microtubules. (C) A pre-formed mScarlet-Elp456:647-Tubulin complex can be recruited to Elp123-decorated microtubules. Scale bars = 5 μm. Source data are available online for this figure.

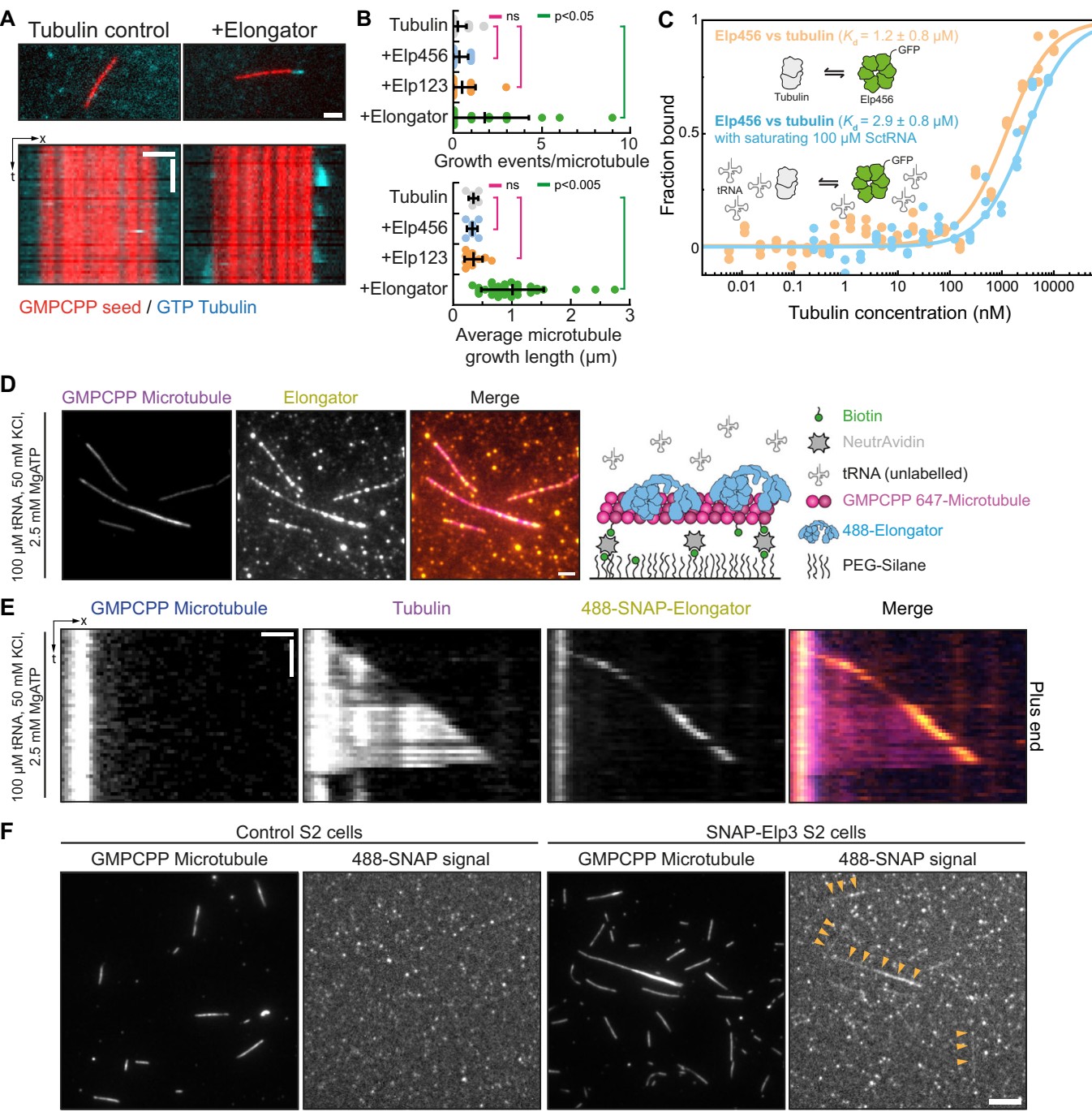

**Figure EV3. Elongator can discriminate between microtubules and tRNA.**

(A, B) Elongator decreases the effective critical concentration of tubulin for microtubule elongation. (A) Rhodamine-labelled GMPCPP stabilized microtubules incubated with GTP-tubulin near the critical concentration (6 µM) and imaged by TIRFM in the presence or absence of Elongator. (B) Quantification of the number of microtubule growth events observed, and their length, in (A). (Upper panel) $n =$ quantified microtubules $= 20$. $P$ values were calculated for a two-tailed Kruskal–Wallis test followed by Dunn's multiple comparison test. Exact $P$ values: Tubulin vs Elp456 > 0.9999, Tubulin vs Elp123 > 0.9999, Tubulin vs Elongator 0.0320. (Lower panel) $n =$ observed growth length in the growth events quantified for the upper panel $= 6$ (Tubulin), 7 (+Elp456), 10 (+Elp123) and 36 (+ Elongator). $P$ values were calculated for an Ordinary one-way ANOVA test followed by two-tailed Turkey multiple comparison. Exact $P$ values: Tubulin vs Elp456 > 0.9999, Tubulin vs Elp123 0.9995, Tubulin vs Elongator 0.0024. Elongator concentrations used are as in Fig. 4. (C) Measurement of the affinity between tubulin and eGFP-Elp456 in high tRNA buffer (100 µM tRNA, 2.5 mM MgATP) using microscale thermophoresis (see "Methods"). Estimated dissociation constant ($K_d$) values are indicated (mean ± s.d.; $n = 3$). Note that the "Elp456 vs tubulin" dataset is the same as the one presented in Fig. 5A, shown here for convenience. (D) Rhodamine-labelled GMPCPP stabilized microtubules incubated with 25 nM SNAP-Elongator labelled with AlexaFluor488 dye in a buffer containing 100 µM tRNA, 50 mM KCl and 2.5 mM MgADP and observed by TIRFM. Salt and high tRNA concentrations do not prevent microtubule binding by Elongator. (E) Representative kymograph showing Alexa488 SNAP-Elongator complex tracking the plus end of a microtubule in the same conditions as in (D). (F) Total cell extracts of *Drosophila* S2 cells incubated with rhodamine-labelled GMPCPP stabilized microtubules and Alexa-488 SNAP ligand . In non-transfected cells extracts (left) no signal can be detected on microtubules. In extracts of cells expressing SNAP-Elp3 (see methods), clear signal can be observed on microtubules (orange arrows). Scale bars $= 2$ µm (A, D), 2 min/2 µm (A, E), 5 µm (F). Source data are available online for this figure.

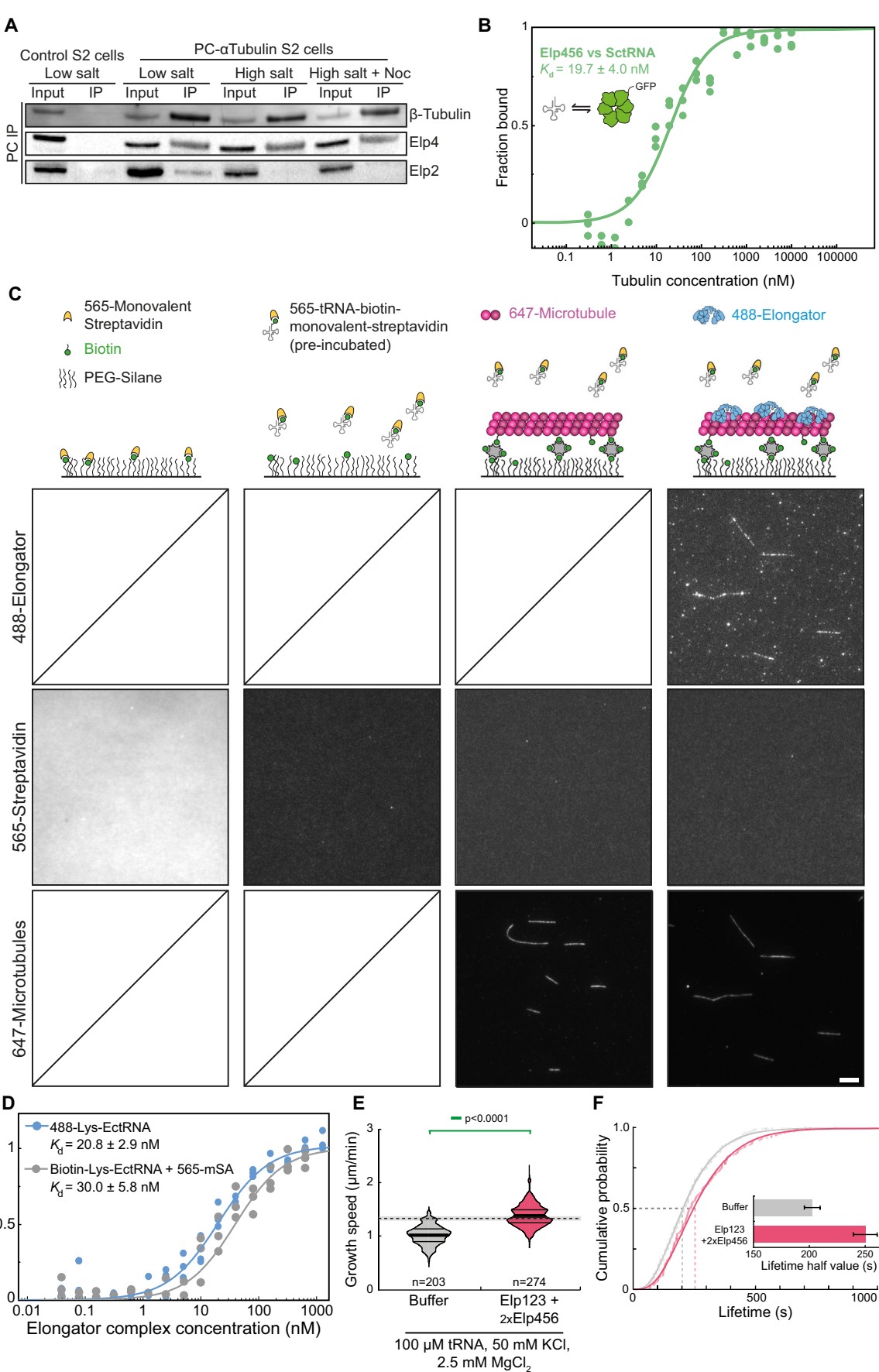

◄

**Figure EV4. Elongator does not bind to tRNA while bound to microtubules.**

(A) PC-tag immunoprecipitation of PC-α-Tubulin from S2 cells expressing PC-α-Tubulin. Low salt: 50 mM KCl. High salt: 300 mM KCl. Note that his PC-tag immunoprecipitation is performed at 4 °C and in the presence of 1 mM CaCl$_2$, conditions which are well established to depolymerize microtubules and therefore lead to tubulin dimers rather than microtubules. (B) Measurement of the binding affinity between eGFP-Elp456 and tRNA (total tRNA from *S. cerevisiae*). Calculated dissociation constant ($K_d$) values are indicated (mean ± s.d.; $n = 3$). (C) (First column) ATTO565 labelled monovalent streptavidin (mSA2-565) is functional and binds to the PEG-Silane-Biotin. (Second column) mSA2-565 saturated with tRNA-biotin (Promega) using a 10:1 ratio (tRNA:mSA2-565) does not bind to the surface. (Third column) tRNA-biotin-mSA2-565 (10:1 ratio as in the second column) does not bind rhodamine-labelled, biotinylated GMPCPP-microtubules attached to the surface via neutravidin. Note that after the addition of microtubules, a solution of 500 µg/ml biotin was added in the chamber to quench the neutravidin. Then, the tRNA-biotin-mSA2-565 mixture was added. (Fourth column) While bound to rhodamine-labelled, biotinylated GMPCPP microtubules, SNAP-Elongator labelled with Alexa488 does not bind to tRNA-biotin-mSA2-565. (D) tRNA-biotin-mSA2-565 binds to (unlabelled)-SNAP-Elongator with similar affinity as to 488-tRNA. (E, F) Effect of the indicated conditions on the growth speed (E) and lifetime (F) of microtubules at the plus end imaged by TIRFM in the presence of 100 µM tRNA, 50 mM KCl and 2.5 mM MgCl$_2$. (E) *n*, number of microtubule-growing events analysed. *P* values for a Kruskal–Wallis test followed by Dunn's multiple comparison test are indicated. Dashed line represents an increase of ~1.4 in the speed of microtubule growth. Thick line, median; thin line, quartile. (F) Microtubule lifetime estimate ± error from the bootstrapped mean lifetimes (see "Methods") are indicated in the right panel. Number of microtubule-growing events analysed as in (E). Scale bar = 5 µm. Source data are available online for this figure.

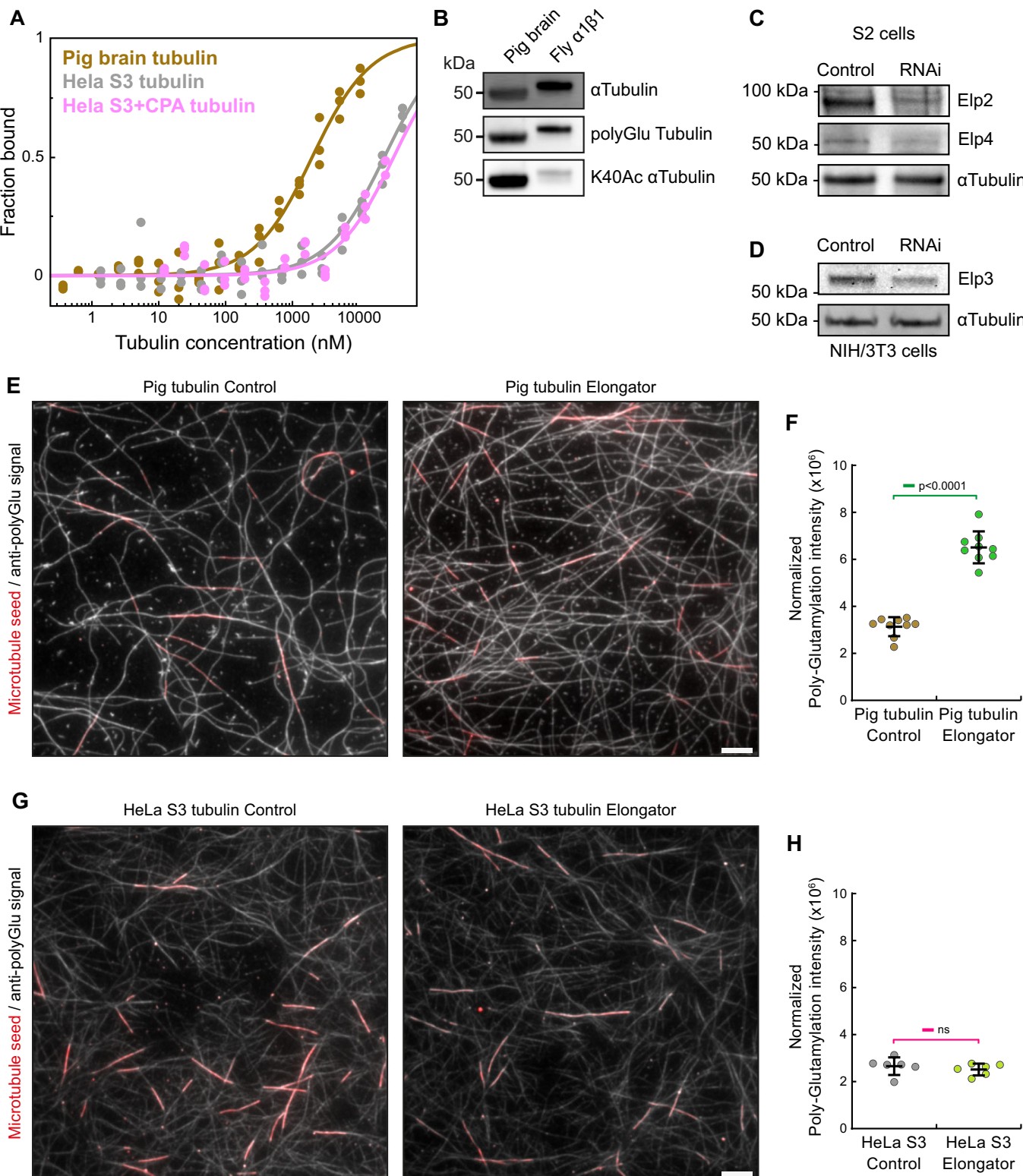

◀  **Figure EV5.   Elongator is a polyglutamylated tubulin polymerase.**

(**A**) Measurement of the affinity between eGFP-Elp456 and αβ-tubulin heterodimers (as indicated) using microscale thermophoresis. Calculated dissociation constant ($K_d$) values are indicated (mean ± s.d.; $n = 3$). See also Fig. 5A,B. (**B**) Western blot for poly-Glutamylation and Lysine 40 acetylation (K40Ac) for pig brain and recombinant *Drosophila* α1β1 tubulin. (**C, D**) Western blots confirming the RNAi treatment in S2 cells (**C**) and NIH/3T3 cells (**D**). Note that Elp3 was not probed for in (**C**) as the Elp3 antibody does not work in lysates. (**E, F**) Representative field of view of microtubules labelled with fluorescently-labelled anti-polyglutamylated tubulin antibodies (gray) and HiLyte488-labelled GMPCPP stable seeds (red). Conditions as indicated. *P* value for an unpaired two-tailed t-test. Error bars are mean ± standard deviation for quantified intensities. *n* (fields of view analysed) = 9 from three independent experiments. (**G, H**) Quantification of total poly-glutamylated tubulin signal (see "Methods"). *P* value for an unpaired two-tailed *t* test (*P* value = 0.4547). Error bars are mean ± standard deviation for quantified intensities. *n* (fields of view analysed) = 6 from three independent experiments. Scale bars = 5 μm. Source data are available online for this figure.

