## [Peer Review File · The EMBO Journal]

Elongator is a microtubule polymerase selective for polyglutamylated tubulin

Vicente Planelles-Herrero, Mariya Genova, Lara Krüger, Alice Bittleston, Kerrie McNally, Tomos Morgan, Gianluca Degliesposti, Maria Magiera, Carsten Janke, and Emmanuel Derivery

Corresponding authors: Vicente Planelles-Herrero (vicente@mrc-lmb.cam.ac.uk) , Emmanuel Derivery (derivery@mrc-lmb.cam.ac.uk)

Review Timeline:

Submission Date:	29th Aug 24
Editorial Decision:	11th Oct 24
Revision Received:	4th Dec 24
Editorial Decision:	10th Dec 24
Revision Received:	12th Dec 24
Accepted:	19th Dec 24

Editor: Ieva Gailite

Transaction Report:

(Note: Please note that the manuscript was previously reviewed at another journal. As EMBO Press has a transfer agreement with that journal, revision was invited based on the reports from that previous external submission. With the exception of the correction of typographical or spelling errors that could be a source of ambiguity, letters and reports are not edited. Depending on transfer agreements, referee reports obtained elsewhere may or may not be included in this compilation. Referee reports are anonymous unless the Referee chooses to sign their reports.)

Reviewers' comments:

Reviewer #1 (Remarks to the Author):

Comments for the authors

In the presented study Planelles-Herrero and colleagues expand their recent work on the link between Elongator and microtubule dynamics. In detail, they aim to unravel the molecular mechanism by which Elongator affects the growing tip of microtubules and the incorporation of specifically modified tubulin dimers.

Elongator has been identified as a macromolecular complex that acetylates histones and facilitates transcription elongation by RNAPII – hence the name. Subsequently, the complex has been associated with numerous cellular activities, which were all more or less connected to its proposed histone/protein acetylation activity. Over the last decade, several studies showed that the complex is rather involved in translational control and the genuine enzymatic activity of the complex relates to the specific modification of tRNAs at their wobble base position. The first step of the reaction involves the acetylation of uridines at the C5 position and the modification is crucial to assure proper ribosomal elongation speed. Therefore, most of the associated cellular activities can be linked to problems in translation of specific proteins with a certain codon bias. Of note, several clinically relevant mutations have been found in patients that affect Elongator subunits. It was concluded that the tRNA modification activity of the complex is compromised by the mutations, which causes a specific response in neurons and triggers the observed diseases.

In a recent study also published in Nature Cell Biology the authors provided an exciting new observation – namely the asymmetric association of Elongator with microtubules of the mitotic spindle. The complex apparently affects the dynamics of spindle formation and importantly, this activity does not require the acetylation activity of the complex.

In this study the authors provide a follow-up study to understand the molecular mechanisms behind this moonlighting function during cytokinesis. They show that Elongator binds to the tip of GTP-bound microtubules and also to heterodimers of free tubulin. Furthermore, they show that the two Elongator subcomplexes (Elp123 and Elp456) have different binding preference – Elp123 binds to microtubules and Elp456 binds to free tubulin. Strikingly, these activities must be coupled for Elongator to stabilize microtubules. Last but not least, they found that Elp456 shows strong selectivity towards polyglutamylated tubulin dimers and that Elongator seems to selectively enrich microtubules with polyglutamylated tubulin.

I very much appreciate the work and the attempts to understand the mechanistic details of the observed activity. However, I feel that despite the new insights the advances of this study the authors need to more carefully/critically assess and validate their findings, before making far reaching mechanistic conclusions. I am very excited about the work of the authors, but I feel that the authors overinterpret their findings and make several overstatements. I have a few major issues that need to be addressed before considering this study at Nature Cell Biology. At the moment, I still have major reservations and I don't support the publication of the work in its current form.

*We thank the reviewer for this positive assessment of our work and want to warmly thank them for their very constructive comments, addressed below, which we think do experimentally and conceptually strengthened the paper. We do hope that the extensive novel *in vitro* and *in vivo* data we provide in this revised version, as well as the extra controls we added throughout the paper, will convince this reviewer to now support publication.*

Major

RI.1. *In the previous study, it was very obvious that Elongator associates with the mitotic spindle and promotes its asymmetry. In this study, it seems that Elongator (or at least Elp123) can bind to any microtubule. I am missing a rational experimental approach that answers this conflicting observations and a clear mechanistic explanation. From the provided experiments, it is simply not clear in which cell type the observed activity would be necessary and in which cell types not – understand the cellular context would allow to draw more precise conclusions about the neuron- or disease relevance of the observed mechanism. Could it be that the link between Elongator mutations and neurodegenerative diseases is related to its microtubule association, its tRNA modification activity or both?*

There are two excellent questions here: i) what is Elongator selectivity regarding the various microtubule species present in cells, and ii) in which cells Elongator's microtubule binding activity would be relevant. Obviously, both are related, but for clarity we will address them independently:

i) Selectivity towards microtubules

The reviewer is right that in the current paper, we show that purified Elongator, either the semi-endogenous complex purified from *Drosophila* S2 cells, or the reconstituted complex purified from Sf9-baculovirus (Elp123) and bacteria (Elp456), binds to any kind of microtubule *in vitro*, no matter their origin (mammalian brain microtubules, which contain all modifications/isotypes, subtilisin-treated brain microtubules, which contain multiple isotypes but no modifications, HeLa S3 microtubules, with low modifications, as well as *Drosophila* recombinant microtubules).

The reviewer is also right that in our previous paper, we found that Elongator decorates mitotic microtubules during asymmetric cell division in *Drosophila* Sensory Organ Precursors (SOP cells, Planelles-Herrero *et al.* Nature Cell Biology 2022).

But both can be true at the same time. Indeed, in our previous study we did not claim that Elongator was *only* binding to spindle microtubules in SOPs, nor did we say that Elongator was *only* having a phenotype in SOPs. It's just that this is where we looked and found a phenotype (that paper was focused on the mechanisms of central spindle asymmetry, which we discovered in these cells). In fact, we did not state that Elongator was *only* present on spindle microtubules, and we provided *in vitro* experiments in which the semi-endogenous Elongator complex purified from *Drosophila* S2 cells fully decorated pig brain microtubules. Again, that study was focussed on spindles in SOP cells, and we did not investigate other non-spindle microtubules, such as astral microtubules, as they are very hard to image in these small cells.

In fact, the novel paradigm we propose where Elongator's activity changes the pattern of microtubule poly-glutamylation only makes sense if the part of Elongator that binds to microtubules does not have any specificity, but the part that binds to free tubulin does. Indeed, if Elongator was selectively not binding to poly-glutamylated microtubules, or in other words would detach from poly-glutamylated microtubules, then Elongator activity would never lead to microtubules selectively elongated with poly-glutamylated tubulin, as Elongator would stop binding after few poly-glutamylated dimers have been added to the preexisting microtubule. Conversely, if Elongator was only binding to poly-glutamylated microtubules, then obviously the pattern of poly-glutamylation would not change.

Now, this being said, it is clear from our data and the mechanism we propose that Elongator does need an on/off switch that controls its activity towards microtubules, for three main reasons:

1) When bound to microtubules, Elongator cannot bind to tRNAs (new Sup. Fig. 6). Cells must therefore have a way of switching off Elongator's binding to microtubules, otherwise tRNAs would never be modified.

2) Similarly, since Elongator enriches microtubules in poly-glutamylated tubulin as they grow, there must be a way of controlling its activity, as it's well established that not all microtubules in cells are poly-glutamylated, especially in polarized cells.

3) When we assessed the ability of Elongator to modulate microtubule poly-glutamylation levels *in vivo*, we found that it seemed to only have an effect in mitosis, not interphase (see new Fig. 6E). This suggests that indeed the activity of Elongator towards microtubules *in vivo* can be modulated in time.

We envision two scenarios: either the binding to microtubules is regulated via Elongator post-translational modifications (PTMs), or via a protein partner. This is consistent with the fact that in our *in vitro* microtubule pelleting assays, there is a reproducible fraction of Elongator that remains in the supernatant after centrifugation (Planelles-Herrero *et al.* Nature Cell Biology 2022, and this study new Sup. Fig. S2D, E). It is thus possible that a PTM regulates the binding to microtubules, and the supernatant fraction represents the pool of Elongator that is not able to bind to microtubules (i.e. "inactive"). We previously attempted to analyse the PTM content of both fractions by mass-spectrometry, but we found that the amount of Elongator is too low to provide consistent data. Alternatively, the binding of Elongator could be regulated via a binding partner. As the purified Elongator does bind to microtubules, this could suggest the existence of a negative regulator, switching Elongator's activity from microtubule-binding to tRNA-binding.

To conclude, in a nutshell, we think it works the other way around: it's not preexisting microtubule modifications that selectively recruit Elongator during mitosis, it's an unknown regulatory mechanism that activates Elongator during mitosis to selectively recruit poly-glutamylated microtubules to the spindle. It is very tempting to speculate that mitosis itself is at the heart of this switch. Indeed, it is well established translation shuts down during mitosis, so it would make sense that the same trigger that turns down translation also switches the activity of elongator from its role in modulating translation to its role in modulating microtubule dynamics. But obviously this is speculating beyond our data so we would rather remain cautious. So, while the reviewer raises an excellent point, and we agree that the next burning question is to understand how Elongator activity is regulated in cells, we felt it would be beyond the scope of this first paper to experimentally address this point as our recent data suggest that this will be a project in itself. We have nevertheless included a paragraph in the revised version to highlight this point (line 282):

“On the other hand, however, we found that microtubule binding by Elongator prevents the tRNA-Elongator interaction (Sup. Fig. S7A). Indeed, using TIRFM we observed that microtubules decorated by Elongator could not recruit fluorescently-labelled tRNA, suggesting that the conformation adopted by Elongator on the microtubules is not compatible with tRNA binding. Importantly, Elongator alone (i.e., in the absence of microtubules) was able to bind to fluorescent-tRNA (Sup. Fig. S7B). Since tRNA binding induces a conformational change on Elp123²⁴, particularly on the Elp1 subunit, it is conceivable that binding to microtubules prevents this rearrangement, and thus efficient tRNA binding. Together, our data suggests that Elongator binds and stabilizes microtubules in physiological conditions containing another highly abundant substrate, tRNA. However, upon binding to microtubules, Elongator loses its ability to interact with tRNA, functionally separating its microtubule stabilizing and tRNA modifying activities. Understanding the regulatory mechanisms that switch Elongator between microtubule binding and tRNA modification is a critical question for future studies.”

B) Cell type specificity

The reviewer raises a valid point regarding the cell type specificity of Elongator.

Our previous work demonstrated the role of Elongator in controlling central spindle asymmetry in *Drosophila* Sensory Organ Precursors. In our revised manuscript, we provide *in vivo* evidence that Elongator indeed controls the levels of poly-glutamylation of the spindle microtubules not only in various cultured cells in different species (*Drosophila* S2, mouse 3T3 fibroblasts), but also in actual tissues in *Drosophila* Sensory Organ Precursors (new Fig. 6E-G). Indeed, in cells or flies depleted for Elongator, microtubules at the spindle show a significant decrease in their levels of poly-glutamylation. This highlights how the activity of Elongator towards microtubules that we characterised *in vitro* is relevant in different cell types and organisms. As discussed above, our data also suggest that the position in the cell cycle might also matter for Elongator's activity.

Now, the reviewer is right that the phenotypes of mutations in Elongator subunits suggest a more prominent role of Elongator in neurons and in the brain, and we agree with the reviewer that it is possible that some of the Elongator mutations linked with neurodegenerative diseases involve its activity towards microtubules, tRNA modification or both. Indeed, several patient-derived mutations cluster on the surface of Elp456 involved in the binding with Elp123 (new Sup. Fig. S7E), which would affect both activities. There are also results in the literature, however, suggesting that the effect on microtubules is relevant in neurodegenerative diseases. For instance, Elongator depletion leads to defects in neurite elongation in neuroblastoma cells, a phenotype similar to that of TLL7, an important brain poly-glutamylase (Creppe *et al.* Cell 2009, Ikegami *et al.* Journal of Biological Chemistry 2006). And obviously any role of Elongator during mitosis could affect neuronal development during embryogenesis.

We hope the reviewer agrees that whilst this a very interesting avenue for future research, it is beyond the scope of this current study to characterise this link in depth. Instead, we focused on showing how the activity of Elongator is relevant *in vivo* and *in cells* from different lineages and organisms, highlighting the relevance of the activity of Elongator towards microtubule in physiological conditions. Given all the new data we provide, we hope the reviewer will agree with our reasoning.

RI.2. *The authors mention the other (canonical) activity of Elongator, but none of their experiments addresses a possible link between tRNA modification activity and microtubule association. The authors should check whether the initial binding of Elp123 is physiologically relevant in the presence of an excess of tRNAs. tRNA are highly abundant and bind Elp123 with a similarly high affinity – I am very curious, if Elp123 binding to microtubules would happen if Elp123 is loaded with tRNAs. In addition, I was wondering whether the assays in Figure 3,4,5 and 6 contains ATP (or a nucleotide derivative), which is known to have a strong effect on the interaction between Elp456 and tRNA?*

This is an excellent point raised by both reviewer 1 and reviewer 2. We really want to thank both reviewers for this constructive comment and excellent suggestions, which strengthened the manuscript.

As pointed out by this reviewer, tRNA and tubulin are of a roughly similar size, both molecules are very abundant and heavily negatively charged, and both bind to Elongator. So the key question is to know if the activity of Elongator towards tRNA and microtubules can be untangled, or whether they are actually two sides of the same coin. In other words, who wins?

Critically, we now provide multiple orthogonal lines of evidence both *in vitro* and *in vivo* that **all the activities of Elongator towards microtubules still occur in the presence of physiological concentrations of tRNA, ATP and salts**. Specifically, we now show that:

1) Elp123 binds to microtubules and tracks their growing ends in the presence of 100 μ M tRNA and 50 mM KCl (see new Sup. Fig. S5A, B). Note that 100 μ M tRNA represents a huge excess compared to both Elp123 (25 nM) and tubulin (16 μ M) present in these assays. These assays also included 2.5 mM MgATP in our buffers, a reported optimal concentration to favour the interaction between Elp456 and tRNAs (Glatt *et al.* Nat Struct Mol Biol 2012, Jaciuk *et al.* Nucleic Acids Res 2023). Additionally, note that we decided to use 50 mM KCl for our *in vitro* experiments as buffers with higher concentrations of salts are known to disrupt the Elp123 to Elp456 interaction (our own observation, but also reported in Setiaputra *et al.* EMBO Reports 2017).

2) Elp456 binds to tubulin with similar affinity as without tRNA (See new Sup. Fig. S5E, F, conditions are 100 μ M tRNA, 50 mM KCl and 2.5 mM MgATP).

3) We consistently show that the effects of Elongator on microtubule dynamics (increase of polymerisation speed and microtubule lifetime) are quantitatively the same in the presence of 100 μ M tRNA and 2.5 mM MgATP, as in their absence (see new Sup. Fig. S5G, H).

This establishes that the known function of Elongator to bind to tRNA does not affect its ability to modify tubulin dynamics, at least in our *in vitro* assays. But following a line of reasoning of reviewer #2, we thought to go one step further and show that Elongator binding to microtubule and tubulin still occurs in extracts (in other words in presence of not only of tRNAs, but also of all other nucleic acids, microtubules associated proteins and other metabolites found in the cytosol in the right physiological ratios). Indeed, we could further demonstrate that:

4) Elongator binds to microtubules in cell extracts (new Sup. Fig. S5C). In this experiment, we mixed GMPCPP-stabilized microtubules with cytosolic extracts of insect cells expressing virtually endogenous levels of fluorescently labelled SNAP-Elongator (as the subunit we express, Elp3, is titrated by the other subunits) and detected clear decoration of microtubules by Elongator. As we independently demonstrated that Elongator directly binds to microtubules, this confirms that the Elongator-Microtubule interaction still occurs in the presence of physiological concentrations of tRNA, as well as other Elongator binding partners. Moreover, this key experiment further demonstrates that the binding of Elongator to microtubules is strong enough, and also specific enough, to be observed in the presence of all other cytosolic microtubule-associated proteins that could compete with Elongator for microtubule binding. As the direct observation of Elongator in cells during interphase is currently not technically possible (see our answer to comment **R2.2** of reviewer #2), this is the closest that we can get to observing Elongator decorating microtubules in physiological conditions.

5) Additionally, we show that the second binding mode of Elongator, namely of Elp456 to globular tubulin, also happens in extracts (see new Sup. Fig. S5D).

Together, these experiments confirm that the binding of Elongator to tRNA and microtubules/tubulin are independent and can be untangled, or, in other words, that **there is no competition between tRNA and microtubules to bind to Elongator**. To further strengthen this point, and to map the surface of Elp456 involved in tubulin binding, we also performed crosslinking coupled to mass-spectrometry with a poly-Glu (Glu₁₀) peptide with Elp456. We decided to use a synthetic peptide to mimic a poly-glutamylated tubulin tail because native tubulin is too heterogeneous to unambiguously assign crosslinks to the tails in our hands, even when using isotype-pure tubulin. Using this peptide,

we detected reproducible crosslinks with several residues on the surface of Elp456 (see new Sup. Fig. S10, new Sup. Table 2 and our reply to point **R1.3** below). This likely explains why there is no competition between tubulin and tRNA binding, as tRNA binding was mapped to the centre of the Elp456 ring, away from the detected crosslinks (Glatt *et al.* Nat Struct Mol Biol 2012). However, we found that once bound to microtubules, Elongator cannot bind to tRNA (new. Sup. Fig. S6A). Since tRNA binding has been shown to induce a conformational change on Elp123 (Dauden *et al.* Sci Adv 2019), our results suggest that the binding to microtubules prevents this rearrangement.

6) Last, we now demonstrate that the novel microtubule activity we uncovered *in vitro* for Elongator, namely to control the levels of poly-glutamylation of microtubules, also occurs *in vivo* at the spindle in various cultured cells in different species (*Drosophila* S2, mouse 3T3 fibroblasts), as well as in *Drosophila* Sensory Organ Precursors (see new Fig. 6E-G). These *in vivo* results imply that Elongator's activity towards microtubules can occur in the presence of physiological concentration of tRNA and nucleotides (and whatever else contained in the cytosol for that matter).

We think that the combination of novel *in vitro*, *ex vivo* and *in vivo* evidence we provide in the revised version of our paper, where experiments at one scale confirm experiments at another scale, makes an unambiguous point that the effects of Elongator on microtubule dynamics are relevant under physiological conditions and truly represent a moonlighting function of Elongator in addition of its canonical role on tRNA.

We have added a full new section in our manuscript detailing these results “Elongator can discriminate between microtubules and tRNA” starting in line 245, and introduced our new XL-MS (line 431) and *in vivo* results in a new paragraph at line 484.

R1.3. *The authors try to determine the cryo-EM structure of Elp456 bound to tubulin, but they don't see any additional density in their cryo-EM maps. This is one of the major weaknesses of the manuscript and needs to be addressed to provide the foundation of a mechanistic model. The shown 2D classes show a clear orientational bias and I don't fully understand the provided rationale of the authors – “We thus hypothesized that the binding must involve very flexible regions of Elp456, tubulin, or both, which could explain the absence of extra density for tubulin after averaging”. In low pass filtered maps, a signal should be detectable. Alternatively, the authors could use a similar XLMS approach to map the interface between Elp456 and tubulin – like they did for the interaction between Elp123 and Elp456. In general the description of technical details, dataset quality and processing steps of the cryo-EM workflow is very weak and lacks detailed information.*

We agree that our initial description of the cryo-EM workflow was incomplete, this was also a concern of reviewer #3 (point **R3.5**). We have now included a more detailed explanation of our workflow, processing and rationale. We similarly include now 2D classes where additional signal is detectable outside of the Elp456 molecule (new Sup. Fig. S8A-D).

However, we still could not unambiguously attribute the extra signal to tubulin. Indeed, as the binding of Elp456 to tubulin involves the tubulin tails, this would mean that the tubulin is only anchored via one or two points to Elp456, and so it is not locked to a single position or orientation. This represents too many degrees of freedom to allow the correct alignment of both molecules, and indeed it can be observed in the new 2D classes (new Sup. Fig. S8C) that the alignment algorithms cannot deal with such big variability (both using RELION and cryoSPARC). Note that due to the orientation bias, pointed by the reviewer, and the inability to properly align Elp456 and tubulin, we could not reconstruct a reasonable 3D map. We nevertheless provide the resulting highly anisotropic map

obtained from the selected 2D classes clearly showing a signal outside of Elp456 (new Sup. Fig. S8E). Note that we tried to solve the orientation problem using all available tools to our reach, as described now in the methods section (see below), but without success. Preliminary data suggest that this problem is specific to the *Drosophila* Elp456, however, as the human Elp456 does not seem to have such a big orientation bias in our hands. However, as the structure of the human Elp456 is known (Gaik *et al.* EMBO Molecular Medicine 2022), and as this would not solve the problem of the inability to align tubulin, we have not collected a full dataset of this sample.

So, instead of pursuing a CryoEM approach, we thought to independently confirm the interaction between Elp456 and tubulin tails by XLMS, as suggested by the reviewer. Specifically, we performed XLMS of *Drosophila* Elp456 and a tubulin tail-mimicking peptide, poly-Glu (Glu₁₀) peptide. This identified several residues on the surface of Elp456 that interact with a poly-glutamylated tubulin tail, as well as likely explain molecularly why tRNA and tubulin binding are not in competition (see new Sup. Fig. S10, new Sup. Table and our answer to point **R1.2** above). Together with our biochemical affinity measurements with purified modified tubulin clearly indicating that tubulin poly-glutamylation affect Elp456 direct binding, we think that these new results make the point that Elp456 directly interacts with the poly-glutamylated tail of tubulin. We hope the reviewer agrees with our reasoning.

We added the following text in the revised manuscript:

In the results section, starting at line 367:

“[...] This suggested that the binding could involve very flexible regions of Elp456, tubulin, or both, which could explain the absence of extra density for tubulin after averaging. We thus processed the datasets aiming at maximizing the signal outside of the Elp456 ring (see methods). This resulted in a blurry signal located in close proximity to the Elp456 ring (Sup. Fig. S9B). Further optimization of the 2D classification process resulted in a defined signal next to the Elp456 ring (Sup. Fig. S9C, D), compatible with the size of a tubulin dimer⁴⁶. Note that during this 2D analysis step we cannot exclude the extra-signal belonging to tightly-packed neighbouring Elp456 molecules. However, due to the preferential orientation adopted by the complex on EM grids we could not further 3D reconstruct the data (see methods for strategies followed to mitigate the preferential orientation). Indeed, 3D reconstruction of the classes shown in Sup. Fig. S9C results in a heavily anisotropic, low-resolution map of a six-lobed ring, in agreement with the 2D averages and with previously solved CryoEM structure of the Elp456 subcomplex^{28,39} in close proximity to a non-identified signal compatible with the size of a tubulin dimer (Sup. Fig. S9E). Thus, due to the limitations imposed by the sample we could not identify the surfaces involved in the binding by CryoEM.”

In the methods section, starting at line 1064:

“[...] Note that different strategies were tried to reduce the preferential orientation problem observed in the sample (including, but not limited to changing buffer composition and pH; detergents; protein concentration, tags, and batches; grid type; grid support; freezing conditions...). Since we could not detect any extra signal outside of the Elp456 ring, processing was tried in CryoSPARC⁸⁶⁻⁸⁸ with initial similar results as in RELION. However, the use of a bigger extraction box and mask diameter (~300 and 260 Å, respectively) and extended 2D classification iterations resulted in some classes displaying extra signal in close proximity to the Elp456 ring (see pipeline in Sup. Fig. S9D). The particles associated with these classes were further classified and used for TOPAZ training⁸⁹ and picking in CryoSPARC. The picked particles were 2D classified for several rounds using the same extraction box and mask diameter, resulting in the classes displayed in Sup. Fig. S9C. Then, the particles were used for ab-initio 3D reconstruction and refinement (including homogeneous, heterogeneous and non-

Uniform refinements). The highest quality model, corresponding to homogeneous refinement followed by NU refinement, is displayed in Sup. Fig. S9E. Note that similar results were obtained when ab-initio reconstruction and heterogeneous refinement was tried right after the first round of 2D classification to reduce bias. Similarly, using the selected particles in RELION for 3D classification, with and without alignment, did not result in any further improvement (not shown).”

RI.4. *The authors do not show how Elp123 would bind to microtubules. I assume the authors should be able to locate Elp123 on microtubules using either single particle cryo-EM reconstructions or cryo-tomography (or CLEM). Elp123 is relatively large and should be visible even at a low resolution reconstruction. This would reveal how the complex is able to specifically bind microtubules. I could imagine that a XLMS approach could also reveal the interface or domain of Elp123 that is needed for the interaction. Is the dimer of Elp123 required for microtubule binding – Xu et al 2015 describe a very specific mutant in the Elp1 dimer interface that results in Elp1 monomers. Did the authors test this mutant for microtubule binding? This could give additional hints on the interface. The additional data needs to be incorporated in Fig. 2 to support the major claims of the authors.*

We agree that knowing structurally how Elp123 binds to microtubules would provide a deeper mechanistic understanding of how Elongator stabilizes microtubules, and specifically what features Elongator recognises to track the growing ends of microtubules.

However, despite all our attempts over the last years, we have not managed to generate a reconstruction of Elp123 bound to microtubules, even at low resolution. This is mainly caused by the high molecular weight of the complex and the relatively low yield of pure fly Elp123 that we obtain from our purifications. Indeed, cryo-EM reconstructions of decorated microtubules typically use small proteins at concentrations above 25 μM and up to 100 μM . For instance, the last entries of decorated microtubules in the EMDB for which the concentrations are stated in the methods are:

- DOI 10.1038/s41467-023-41615-w: using a 75 μM solution of a 39 kDa Kinesin motor domain.
- DOI 10.1038/s41467-024-46260-5: using a 25 μM solution of a 14 kDa protein.

In contrast, Elp123 is a ~600 kDa complex for which the maximum concentration we ever achieved is ~2 μM . Therefore, even by using conditions favouring binding and decoration (i.e., using GMPCPP-stabilized microtubules to promote lattice-binding, or by repeatedly applying sample protein to pre-adsorbed microtubules) we could not observe unambiguous signal on microtubules, in single particle or in tomography using a tilt series. This could also be explained by the fact that Elongator is not only large, it is also very flat and long (~250Å) compared to a Kinesin motor domain (~50Å), spanning several tubulin repeats and thus making the 2D/3D alignment non trivial.

We therefore thought of using monomeric mutants, as suggested as well by the reviewer. We attempted to clone and purify several mutants, including a Familial Dysautonomia truncation of Elp123 predicted to be monomeric (Drosophila residues 1-656 together with the extra residues resulting from exon skipping), a smaller C-terminal truncation predicted to be monomeric (res. 1-872), as well as a point mutant equivalent to the one described in Xu et al. 2015 (R940A in Drosophila Elp123). Sadly, in all cases, we were unable to purify a stable Elp123 complex, and the three subunits eluted separately after gel filtration.

So while we agree with the reviewer that a structural characterisation of Elp123 bound to microtubules would bring new insights into how Elongator recognises and binds microtubules,

despite our multiple attempts we were not able to obtain EM grids allowing us to do so for technical reasons.

However, it must be emphasized that while it is true that we lack an understanding of the Elp123-Microtubule interaction from a structural point of view, we actually understand quite well this interaction from a functional point of view. In other words, we understand what functional features of the microtubules Elp123 recognises, specifically:

- 1) Elongator recognises the nucleotide state of the microtubule, as it binds specifically to GTP or GTP-like microtubules (GMPCPP and GDP·BeF₃), whilst is absent in the GDP lattice, as can be seen in Fig. 1, which now has an additional panel (Fig. 1C) to highlight this point.

- 2) Elongator detaches when microtubules undergo catastrophe (Fig. 1E). In other words, as the microtubules lose their GTP portion, which Elongator specifically recognises, it detaches.

- 3) Elongator seems to accumulate at the transition between the GMPCPP lattice and the GDP lattice (Fig. 1E, new Sup. Fig. S1B). This is a feature seen in end-binding (EB) proteins, and this was hypothesized to be caused by lattice discontinuities between the two lattices (as they likely contain different number of protofilaments), defects and holes at the transition point (Reid *et al.* eLife 2019).

Together, these observations suggest that Elongator uses a way of binding very similar to EB proteins, although not identical (i.e., contrary to Elongator, EBs bind preferentially to GDP·BeF₃ microtubules over GMPCPP), that it uses to track the growing ends of microtubules. This is perfectly in line with the model we propose, and we believe we have enough independent observations to convince the reviewer and the reader of the functional relevance of the mechanism we propose.

RI.5. The interaction between Elp123 and Elp456 seems to be highly dynamic – the authors use a mutation that decreases the affinity slightly, but the affinity is still very high (~67 nM). It is also not clear if the conformation (and the mentioned interfaces) that was recently imaged in the fully assembled complex is relevant for the observed activity. It might well be that the used “solo” mutant should not have any influence on the described activity. I think the authors overinterpret their data.

*Related to Figure 4 – if higher excess of Elp456 is added to Elp123 – would the authors see an additional stabilization and increase in lifetime of the microtubules – both for wildtype Elp456 and the mutant? I would simply expect that the decreased affinity of the mutant should be compensated with increasing concentrations of Elp456 (or Elp123), which would clearly show that the intermediate effect is indeed caused by the lowered affinity. Related to this - a previous report shows that the N-terminus of Elp4 is crucial for the interaction between Elp1 and Elp456 (Dauden *et al.* 2017) – did the authors test, if this region is crucial for the observed effect? The N-terminus would provide sufficient flexibility and at the same time keep the subcomplexes attached to each other – in this case the mapped interface would not be relevant.*

There are two points here, the characterization of the Elp456 “solo” and N-terminus of Elp4, which we will discuss separately for clarity in the following.

A) Characterization of the Elp456 “solo” mutant in relation to our hypothesis

The reviewer is totally right that the affinity of the “solo” mutant is still high. But because this affinity is within the range of concentration that we use for our *in vitro* experiments, we would argue that it

actually provides even a *better* understanding of the mechanism compared to a complete null mutant if one performs the dose-dependence experiment suggested by the reviewer.

Indeed, if we were to make a mutant that abolishes the binding between Elp456 and Elp123, and do not see any effect on microtubule dynamics, we cannot determine whether it is because the coupling between Elp456 and Elp123 is needed (our hypothesis), or if something else in the mutant is wrong because we could never *rescue* this mutant. But because the affinity of Elp456 *solo* is lowered only by a factor 4.5, and that affinity is in the range of the concentrations we use, we can experimentally address if adding more of the Elp456 *solo* in the experiment “rescues” the effect (i.e., increase of the speed of growth and lifetime) by compensating the decrease of affinity. If successful, this experiment would demonstrate that it’s indeed because the affinity of Elp456 *solo* for Elp123 is lower that the effect is lower, and not anything else.

Prompted by this comment, we have now characterised the effects of the Elp456 “solo” mutant on microtubule dynamics across an extended range of concentrations (new Fig. 5C, D):

1) At 25 nM Elp456, which is above the calculated K_d for the *wild type* (~16 nM) but below that of the “solo” mutant (~67 nM), *wild type* Elp456 already displays a significant effect on microtubule dynamics, whilst the mutant does not.

2) At 50 nM Elp456, the effect is even higher for the *wild type* protein, whilst the “solo” mutant still does not have any measurable effect on microtubule dynamics.

3) At 100 nM and 200 nM, however, both *wild type* and mutant have a significant effect on microtubule dynamics, with the effect being more pronounced at higher concentrations. Yet, the effect of the *wild type* protein is still higher than that of the mutant, consistent with their relative calculated affinities.

Therefore, our data shows that, indeed, weakening the interaction between Elp456 and Elp123, without affecting the binding to tubulin (Sup. Fig. S7G), decreases the effect of Elongator on microtubules, and that we can compensate this by adding more Elp456 *solo* to push the equilibrium towards Elp123 binding. This is in perfect support of our model, and we warmly thank the reviewer for their constructive comment and key experimental suggestion, which we think strengthened the paper.

We have added the following text to the manuscript describing the effect of the “solo” mutant:

“We then verified this prediction by using the Elp456 *solo* mutant in microtubule dynamics assays (Fig. 5C-E) and a range of concentrations, including data points below and above the calculated K_d . Strikingly, the reconstituted Elongator containing Elp456 *solo* had an intermediate effect between the buffer alone and reconstituted Elongator containing *wild type* Elp456, both for the microtubule growth speed (Fig. 5C) and the lifetime (Fig. 5D, E) of microtubules. We observed a dose-dependent effect when increasing amounts of *wild type* or *solo* Elp456 were added (Fig. 4B-D, Fig 5C-E). At concentrations below the K_d , the *solo* mutant, together with Elp123, does not display any effect on the microtubules. At higher concentrations, however, the weaker affinity is compensated and the *solo* mutant increases both the speed of growth (Fig. 5C) and the lifetime (Fig. 5D, E) of microtubules. Importantly, the effect of the *solo* mutant is always weaker than that observed for wild-type Elp456, although this difference is smaller at very high concentrations (i.e. 200 nM), that is, greatly above the calculated K_d . Similar results were observed for the minus ends (Sup. Fig. S3C, D). These intermediate, rather than total, effects are consistent with the relative affinity of Elp456 *solo* towards Elp123 (Fig. 5B).”

Similarly, we have added a paragraph in the discussion addressing the question of the reviewer regarding the effect of a higher concentration of Elp456 on microtubule dynamics:

“It is worth noting that we detected a higher effect on microtubule stability when using an excess of Elp456 over Elp123 to reconstitute Elongator (Fig. 4B-D, Fig. 5C-D, Sup. Fig. S3A-D). However, when using the full Elongator complex purified from *Drosophila* cells, no dose dependence was observed³¹. Since the purified full Elongator complex has a defined Elp123:Elp456 ratio, this hints at a mechanism in which Elp456 is the limiting factor. One plausible hypothesis is that Elp456 acts as a “shuttle” to bring tubulin dimers to Elp123, located at the growing end of microtubules (Fig. 5A). In this scenario, a higher availability of Elp456 would bring more tubulin to the microtubules, further increasing their speed of growth and lifetime. Additionally, it has been described that Elongator can recruit an extra Elp456 in low-salt conditions or when additional Elp456 is added^{28,29}. Together, our data suggests a mechanism in which the effect of Elongator on microtubules can be fine-tuned by controlling the amount of available Elp456, providing cells with an additional way to control the effect that the Elongator complex has on microtubules.”

B) Characterization of the N-terminus of Elp4

We thank the reviewer for the suggestion regarding the N-terminus of Elp4. Indeed, in our draft we explored the idea that Elp456 uses a flexible region in the complex to bind to the flexible tubulin C-terminal tails (with the ΔpolyN mutant). We reasoned, as this reviewer, that such a binding mode would help explain the difficulty to align the particles from our CryoEM dataset. However, the 10 N-terminal residues shown in Dauden *et al.* to be required for the interaction with Elp1 are completely absent in the *Drosophila* Elp4 (see alignment below), so we cannot perform the experiment suggested.

Yeast	MSFRKRGEILNDRGSGLRGPLLRRGPPRTSSTPLRTGNRRAPGNVPLSDTTARLKKLNIA	60
Fly	-----M	1
Yeast	ESKTKMGLDSSHVGVVRPSPATSQPTTSTGSADLDSILGHMGLPLGNSVLVEEQSTTEFH	120
Fly	TSFRKRTVQKPIRGTRTSPHTAQVITSSGNPYLDVVIG-GGLPMGSICLIEEDRFMTHAK	60
	* * :. . *.* ** *:* **:* ** ::* ***:* *:** . .	

RI.6. *The interpretation of the co-migration and co-IP experiments are also not free of bias – Elp456 can also be seen in the control experiment, showing a significant background signal (Fig S3C). This experiment should also be performed while pulling on Elp456, like for Elp123 (Fig S3D). The co-migration assay needs to be complemented by WB blotting. As I don’t exactly know, which fractions are used for the detection, it is hard for me to give a clear judgment. At the moment, I don’t have enough information to judge if a clear shift of the complex is observed.*

We thank the reviewer for this constructive comment. We have experimentally addressed all these concerns as follows:

- We have repeated the PC-tubulin co-IP experiment in Fig. S3C and quantified the intensity of the Elp5 and Elp6 bands (new Fig. S3D). As the experiment is an SDS-PAGE gel stained with colloidal Coomassie blue, the bands intensities directly correlate with the amount of protein (unlike a western blot). This analysis confirmed that indeed, Elp5 and 6 (and presumably Elp4) is enriched in the elution when immunoprecipitating tubulin compared to the control.
- We tried to perform the same experiment pulling on Elp456 instead using (His)₆-tagged Elp456 and NiNTA resin. However, in the control, tubulin binds non-specifically to the resin

(tubulin is known to be a very sticky protein) and elutes from the beads (see image below). Therefore, we cannot perform this experiment for technical reasons. We have tried with different resins and buffers with similar results.

- We have repeated the co-migration assay and complemented it by western blotting as suggested by the reviewer (new Figure 3B, C). We have also labelled the fractions for easy identification. It is clear now that Elp456 peaks at fractions 6-8 when injected alone, but at fraction 2-4 when in complex with tubulin. Similarly, tubulin migrates alone at fractions 8-12 but has an extended profile when co-injected with Elp456.

We hope that these new experimental characterisations will clear any remaining doubts from this reviewer about the Elp456/ tubulin interactions.

Minor

R1.7. *Could it be that Elongator is recognizing the tip and Elp123 is left behind after the tip is elongated?*

This is an interesting idea. Since Elongator is not an active motor, there are mainly two methods that it could use to achieve tip tracking: it either unbinds when the GTP is hydrolysed to GDP and rebinds again at the tip (as proposed for EBs, for example Roostalu *et al.* eLife 2020) or it diffuses on the GTP part of the tip, therefore “catching up” with the newly incorporated tubulin without detaching (as proposed for XMAP215, Brouhard *et al.* Cell 2008). As we observe Elongator diffusing on GMPCPP seeds, and even “jumping” from GMPCPP seeds to tip-tracking (new Sup. Fig. S1C), we would tend to favour the second solution, namely that Elongator achieves tip-tracking by diffusing on the GTP part of the tip in a way similar to XMAP215.

We have highlighted this point in the Discussion (line 530):

“Presumably due to this, Elongator seems to have a distinctive mode of binding and recognising the microtubule ends, combining features of both EB proteins, like nucleotide-dependent binding (Fig. 1); and TOG-containing proteins, such as lattice-diffusion and dual microtubule- and tubulin- binding (Sup. Fig. S1C, Fig. 3, 4).”.

RI.8. *As the authors how that the fly-specific region in Elp4 has no influence on the observed effect, I was wondering whether the authors would observe a similar effect using yeast, mouse or human Elp456 – these complexes have been produced/purified and would provide a simply analyses to check whether the observed effect is fly-specific or conserved among other organisms.*

We thank the reviewer for this excellent idea! We agree that demonstrating the evolutionary conservation of our findings was important to increase the scope of our work (and our previous study).

We thus first purified human Elp456 from bacteria, using the same method we developed to purify *Drosophila* Elp456 (new Sup. Fig. S9D). When used with *Drosophila* Elp123 purified from insect cells, we observed that human Elp456 also increases the speed of growth of microtubules (new Sup. Fig. S9E) and their lifetime (new Sup. Fig. S9F, G), to the same extent as its *Drosophila* counterpart. This establishes that human Elp456 can replace fly Elp456 and thus that the effect of Elongator onto microtubules *in vitro* are not fly specific.

Given these positive results, we thought that re-cloning expressing and purifying human Elp123 in order to have a purely human complex would not provide significant novel insight, thus we rather focused on demonstrating that, *in vivo*, the effect of Elongator on microtubules was conserved between fly and mammals. While this was not specifically requested by this reviewer, we believe it would address the question even better than the *in vitro* experiments. Importantly, we now show that Elongator controls microtubule poly-glutamylation not only in *Drosophila* cultured S2 cells and Sensory Organ Precursors in developing flies, but also in mouse fibroblasts (see new Fig. 6E-G). As the control of microtubule poly-glutamylation directly stems from the molecular mechanism we propose for how Elongator controls microtubule dynamics, we think this confirms that effects of Elongator on microtubules is not *Drosophila* specific.

We thank the reviewer for their comment, and we hope they agree that the new results significantly expand the scope of our work. We have added a paragraph presenting these new results in the Results:

In vitro, line 391:

“Since this region is the major divergence observed between *Drosophila* and human Elp456, we then decided to verify that the effect observed on microtubules is not *Drosophila* specific. For this, we purified recombinant human Elp456 (Sup. Fig. S10D, right lane) and used it in our reconstitution assays together with *Drosophila* Elp123. Strikingly, the reconstituted fly Elp123 + human Elp456 also recapitulates the effect measured with fly Elp456 both for the microtubule growth speed (Sup. Fig. S10E, Sup. Fig. S3E) and lifetime (Sup. Fig. S10F, G, Sup. Fig. S3F), confirming that microtubule stabilization is not exclusive of the *Drosophila* Elongator complex.”

In vivo, line 484:

“Finally, we confirmed that Elongator controls the poly-glutamylation levels of microtubules *in vivo*. Indeed, Elongator depletion results in the reduction of poly-glutamylation levels at the mitotic spindle in various cultured cells of different species (*Drosophila* S2, mouse NIH/3T3 fibroblasts), but also in

Drosophila Sensory Organ Precursor (SOP) cells in developing flies (Fig. 6E-G, Sup. Fig. S9H, I). Interestingly, the levels of poly-glutamylation of the stable interphase microtubules are not affected by the depletion (Fig. 6E), suggesting that Elongator's activity is more relevant in highly dynamic microtubule structures (the turnover rate of spindle microtubules is ~9 times higher than that of interphase microtubules⁵⁹). Taken together, our data confirm that the activity of the Elongator complex on microtubules regulates their poly-glutamylation levels, serving as a mechanism for the *in vivo* regulation of microtubule poly-glutamylation at the mitotic spindle.”

R1.9. *The findings related to poly-glutamylated monomers (Fig. 6) are very intriguing and highly interesting. However, I was wondering whether the authors have any hint, how Elongator would release the tubulin dimers during deposition at the tip? Do the authors have any indication that Elp456 can sense the GTP state of tubulin or even trigger the hydrolysis? Would the observed effects be affected by GTP or ATP?*

We thank the reviewer for their suggestions, which we have experimentally addressed.

As we do not have a structure of Elp456 bound to tubulin, and do not see technically how we could determine one, it is difficult to unambiguously state how tubulin is released. Our XL-MS data (see answer to comment **R1.2**) roughly identified the surface involved in tubulin binding. This region is in close proximity to the surface used for Elp456 to bind to Elp123, but there is not a clear overlap between both surfaces. As we show that Elp123, Elp456 and tubulin can form a co-complex on microtubules (new Fig. 3A, 4A, Sup. Fig. 4), the release cannot simply be explained by Elp456 releasing tubulin when binding to Elp123 because of steric incompatibility. More structural work, possibly involving the full Elongator+tubulin complex on microtubules will be needed in order to fully understand the release mechanism. But we think this is a project in itself and beyond the scope of the current paper.

We do not think that Elp456 can sense the GTP state of tubulin or affect the hydrolysis rate. Rather, we now show that Elongator acts by decreasing the apparent critical concentration of tubulin for microtubule elongation (new Sup. Fig. S6C), supporting our model. This comment echoes that of reviewer #3, and we kindly orient the reviewer to read our reply to their point **R3.3**.

Finally, we have tested the effect of ATP on microtubule dynamics on Elongator and we cannot detect any qualitative difference with the addition of the nucleotide, neither for tracking nor microtubule growth (see reviewer only figure 1). This is in line with the fact that we could not see any effect of MgATP on the affinity of the interaction between Elp456 and Tubulin (see new Sup. Fig. S5E) and our answer to comment **R1.2** above).

R1.10. *Maybe I don't fully grasp Fig 4A, but I would have expected to see the described enrichment at the growing tips – what I see is a rather disperse appearance of Elp123, Elp456 and tubulin throughout the body of the microtubules. Could the authors describe/illustrate this figure more clearly.*

We apologise for the confusion. Indeed, we forgot to label the AMCA microtubules in Fig. 4A as “AMCA GMPCPP microtubules”. As GMPCPP microtubules mimic a GTP lattice, Elp123, Elp456 and tubulin are expected to decorate the whole lattice of the microtubules, as in Fig. 3A. As discussed in the paper, Elongator seems to recognise the GTP conformation of the microtubule lattice, and is

thus expected to bind throughout the length of GMPCPP microtubules (Figs 3A,4A), but only to the tip of *dynamic* microtubules, which is where the GTP cap is (like in Fig.1).

This labelling error has been corrected in the updated manuscript, and we warmly thank the reviewer for point it out!

Reviewer #2 (Remarks to the Author):

*The regulation of microtubule dynamic polymerization is crucial for various forms of cell division. Regulators of microtubules play important roles in modulating tubulin polymerization at the ends of microtubules or regulating microtubule polymerization rates or phases. Well-characterized classes of microtubule regulators have been identified which bind and track along the plus ends of microtubules. These include the EB1-3 family, which bind the GTP-tubulin cap at microtubule plus ends, and the XMAP215/chTOG proteins, which processively track polymerizing microtubule plus ends and recruit tubulin dimers at the tip of a growing microtubule. These proteins have unique effects and bind at unique sites at MT plus ends. Recent studies from the Derivery lab have suggested that the Elongator complex plays a role in regulating asymmetric mitotic division in *Drosophila* and may directly form a unique new type of microtubule regulator. In this manuscript, Derivery and colleagues utilize a combination of in vitro reconstitution methods to show that the Elongator complex binds growing microtubule plus ends localizing with GTP caps forming a new type of microtubule polymerase. The elongator complex binds soluble tubulin via Elp456 subunits while binding microtubules via Elp123 subunits. The microtubule polymerase activity depends on the concentration of the Elp456 component but requires the Elp123 microtubule-binding activity. Uncoupling the Elp456 from Elp123 leads to defects in the polymerization process suggesting their interactions to form a holo elongator complex is crucial for the process of polymerase activity. Elongator microtubule polymerase activity depends on tubulin modified by glutamylation activity and their absence leads to defects in the newly described activities suggesting the elongator complex prefers polyglutamylated tubulin.*

I don't believe the studies that are presented by the author (in conjunction with their previous paper) demonstrate Elongator functioning directly on regulating microtubule plus end polymerization as a bonafide microtubule regulatory protein. This work is not suitable for publication unless the authors make a serious effort to prove the microtubule polymerase activity persists in near physiological or in vivo settings (see below). Despite the previous in vivo study from this group on the impact of the Elongator complex on asymmetric microtubule spindle cell divisions, there is insufficient evidence about the direct physiological relevance of the Elongator function on microtubules. This study raises serious concerns about the Elongator activities presented here being an artifact of in vitro studies. There is little biological evidence in vivo supporting the Elongator being truly a microtubule polymerase localizing to microtubule ends. Regarding this work, there is a very real concern that the results presented represent in vitro artifacts of the Elongator complex activities binding purified tubulin or GTP-microtubules. As you will see below, the activities make little detailed sense with known and well-characterized microtubule regulators. Specifically, I am very concerned that the Elongator has active sites that bind acidic tRNAs, these pockets may bind negatively charged tubulin surfaces and particularly its acidic c-terminus which can become more acidic upon polyglutamylation. There is a complete lack of real biological evidence for the microtubule plus end tracking and how the Elongator would track microtubules at physiological conditions in the presence of tRNA, end binding and chTOG/XMAP215 proteins (see below). The spindle defects observed in the authors' previous paper could be completely an indirect effect of Elongator inactivation on translation/transcription.

The most crucial missing evidence is that the authors must demonstrate the Elongator tracks microtubule plus ends in vivo in cells. At the very least the authors must demonstrate Elongator activity persists in semi-ex vivo reconstitutions with nearly full cytoplasm containing tRNA and or the full complement of microtubule regulators and at physiological conditions. Such evidence will be critical to validate directly the biological significance of their findings as all microtubule regulators

that have been so far studied in this well-established field, have a direct correlation between the observations made *in vitro* and *in vivo*.

This referee is very explicit about that she/he does “not believe” in our current and past *in vitro* work showing an impact of Elongator on microtubule dynamics. We would like to bring the argument back to a factual basis and demonstrate that the reviewer's concerns can be addressed, and how we addressed them.

Demonstrating that the role of Elongator on microtubule dynamics *in vivo* was independent from its established role in translation was obviously the main point we were required to achieve for our previous paper (Planelles-Herrero, V.J., et al. Nat Cell Biol 24, 1606–1616 (2022)). To recapitulate our previous study, we provided three orthogonal pieces of evidence that the effects of Elongator we measured *in vivo* cannot be due an indirect effect coming from Elongator's role on translation, namely:

1) **Abolishing the tRNA-modifying activity of Elongator does not affect its ability to modulate microtubule dynamics *in vivo*.** Indeed, we showed that the Elp3 YY531AA mutant, which abolishes Elongator's effects on translation by abolishing the only known biochemical activity of Elongator towards tRNA, does quantitatively rescues the Elongator phenotype on central spindle asymmetry *in vivo* in flies. It also rescues the physiological function of this spindle asymmetry, namely the asymmetric segregation of signalling endosomes containing fate determinants (Fig 4 in Planelles-Herrero, 2022). We further demonstrated that this mutant has no effects on the ability of Elongator to affect microtubule dynamics *in vitro* (Fig 5 in Planelles-Herrero, 2022). We would consider these quantitative genetic and biochemical observations as direct disprove of the reviewer's hypothesis that Elongator's function on the spindle stems from its role in modulating translation of an unknown microtubule regulator: how could Elongator's effects be due to translation if a translation-incompetent Elongator retains said effects?

2) **Downstream perturbation of the Elongator-mediated tRNA-modifying pathway has no impact on central spindle asymmetry.** Indeed, CTU1, which is well established in the Elongator field to be essentially required for Elongator's effects on translation (Hawer et al. Genes, 2019 Björk et al. RNA 2007), does not have any effect on central spindle asymmetry or asymmetric segregation of signalling endosomes *in vivo* in flies (Fig 4 in Planelles-Herrero, 2022). Again, these observations exclude a scenario where Elongator would regulate spindle asymmetry by modulating the translation of some (unknown) microtubule regulator.

3) **Elongator sub-cellular localization controls microtubule polarization.** By capitalizing on a nanobody-based assay we previously developed, we showed that the sub-cellular distribution of Elongator controls its effects on reshaping the microtubule landscape in flies (Fig 6 in Planelles-Herrero, 2022). In other words, the asymmetry of Elongator controls the asymmetry of microtubules (not the reverse). Given that these cells (Sensory Organ Precursor or SOP cells) are tiny (10 μm in diameter), it is very hard to envision how this result could be explained by some kind of localized translation, which would be necessary to invoke to fit the reviewer's hypothesis that Elongator's function on the spindle stems from its role in modulating translation (plus, translation is supposed to be repressed in mitosis anyway).

This combination of genetics, *in vivo* and *in vitro* evidence satisfied all three reviewers of the previous paper (as well as the editor), and we purposely made our rebuttal letter accessible on the NCB website for the community to read our full set of arguments, which we respectfully invite the reviewer to do if they haven't. Importantly, if there were any remaining concern that the effects of Elongator we see could not be direct but had something to do with translation, surely the Elongator

specialist, reviewer #1, would have flagged it, which is not the case. Comments of this referee are tough, but fair and constructive to make the paper better, which we welcome.

Obviously, it is not the point of the current paper to do yet another round of reviews of our previous work, but given the comment of this reviewer, we decided to strengthen our manuscript by adding evidence that the activities that we report *in vitro* in the current manuscript also occurs *in vivo*. While this was not directly requested by the reviewers, we thought it was important to provide this novel work to clear out any remaining doubts that our observations are not physiologically relevant.

In particular, we now show that the novel microtubule activity we report *in vitro* for Elongator, namely to control the levels of poly-glutamylation of growing microtubules, also occurs *in vivo* at the spindle. Importantly, we showed this not only in cultured cells from different species (*Drosophila* S2, mouse 3T3 fibroblasts), but also in an actual organism, namely in Sensory Organ Precursors in flies (see new Fig. 6E-G). In other words, not only the biochemical activity we report for Elongator *in vitro* is **physiologically relevant**, but it is also **evolutionary conserved**. In addition, we also followed the excellent suggestion of this reviewer to show that “*Elongator activities remain in semi ex-vivo reconstitution*”, and we now also show that Elp123 binding to microtubules, as well as Elp456 binding to tubulin also occur in cytosolic extracts in the presence of physiological amounts of all Elongator binding partners (not just tRNA). So again, as in our previous work, we have remarkable consistency between our *in vitro* and *in vivo* findings (see also our answer to points **R1.2** of reviewer #1 and **R2.1** below).

We sincerely hope that the combination of the new data we provide here, in conjunction with the genetics and quantitative cell biology data we provided in our previous paper will convince this highly sceptical reviewer. Obviously, we must always be prudent as by definition, we cannot formally prove a hypothesis, only disprove it, but at the current state of understanding, we don't have a single piece of evidence that suggests that our results could come from an indirect effect of Elongator on translation, as all our experiments points to the contrary. Furthermore, we were happy to see that this reviewer shares our vision of biology that the best way to validate the biological significance of one's findings is to observe a “*direct correlation between the observations made in vitro and in vivo*”. As detailed above and below, we would argue that we provided this exact level of demonstration for our claims about Elongator's role in microtubule dynamics, where all our *in vitro* data is coherent with our *in vivo* data, in multiple organisms.

I have made some detailed concerns and suggestions that would help the authors experimentally address this work:

R2.1. *The authors carried out all the in vitro reconstitution studies in the absence of the native Elongator substrate -- tRNA. What would be the impact of the presence of physiological concentrations of tRNA in these assays on the microtubule polymerase activity?*

This was also a concern of Reviewer #1. We kindly invite the reviewer to read our answer to point **R1.2**, which we won't duplicate here for brevity.

In a nutshell, we found that none of the *in vitro* activities of Elongator were qualitatively or quantitatively affected by a physiological amount of tRNA (100 μ M), which represents a huge excess compared to both Elp123 (25 nM) and tubulin (16 μ M) in these assays. Furthermore, adopting the “semi ex-vivo” approach suggested by this reviewer above, we also demonstrated that when mixing microtubules with cytosolic extracts expressing fluorescently labelled Elongator at endogenous

levels, clear Elongator signal can be detected on microtubules. This demonstrates that binding of Elongator to microtubules is strong enough, and specific enough, not to be outcompeted by all the tRNA or any microtubule associated proteins present in the extract. We warmly thank both reviewers for this constructive comment and excellent suggestions, which we think strengthened the paper.

R2.2. *The group's previous paper suggests microtubule lattice localization in vivo. However, microtubule plus end localization was not observed. All well-studied microtubule regulators track growing microtubule ends in vivo, and if the observed in vitro features are true, they should be observable in vivo.*

We do not see how our previous study is at odds with our current results, as implied by this reviewer. The core of the issue seems that we previously reported a more or less homogenous staining of Elongator at the spindle *in vivo* (Planelles-Herrero et al., 2022), while here, we report that Elongator can track the tips of microtubules *in vitro*.

Classically, to assess tip tracking, one would make a movie of the protein of interest and see if diffraction-limited spots follow a straight trajectory (or better, simultaneously image a microtubule marker). This is in fact what we did to assess this property of elongator *in vitro* in our current manuscript. Now, our previous paper described the localization of Elongator in **fixed** tissues, namely Sensory Organ Precursor (SOP) cells in the *Drosophila* notum. We obviously attempted to perform live cell imaging of Elongator during that project, but current Elongator live-imaging tools, and microscopes for deep-tissue imaging are just not good enough yet. This is because Elongator is abundant in the cytosol, and thus it is not trivial to distinguish between the free pool and the microtubule-bound pool (it's very akin to imaging endogenous dynein on microtubules in neurons, which yields a homogenous blur unless HiLo illumination is used, as recently demonstrated by the Carter lab, but HiLo is obviously not possible in tissues). We think that the reason why we were able to image Elongator on the spindle is because of the massive signal amplification of an immunofluorescence which compounded to the high density of microtubule ends, both plus and minus, and so of Elongator, at the spindle.

Now, even if fixed microtubules obviously maintain their ends, we do not think it is trivial to assess tip localization in fixed samples at the spindle in our samples. Indeed, the mitotic spindle is not composed of long, continuous microtubules going from the centrosome to the mitotic plane, but rather a constellation of small microtubules (see all the experimental and theoretical work from the Bruges lab, and the EM reconstructions of the and Müller-Reichert lab, for instance). So there are many microtubule ends within the spindle, which, when convolved with the PSF of the microscope and the fact that SOP cells *in vivo* are much smaller than traditional mammalian cells in culture, likely makes it very hard to resolve individual ends. In fact, when we looked in the past at Patronin *in vivo*, a well-established minus-end marker, in SOP cells, we saw a more or less homogenous signal, even in live, and had to rely on automated movie averaging to reveal the minus end localization of Patronin (Derivery et al. 2015). Additionally, and compounding with this, we would argue that the concern expressed by the referee would be perceivable if Elongator would have a very restricted binding region, like EB proteins do. However, as described in point **R2.4** Elongator has a different binding mode from EBs, and thus could have an extended binding region as compared to EBs. Therefore, under conditions where Elongator is bound to short microtubules, in a short spindle, in the presence of high cytosolic signals, it is not surprising that the decreased resolution of deep-tissue imaging makes the Elongator *in vivo* appear more or less homogenous.

This being said, and following an excellent suggestion of this reviewer to use cell extracts to validate our findings, we now show that Elongator decorates GMPCPP microtubules in extracts expressing endogenous levels of fluorescent Elongator. We hope this experiment addresses the referee's concern, and we believe that it is as close as we can get to cytosolic conditions while allowing high signal-to-noise ratio microscopy (see new Sup. Fig. 5C,D). We introduce these new results, together with all our new controls, in a new section starting at line 245 "Elongator can discriminate between microtubules and tRNA".

R2.3. The well-known microtubule regulators (described below) are generally present in cells at high concentrations and these regulators normally dominate localization at microtubule ends in cells. How does the elongator GTP-dependent Microtubule tracking activity *in vitro* get affected in the presence of EB3 and or chTOG/XMAP215 microtubule polymerases?

This is an interesting point, which we have addressed experimentally.

We purified EB1 and assessed a potential competition between Elongator and EB1 in dynamic microtubules assays *in vitro*, but could not find any evidence of competition. In fact, EB1 and Elongator actually compensate each other. Indeed, we reproduced the well-established result that EB1 reduces the speed of growth of microtubules *in vitro*, but found that when Elongator was added, this effect was rescued (see Reviewer-only Figure 2).

We could have redone the same kind of experiments with other MAPs in our freezer, but fundamentally the key question here is whether Elongator can bind to microtubules in the presence of all the cytosolic MAPs in their physiological relative concentrations. So we thought to follow this reviewer's excellent suggestion of capitalizing on a "semi *ex vivo* reconstitution" and used extracts expressing physiological concentrations of fluorescent Elongator to assess if Elongator could still decorate microtubules within this complex mixture of dozen of MAPs all competing with each other. Indeed, we found that it does (see new Sup. Fig. S5C-D). This key result implies that even if it is conceivable that there are some MAPs that have the capacity to compete with Elongator for the same binding site, at least in the conditions of the cytosol, Elongator wins (or at least does not lose). Note that the stable cell line expressing fluorescent Elongator has been characterized previously as a bona fide source of fluorescent complex (Planelles-Herrero et al., 2022), and that the expression of Elp3 is titrated by the other subunits and so it is close to endogenous levels (i.e., when purifying "overexpressed" Elp3 via an affinity tag, there is no excess of Elp3 compared to the other subunits, suggesting that the expression of Elp3 is reduced to match that of the other subunits).

Furthermore, we now show that the novel activity of Elongator we originally described in the paper (namely that it can tune the level of poly-glutamylation in newly polymerising microtubules *in vitro*) is physiologically relevant, as we now show that Elongator depletion lowers polyglutamylation levels at the mitotic spindle - not only in fly cells and *in vivo* in actual flies during asymmetric cell division, but also in mouse fibroblasts (see new Fig. 6E-G). This confirms that Elongator binding for microtubules is strong enough - and specific enough - to have a measurable effect on microtubules *in vivo*, even in the presence of all the other MAPs and tRNA. Note that this also means that this function of Elongator is not redundantly covered by any other MAP.

We have added an entire new section in our manuscript detailing these results "Elongator can discriminate between microtubules and tRNA" starting in line 245 and the *in vivo* results in a new paragraph at line 484.

R2.4. *The features of Elongator as a microtubule polymerase do not make logical sense within the framework of well-studied microtubule regulators. The chTOG/XMAP215 Microtubule polymerases generally recruit soluble tubulin to the tip of a growing microtubule end by binding soluble tubulin via their TOG domains and interacting with the extreme ends of polymerizing microtubules. Their activities are not dependent on the GTP-nucleotide state of microtubule ends, and they still bind the plus ends of GMPCPP microtubules. End-binding plus tracking proteins, such as EB1 or EB3, only bind newly polymerized microtubules composed of GTP tubulin (GTP-cap) and they do not bind soluble tubulin. EB1 and EB3 dissociate from plus ends upon microtubule lattice transition from GTP to GDP and they fully decorate GTPgS microtubules. Both above regulators localize at unique sites at different zones of polymerizing microtubule ends. The Elongator seems to bind soluble tubulin and uniquely bind the GTP lattice of the microtubule. Following the rationale of known regulators, these two features are not compatible with being a microtubule polymerase. Based on the above properties of the EB and chTOG/XMAP215 proteins, the activity of the Elongator makes little sense as a microtubule polymerase, since Elongator localization will likely be treadmilling, assuming their binding is similar to EB3, suggesting it will not likely physically track with microtubule ends as chTOG/XMAP215. This probably explains the very modest microtubule polymerase activity and is consistent with the very low effect on polymerization rates of plus ends described by the group's previous paper, compared to the ten-fold increase mediated by chTOG/XMAP215. Spiking experiments should be performed if the authors would like to explain the plus-end GTP-dependent tracking of the Elongator in relation to the above well-characterized regulators.*

We agree with the reviewer that Elongator has some **distinct** properties when compared to other microtubule polymerases and/or end-binding proteins. However, we strongly disagree that this implies that Elongator's features "*do not make logical sense*" with other microtubule regulators.

Indeed, as perfectly summarized by the reviewer, Elongator shares some similarities with EB proteins (also summarized in response to the point **R1.4** by reviewer 1):

1) Elongator recognises the nucleotide state of the microtubule, as it binds specifically to GTP or GTP-like microtubules (GMPCPP and GDP·BeF₃), whilst is absent in the GDP lattice, as seen in Fig. 1, which now includes a new panel (1C) making this point clearer.

2) Elongator detaches when microtubules undergo catastrophe (new Fig. 1E). In other words, as soon as microtubules lose their GTP cap, which Elongator specifically recognises, it detaches.

3) Elongator seems to accumulate at the transition between the GMPCPP lattice and the GDP lattice (new Fig. 1E, new Sup. Fig. S1B). As discussed in original manuscript, this is a feature seen in other end-binding proteins, such as EBs, and this was hypothesized to be caused by discontinuities between the two lattices (as they likely contain different number of protofilaments), defects and/or holes at the transition point (Reid *et al.* eLife 2019).

Elongator, however, binds with the same apparent affinity to GDP·BeF₃ and GMPCPP microtubules (Fig. 1F), whilst EBs prefer GDP·BeF₃ over GMPCPP. And, importantly, EB proteins do not stabilize microtubules on their own - on the contrary to Elongator, and decrease the microtubule growth rate, while Elongator increases it (see Reviewer-only Figure 2).

To consider the differences between EBs and Elongator, one must consider not only the nucleotide states of tubulin in the filament, but also the geometry of the microtubule, including lattice conformation, expansion and twist. Microtubules are proposed to undergo several GTP lattice transitions, involving different conformations, before forming a mature GDP lattice. Binding by EB proteins is not simply nucleotide dependent, as they preferentially bind to negatively twisted regions,

possibly transitioning from an expanded lattice to a compacted one: this is the "EB comet" region (see, for example, Estévez-Gallego *et al.* eLife 2020, Roostalu *et al.* eLife 2020). Elongator's binding domain, corresponding to the full E1p123, is ~270Å in size, an order of magnitude bigger than EB's (~30Å). It is therefore imperative to consider the geometry of the microtubule, as likely Elongator's binding to the lattice comprises several tubulin subunits and/or protofilaments (i.e., binding along a protofilament vs across them). Elongator binding might also be dependent of the seam structure, which has been shown to be different in different microtubule states (LaFrance *et al.* PNAS 2022).

On the other hand, chTOG/XMAP215 does not "read" the nucleotide state of the microtubule. Rather, it binds specifically to the exposed end of the microtubule (the "tip's tip", Maurer *et al.* Curr Biol. 2014), which is a dramatically different mechanism compared to Elongator. chTOG/XMAP215 binds to both microtubules and tubulin dimers, stabilizing microtubules in a similar manner as Elongator, although chTOG/XMAP215 seems way more efficient.

Therefore, Elongator seems to combine characteristics of both EB and chTOG/XMAP215, whilst at the same time being different to both protein families. But we do not see how this makes Elongator "not logical" as a tubulin polymerase. Elongator is indeed not as efficient as chTOG/XMAP215, but this does not make it **wrong** (or *illogical*), it is just **different**. There are indeed several reasons why Elongator would be a less efficient polymerase than chTOG/XMAP215, namely:

- 1) chTOG/XMAP215 binds and *primes* the tubulin dimers for incorporation, increasing not only the local concentration but **also** facilitating the incorporation of the dimers in the microtubule. On the other hand, Elongator only "holds" the tubulin at the end, increasing the local concentration (as we show now in new Sup. Fig. S6C). In other words, chTOG/XMAP215 has two compounding effects, on the contrary to Elongator.

- 2) chTOG/XMAP215 binds to the end of the microtubule, whilst Elongator will be "lagging" behind due to its binding mode. This would tend to lower the magnitude Elongator's effects on the local concentration of tubulin at the tip.

- 3) It has been recently reported that poly-glutamylated microtubules have intrinsic lower growth rate and are intrinsically less stable (Chen & Roll-Mecak, MBoC 2023) than non-modified microtubules. Therefore, as Elongator enriches microtubules with poly-glutamylated tubulin, it is fighting against this effect: the microtubules grow faster and are more stable, but at the same time are enriched in poly-glutamylated tubulin, making them slower and less stable. It is important to note that poly-glutamylated microtubules were reported to be **28% slower** and **42% less stable**. Considering this, the effect that Elongator must have on the microtubules, if one deconvolves the intrinsic contribution of the poly-glutamylation, would be much higher. Unfortunately, this is something that cannot be measured experimentally, as microtubules without poly-glutamylation are not significantly affected by Elongator (new Sup. Fig. S8G-J).

But, while an admittedly weaker polymerase, Elongator does something that chTOG/XMAP215 cannot do, namely it modulates the polyglutamylation state of microtubules alongside with their polymerisation, a property we now demonstrate also occurs *in vivo*, in various cells/species. As chTOG/XMAP215 does not bind to the tails of tubulin subunits, it is difficult to imagine how it could change the tubulin code of the microtubules it elongates. It is tempting to speculate that this key property of Elongator allows very dynamic microtubules, such as those of the spindle, to be polyglutamylated.

We hope the reviewer understands our reasoning: yes, it is true Elongator has a smaller effect than chTOG/XMAP215 on microtubule growth speed, but it does make sense within the framework of other microtubule regulators by combining properties observed in different families. After all, as there is no evolutionary conservation between Elongator and EBs/TOG, it is not unrealistic that Elongator would end up converging on properties found in both families. We think the fact that Elongator lowers the critical concentration at the microtubule ends qualifies as a polymerase (albeit weaker). Nevertheless, if the reviewer's concern stems from the use of the word "polymerase", we could envision removing it to favour a "shuttle" model.

We explore these concepts now in the discussion in lines 530 and 549.

R.2.5. *There was little description of microtubule minus-end localization of Elongator even though the complex should have a robust effect at microtubule ends. previous work indicates the minus ends are sites for regulation. The authors should look at microtubule polymerization tracking at regimes where minus polymerization is observed.*

We are a bit puzzled by this comment as the first version of the paper already showed minus-end tracking and loss of signal during catastrophe, as for the plus-end (former Fig 1D, now 1E). This panel was specifically called in the text, and we actually used this minus-end data to highlight that the tip signal is lost when the microtubule is undergoing catastrophe.

L131 [...] strong binding of Elongator to both the plus- (Fig. 1C, D) and minus-ends (Fig. 1E) of growing microtubules could be detected in ~10% of the analysed growth events. The end-localization of Elongator was lost when the microtubule underwent catastrophe (Fig. 1E, Sup. Fig. S1B-C, orange [...])

Similarly, while all data presented in the current paper were systematically quantified at both the plus-ends and the minus ends, for simplicity, we plotted the effects at the plus end in the figures, while referring to minus end data in the text, e.g.

L178 [...] Similarly, we also could not detect any effect of Elp123 at the minus ends (control: $0.319 \pm 0.007 \mu\text{m}/\text{min}$, n=90; Elp123: $0.343 \pm 0.010 \mu\text{m}/\text{min}$, n=138) [...]

L236 [...] A similar effect was observed at the minus-ends, again matching the ~1.2 fold increase we reported for the full Elongator complex⁴ (control: $0.348 \pm 0.008 \mu\text{m}/\text{min}$, n=115; Elp456: $0.329 \pm 0.011 \mu\text{m}/\text{min}$, n=72; Elp123: $0.343 \pm 0.010 \mu\text{m}/\text{min}$, n=138; Elp123+Elp456: $0.408 \pm 0.011 \mu\text{m}/\text{min}$, n=176; Elp123+2xElp456: $0.472 \pm 0.011 \mu\text{m}/\text{min}$, n=133). [...]

We opted for this solution, as we previously established that the two main effects of Elongator on microtubule dynamics, namely increase of microtubule growth rate and decrease of microtubule catastrophe rate, occur at both ends in quantitatively similar fashion (Planelles-Herrero *et al.* Nature Cell Biology 2022). This implies that it does not matter whether we study the plus-end or the minus-end in the current study. Note that this property of Elongator to similarly affect both plus-end and minus-end growth was again confirmed throughout the present manuscript. For instance, if we compute the fold-effect for the data presented in figure 4:

	minus end		plus end	
	speed (μm/min)	fold increase compared to control	speed (μm/min)	fold increase compared to control
control	0.348	1	1.0204	1
elp456	0.329	0.945	0.9811	0.961
Elp123	0.343	0.986	1.05	1.029
Elp123+Elp456	0.408	1.172	1.238	1.213
Elp123+2X Elp456	0.472	1.356	1.364	1.337

Finally, we believe it would make the figures less easy to follow if we add all minus-end information in the panels rather than the text, which is why we choose the plus-ends as they allow for an easier and more reliable quantification. We hope the reviewer agrees. This being said, in our revised version, we added supplementary panels for this (see Sup. Fig. S3), and we added the following sentence to the legends of Figures 2 and 4: **Similar results were observed at the minus end (see text and Sup. Fig. S3).**

And we added the following sentences in the methods:

Note that throughout the paper, we plotted results for microtubule growth rate and lifetime only at the plus end in figure panels for simplicity. This is because since Elongator has similar effects on the dynamics of both the plus and the minus ends (Planelles-Herrero et al., 2022b), plus end dynamics measurements are sufficient to evaluate the effects of Elongator. We nevertheless provide , and in supplementary figure S3.

R.2.6. *Other microtubule regulators such as CLASP family of TOG proteins modulate the dynamic state of microtubules by inhibiting transitions of microtubules from polymerization to depolymerization, termed catastrophes, and mediating the reverse transitions, termed rescues. Effects on dynamic instability transition states seem to be more consistent with the effect observed for Elongators as the authors demonstrated a catastrophe inhibition effect on both plus and minus ends in their previous paper, which was not further analyzed here in this work, but it should be re-addressed in more detail.*

We do not fully understand this point. We previously demonstrated that the effects of Elongator in terms of growth rate and catastrophe rate are similar at both ends (Planelles-Herrero et al., 2022). As detailed above, our original manuscript did already report the effects on minus end growth rate for Elongator, as well as all the different mutants and conditions we considered.

Now, while we did measure them, it is true that we only plotted the effects of Elongator on microtubule catastrophe at the plus end, not the minus end. This is because again it seemed redundant to us for the reason described above in light of our previous work. But we agree that for the sake of completeness, it would be important to also report the effects of Elongator, as well as the mutants we consider, on minus end catastrophe. So, in our revised version, we added supplementary panels for this (see Sup. Fig. S11). As can be seen on these new plots, the decrease on the microtubule catastrophe rate induced by Elongator at the minus ends mimics that of the plus end, as we discussed above in our answer to point **R2.5** for Elongator's effects on the microtubule growth rate.

We hope that this additional data satisfies the need for further analysis of the inhibition of catastrophe by Elongator on both plus and minus ends.

R2.7. *How is the microtubule tracking activity of Elongator impacted in the presence of a physiological level of its native substrate tRNA? Is the microtubule polymerase activity inhibited/or unaffected in the presence of tRNA?*

This is an excellent point also raised by reviewer #1. We invite the reviewer to read our detailed reply to point **R1.2** of reviewer 1 which we won't reproduce here for clarity. In a nutshell, we now provide multiple orthogonal lines of evidence, both *in vivo* and *in vitro* that Elongator activities towards microtubules are totally independent from its tRNA binding activity.

R2.8. *What is the effect of physiological salt on the tubulin binding and microtubule polymerase activity or binding to microtubule plus ends? The conditions used for tubulin binding assays and microtubule regulatory activities are almost entirely devoid of salt (BRB80 conditions). Understanding how the Elongator activity performs in physiological salt may alleviate the serious concerns about this being an artifact of *in vitro* studies.*

We agree with the reviewer that BRB80 buffer is far from being physiological, albeit the gold-standard in the field for microtubule dynamics assays (for example, Chen et al. eLife 2023, Torvi et al. eLife 2022). This was also a concern of reviewer #1 and #3, and our revised version provides several orthogonal lines of evidence that Elongator binds and acts on microtubules in physiological conditions, which are detailed in our answer to point **R1.2**, **R2.1** and **R3.4**. In a nutshell:

- 1) Elp123 binds to microtubules and tracks their growing ends in the presence 50 mM KCl.
- 2) Elp456 binds to tubulin with similar affinity in the presence of 50 mM KCl.
- 3) Elongator still binds microtubules in semi ex-vivo conditions with cell extracts prepared using buffer containing 150 mM KCl.
- 4) Elongator depletion affect spindle poly-glutamylated levels *in vivo*. This makes very unlikely that our *in vitro* data that Elongator changes the poly-glutamylated state of the microtubules it elongates is an *in vitro* artifact from the use of salt-free BRB80.

Note that we used 50 mM KCl rather than a more physiological ~150 mM because it is well established that Elp456 detaches from Elp123 at higher salt concentrations *in vitro* (our own observation, but also reported in Setiawati et al. EMBO Reports 2017). This obviously does not mean that Elp123 and Elp456 do not bind *in vivo* at physiological salt concentrations, as *in vivo* obviously the crowding of the cytosol, and the fact that water is limiting, pushes all binding equilibria towards binding.

R2.9 *Glutamate composition of the tubulin c-termini should be determined for various forms of tubulin, using mass spectrometry, rather than just assuming the source of the tubulin from TTLL knockout mice would be less glutamylated.*

We did not assume anything. Rather, we experimentally characterized the specific source of tubulin that was used in these experiments, which we previously published (Genova et al. EMBOJ 2023). In particular, we experimentally demonstrated that this purified tubulin is devoid of alpha- and beta-tubulin polyglutamylated by western blot using established antibodies (see Fig 1E in Genova et al. EMBOJ 2023, using antibodies #AG-25B-0030 and #AG-25B-0039 characterized in Rogowski et al. Cell, 2010 and available from Adipogen). Crucially, we also showed that this kind of tubulin purification from knockout mice yields poly-glutamylated-free microtubules in a remarkably

reproducible way (see Figure EV1 in Genova et al. EMBOJ 2023). We apologize if this was unclear in the initial version of the manuscript. We thus added the following in the main text:

L417 [...] Remarkably, tubulin purified from TTL1/7-double KO mouse brain, lacking poly-glutamylated as previously established (Genova et al. EMBOJ 2023), displays a much weaker affinity towards [...]

And the methods (line 764): The characterization of these tubulin from knockout mice, in particular their content in tubulin polyglutamylated and acetylated was previously published (Genova et al. EMBOJ 2023).

As this characterization is published and the antibodies we used are well established, we do not think it would bring much to redo this characterisation with mass spectrometry, which is, at least in our hands, far less straightforward than western blot with established and well characterized antibodies when using brain purification rather than isotype-pure recombinant tubulin. We hope the reviewer agrees.

Reviewer #3 (Remarks to the Author):

Planelles-Herrero and colleagues have submitted a manuscript focused on the question of how Elongator promotes microtubule polymerization. Using an approach that rests on purification of subcomplexes and in vitro reconstitution assays, they show that Elongator can simultaneously bind to microtubules (via its Elp123 subcomplex) and to unpolymerized tubulin (via its Elp456 subcomplex), that the Elp123 complex can 'track' microtubule ends because it prefers GTP-like states of the microtubule, that the Elp456 complex binds unpolymerized tubulin primarily through tubulin's disordered C-termini, that promoting microtubule polymerization requires both binding activities, and that Elp456 binds best to poly-glutamylated tubulin, and hence through its polymerase activity can influence the composition of the microtubules it acts on. Overall the experiments are well-conducted, and the results are interesting. Some comments for the authors:

We thank this reviewer for this highly positive assessment of our work!

R3.1. *On line 132-133, the authors state that microtubule end-binding by Elongator can be observed in ~10% of the growth episodes. More discussion/context for this observation would be helpful. Is there something different/unique about those 10% of microtubules? Does 10% just reflect the combination of end-binding affinity and Elongator concentration? If the latter, it would help to get some information about how many Elongators are on the microtubule end (this relates to the observation in question but the information is also relevant for considering the mechanism of polymerase activity).*

We thank the reviewer for the comment.

First and foremost, while we do not directly image Elongator on all microtubules in our conditions, all our experiments suggest that Elongator affects most, if not all, the microtubules present in our assays. In other words, it's not that only the 10% of microtubules where we manage to image tip tracking do have higher growth speed for instance. If it were the case, we would expect all growth speed distributions in the paper to be binomial, which obviously they are not (see for instance Fig.4B, new Fig. 5C, new Sup. Fig. 5G). Along the same lines, the increase of microtubule poly-glutamylation we report for Elongator is a global effect, not an effect restricted to 10% of the population. This can again be seen on the distribution of the poly-glutamylation ratio in the microtubule population, which is not binomial). Lastly, when we looked, we could not really see a difference of microtubule growth speed between the microtubules where we could image Elongator compared to the ones where we cannot.

These key quantitative observations convinced us that the fact that we do not see tip tracking on all microtubules is rather due to combination of technical limitations which is convolved with the fact that there are probably few complexes at the tip because of steric hindrance (Elongator is one order of magnitude larger than EBs). Indeed, it is technically difficult to evaluate the degree of labelling of our SNAP-Elongator prep, so we do not know if ALL the Elongator molecules in our preparation are labelled or not. Most importantly, we intentionally use very low laser excitation conditions for all our imaging experiments so as to be as gentle as possible to minimize microtubule photodamages (microtubules are well-known to be photo sensitive). It is therefore possible that we are "missing" a lot of Elongator tip tracking events in our experiments, only managing to image the rarer events when multiple, well-labelled complexes are bound to the same tip at the same time.

R3.2. *The demonstration that microtubule- and tubulin-binding activities are jointly required to observe increased growth rates and decreased catastrophe frequency is interesting and well-demonstrated. The parallels with XMAP-215 are valid, but there is also a contrast that I think might be interesting to do more with. Whereas both XMAP-215 and Elongator increase microtubule growth rates (XMAP-215 more so than Elongator, at least at the concentrations tested for Elongator), XMAP-215 promotes catastrophe whereas Elongator suppresses catastrophe. There could be multiple reasons for this, but it seems reasonable to speculate that a difference could have to do with the mode of binding to unpolymerized tubulin: TOG domains prefer a curved conformation (and can in principle thereby influence the conformation of tubulin at the microtubule end) whereas Elongator presumably has little to no conformation-preference, because it binds mainly through the tails. This difference may underlie the differing effects on catastrophe.*

We thank the reviewer for this insightful comment. Indeed, there are several parallelisms between Elongator and XMAP-215 (and EBs). This was also raised by reviewer #2, and we kindly point the reviewer to our extended discussion in response to their point R2.4 above.

In a nutshell, there are several factors that explain why XMAP-215 increases microtubule growth to a greater extent compared to Elongator, including the fact that XMAP-215 primes the tubulin for their incorporation, as correctly pointed out by the reviewer. Importantly, the intrinsic properties of poly-glutamylated tubulin, which is selectively enriched by Elongator might also play a role. Indeed, thanks to a recent study published after our initial submission (Chen & Roll-Mecak, MBoC 2023), we now know that poly-glutamylated microtubules have intrinsically slower growth rate and are intrinsically less stable than non-modified microtubules. So the slower growth speed observed in the presence of Elongator could just be apparent and rather reflect the intrinsically slower growth speed of polyglutamylated tubulin.

However, there is another major difference, as pointed out by the reviewer: XMAP-215 promotes catastrophe, whilst Elongator suppresses it. This is indeed a non-trivial distinction, and the rationale behind the effect observed for XMAP-215 was that as it promotes microtubule growth, it might introduce more defects in the lattice, disrupting the growing end (Farmer *et al.* JCB 2021), and therefore increasing the catastrophe rate. In other words, XMAP-215 generates microtubules that are more dynamic.

Elongator, however, polymerises microtubules that grow (slightly) faster and yet are more stable. We speculated in the original draft that cells might use different polymerases to achieve different effects. For example, during cell division: whilst XMAP-215 highly dynamic microtubules are needed during metaphase for chromosome capture and pulling, Elongator's activity generating stable microtubules is relevant during late anaphase for the assembly of the central spindle (a structure that is indeed more stable than the metaphase spindle), as we characterised in our previous publication (Planelles-Herrero, V.J., et al. Nat Cell Biol 24, 1606–1616 (2022)). So we speculate that instead of having a “one microtubule polymerase good for everything”, cells have evolved different polymerases for different purposes.

Now, why are microtubules polymerised by Elongator more stable? Unfortunately, we do not have a definite answer for this. We agree with the reviewer that it might be a result of the mode of binding to the tubulin dimers. Another possibility, however, is the way that Elongator binds to the microtubule: whilst a TOG domain from XMAP-215 only binds to a single dimer in a protofilament, Elongator's big size (~270Å, compared to ~50Å for a TOG domain) likely means that it binds across or along protofilaments. This might have an additional effect, stabilizing the lateral or longitudinal interactions, respectively, in the protofilaments, therefore stabilizing the lattice.

We thank the reviewer for pointing out these key mechanistic details, and we have extended our discussion on this topic in the section “**Convergent evolution of the mode of action of microtubule polymerases**”.

R3.3. Overall the mechanism of polymerase activity still seems a little hazy, or at least deserving of a more discussion. Does it make sense that the presence of Elongator(s) at the microtubule end could increase the local concentration of tubulin ~1.4-fold? Does using higher concentrations of Elongator yield even larger increases in growth rate?

We thank the reviewer for these constructive comments, which we experimentally addressed. There are two questions here: i) does Elongator locally increase tubulin concentration and ii) does varying Elongator concentrations affects the microtubule growth rate. We will discuss both points independently for clarity.

i) Elongator-mediated local increase of tubulin concentration

First, we verified that, indeed, Elongator increases the local concentration of tubulin at microtubule ends. All our previous experiments suggested this was the case, but prompted by this reviewer, we directly verified this experimentally. For this, we performed microtubule polymerisation experiments from seeds placing us just below the tubulin critical concentration for microtubule elongation from seeds. At 6 μM tubulin, no growth can be observed in the control condition in our setup, whilst the addition of Elongator results in slow but consistent microtubule growth. Therefore, Elongator acts by decreasing the effective critical concentration of tubulin for microtubule elongation, which is conceptually similar to increasing the local tubulin concentration. This new results are presented in new Sup. Fig. S6C. Additionally, as discussed in the previous point (**R3.2**), poly-glutamylated microtubules, such as those polymerised by Elongator, are intrinsically less stable than regular microtubules. This suggests that Elongator is increasing the local concentration more than 1.4-fold, but that this increase is counteracted by the properties of the microtubules it elongates resulting in an effective 1.4-fold effect.

We have included those experiments in the revised manuscript as follows, in the “**Coordinated action between subcomplexes is critical for microtubule stabilization**” section:

“To test this hypothesis, we performed microtubule polymerization experiments close to the critical tubulin concentration required for microtubule elongation from seeds. In our experimental conditions little to no microtubule polymerization could be observed at a concentration of GTP- $\alpha\beta$ -tubulin heterodimers of 6 μM (Sup. Fig. S7C), in agreement with previously calculated critical concentrations for porcine brain tubulin⁴². Importantly, slow microtubule growth could be observed when 25 nM SNAP-Elongator was added (Sup. Fig. S7C), consistent with Elongator increasing the local concentration of tubulin.”

ii) Dose dependence of the effect of Elongator on microtubule dynamics

Motivated by this reviewer, we also characterised the effect that the concentration of Elongator, or more specifically, of Elp456, have in the growth rate and lifetime of microtubules. Indeed, we previously characterised that different concentrations of the full semi-endogenous Elongator complex had roughly the same effect on microtubule growth and lifetime (Planelles-Herrero, V.J., et al. Nat Cell Biol 24, 1606–1616 (2022)). In these experiments, however, the ratio between Elp123 and Elp456 was always the same, since we were using the full Elongator complex purified from cells.

However, in light of the data presented in this new study, we now know that the Elp123 and Elp456 subcomplexes have different activities: binding to microtubules and to tubulin dimers, respectively. We therefore hypothesised that increasing the concentration of Elp456 would increase the effect on microtubules for a given Elp123 concentration. We have fully characterised this effect in the revised manuscript (see new Fig. 5C, D) using a range of Elp456 concentrations (from 25 nM to 200 nM), on both the growth rate and the lifetime as suggested by this reviewer. In these conditions, the higher the concentration of Elp456, the higher the effect on microtubules, hinting at a mechanism in which Elp456 is the limiting factor. One plausible hypothesis is that Elp456 acts as a “shuttle” to bring tubulin dimers to Elp123, located at the growing end of microtubules. In this scenario, a higher availability of Elp456 would bring more tubulin to the microtubules, further increasing their speed of growth and lifetime.

We have included those experiments in the revised manuscript as follows (starting at line 343):

“We observed a dose-dependent effect when increasing amounts of *wild type* or *solo* Elp456 were added (Fig. 4B-D, Fig 5C-E). At concentrations below the K_d , the *solo* mutant, together with Elp123, does not display any effect on the microtubules. At higher concentrations, however, the weaker affinity is compensated and the *solo* mutant increases both the speed of growth (Fig. 5C) and the lifetime (Fig. 5D, E) of microtubules. Importantly, the effect of the *solo* mutant is always weaker than that observed for wild-type Elp456, although this difference is smaller at very high concentrations (i.e. 200 nM), that is, greatly above the calculated K_d . Similar results were observed for the minus ends (Sup. Fig. S3C, D).”

We hope the reviewer finds that the polymerase activity is better described in the revised draft, and we thank them for their suggestions.

R3.4. *The preference for poly-Glu tubulin is striking and demonstrated in a nice way in binding assays and reconstitutions using tubulins purified from different sources that yield correspondingly different modification states. How sensitive are the tubulin binding affinities to increases in ionic strength? Are the in vitro findings and claims about the effects on the tubulin code relevant to the function of Elongator at the spindle?*

We thank the reviewer for these excellent comments which we think helped us to considerably strengthen the paper.

The question regarding the ionic strength and the physiological relevance of Elongator in cellular contexts was also raised by reviewer #1 and #2. We kindly point the reviewer to our detailed answers to the points **2.8** and **1.2**, respectively.

Briefly, we have extensively characterised Elongator’s activity and found that increases in ionic strength do not affect microtubule binding, end-tracking or tubulin binding *in vitro*, nor do excess tRNA (new Sup. Fig. S5A, B, E). Moreover, we found that Elongator binds to microtubules and tubulin in *Drosophila* cytosolic extracts prepared in 150 mM KCl buffer (new Sup. Fig. S5C, D).

Furthermore, we now show that Elongator controls the levels of poly-glutamylation of microtubules *in vivo* at the spindle not only in cultured *Drosophila* cells and mouse fibroblasts, but also in tissues, namely in *Drosophila* Sensory Organ Precursors in actual flies (see new Fig. 6E-G). This establishes that the new function of Elongator we report to modulate the levels of poly-glutamylation of microtubule *in vitro* is relevant *in vivo* at the spindle. Interestingly, we could not measure an effect of

Elongator depletion on microtubule polyglutamylation in interphase, this was restricted to mitosis. It is tempting to speculate that this could be related to the difference in microtubule dynamics between interphase and mitosis.

What we do not know, however, is whether the role of Elongator in mediating central spindle asymmetry, which we unravelled in our previous study, relates to its role in modulating spindle microtubule poly-glutamylation. In other words, does the role of Elongator in promoting central spindle asymmetry during asymmetric cell division stems from its role in modulating spindle polyglutamylation? As our *in vitro* results suggests that Elongator preferentially binds polyglutamylated tubulin, this is likely the case, but we have not formally demonstrated this point for technical reasons. Indeed, this would require either an Elongator that can elongate non-polyglutamylated tubulin, or a way to perturb tubulin poly-glutamylation levels in a way that it *only* affects Elongator-mediated tubulin dynamics and not all the other functions of the PTM in the cell.

We hope the reviewer agrees that taken altogether, these new results eliminate any doubt regarding the physiological relevance of our *in vitro* findings.

R3.5. *(minor) the analysis of the cryo-EM class averages that apparently did not show evidence of tubulin binding seemed somewhat superficial. This is not an essential point for the manuscript, but in general I think one could dig a little deeper to substantiate the conclusion (or find more diffuse evidence to support a flexible binding mode for tubulin).*

We thank the reviewer for this comment, which was raised by reviewer #1. We kindly direct the reviewer to our answer to their point **R1.3**, which we won't duplicate here for clarity. In a nutshell, we agree that the cryo-EM analysis deserved more detail, and we have improved and expanded the processing of our datasets. Unfortunately, this does not change our conclusions as we are still unable to unambiguously locate the tubulin in the complex with Elp456. But we think that our results that Elp456 binds to the C-terminal tail of tubulin, which we now confirmed with cross-linking mass spectrometry, make perfect sense with these cryo-EM results.

Dear Emmanuel and Vicente,

Thank you for submitting your revised manuscript to The EMBO Journal. Your manuscript has now been seen by two of the original reviewers. While reviewer #2 now supports publication of the manuscript in its current form, reviewer #1 requests to streamline the study by removing the structural information and adding a few control experiments. I will therefore be happy to accept the manuscript for publication in The EMBO Journal after its minor revision as requested by reviewer #1 and reformatting along the guidelines included in the attached document.

Please feel free to contact me if you have any further questions regarding this final revision. Please use the link below to upload the revised files.

Thank you for the opportunity to consider your work for publication, and I look forward to receiving your revised manuscript.

With best regards,

Ieva

We realize that it is difficult to revise to a specific deadline. In the interest of protecting the conceptual advance provided by the work, we recommend a revision within 3 months (9th Jan 2025). Please discuss the revision progress ahead of this time with the editor if you require more time to complete the revisions.

Referee #1:

In the presented study Planelles-Herrero and colleagues expand their recent work on the link between Elongator and microtubule dynamics. I have previously reviewed the manuscript for another journal and I was excited about the results. Nonetheless, at that time I found the findings still too early to draw too many final conclusions. Since, the authors have performed additional experiments and revised the manuscript. In my opinion, the additional experiments and analyses strengthen the conclusions of the paper and provide some important new insights. However, the manuscript is still full of overstatements - like in the abstract "Here, we unravelled the molecular mechanism by which Elongator controls microtubule dynamics." The authors must have a completely different understanding of what molecular insights are. This is repeated on page 4 "Here, we reveal the molecular mechanism by which Elongator stabilizes microtubules." I find the data and the findings really interesting, but I have not found a single molecular explanation on how Elongator stabilizes MTs.

Despite my criticism about these overstatements, I am very positive about the manuscript being published in EMBO Journal after some major reorganization of the manuscript. In short, the authors add strong new data, like the analysis of how "Elongator can discriminate between microtubules and tRNA", the analyses of Elongator proteins from different species and the newly presented in vivo data. For instance, the newly introduced data on the comparison between MT- and tRNA binding is indeed really interesting. The assays in presence of 100uM tRNAs or within the cell lysate do provide additional insights into the interplay between the two functions of Elongator. If I understand correctly, the data now clearly shows that Elongator (more precisely Elp456) is able to bind tRNAs and $\alpha\beta$ -tubulin heterodimers at the same time, but when Elongator binds to MTs it is not able to bind tRNAs anymore. However, the incorporation of high-quality data also make the fact obvious to me, that certain other pieces of data presented in this manuscript do not pass/fulfil the same quality criteria.

Hence, I would suggest to remove the complete cryo-EM analyses from the study, as it does not add anything, except for vague speculations - as mentioned by the authors themselves "Note that during this 2D analysis step we cannot exclude the extra-signal belonging to tightly-packed neighbouring Elp456 molecules." - in my opinion this is a very likely scenario. The mass spec data with AF3 models provides similar level of insights. Furthermore, I would suggest excluding all data on the "solo" mutant that only adds confusion. The mutants is not ideal and even if the authors are reasoning that it is better than a mutant that full abolishes binding, the interpretation of these data leaves too many possibilities. A manuscript that focuses on less, but more solid, data would definitely be a great follow-up on the publications that described the initial observation in flies. In my opinion, a focus on the following topics would be enough to justify publication in EMBO Journal

- Elongator complex tracks the growing ends of microtubules.
- Elp123 binds but does not stabilize microtubules.
- Elongator complex binds to tubulin through Elp456 (with the necessary controls described below).
- Binding to both microtubules and free tubulin is essential for microtubule stabilization.
- Elongator is a tubulin polymerase selective for poly-glutamylated monomers.

Instead of showing more and more data, and writing more and more text to explain weak data, I would suggest to reevaluate the quality of specific experimental approaches. I would highly recommend the publication of simplified manuscript in EMBO J.

In addition, I have a few technical issues and questions that should be addressed and resolved before publication of the work.

- Figure 3B - the specificity of the assay should be tested with other proteins as negative controls. In addition, I did not find a co-migration assay for Elp123 - only an IP assay. The interaction with $\alpha\beta$ 1-tubulin heterodimers needs to be compared with the same assay format. Otherwise, the behavior/specificity of the two subcomplexes cannot be concluded.
- Figure S7C - the provided data lack the respective controls and the basically non-existing labelling of the figure makes hard to comprehend.
- The authors themselves must know that the following statement from the response is not very satisfying and typically the amounts needed for MS are far lower than for any other method - "We previously attempted to analyse the PTM content of both fractions by mass-spectrometry, but we found that the amount of Elongator is too low to provide consistent data." Maybe this data can still be included or at least the authors can comment in the manuscript that no differences in PTMs were detectable.
- Please show the knock-down efficiency of the used RNAi experiments - making sure that the effects are indeed related to the depletion of the proteins and not to other effects (e.g. changes in the cell cycle distribution).
- Label for Figure 5E is missing

Referee #2:

The authors have addressed many of the concerns of this reviewer about their in vitro studies with the additional data. I believe the revised manuscript is now ready for publication.

Referee #1:

In the presented study Planelles-Herrero and colleagues expand their recent work on the link between Elongator and microtubule dynamics. I have previously reviewed the manuscript for another journal and I was excited about the results. Nonetheless, at that time I found the findings still too early to draw too many final conclusions. Since, the authors have performed additional experiments and revised the manuscript. In my opinion, the additional experiments and analyses strengthen the conclusions of the paper and provide some important new insights. However, the manuscript is still full of overstatements - like in the abstract "Here, we unravelled the molecular mechanism by which Elongator controls microtubule dynamics." The authors must have a completely different understanding of what molecular insights are. This is repeated on page 4 "Here, we reveal the molecular mechanism by which Elongator stabilizes microtubules." I find the data and the findings really interesting, but I have not found a single molecular explanation on how Elongator stabilizes MTs.

We thank this reviewer for these encouraging remarks and for acknowledging the strength of the additional experiments and analyses in our revised manuscript.

In light of this feedback, we have revised the language throughout our abstract and text, and we have removed the mention to “molecular mechanisms”. We would like to respectfully highlight, however, that in our view the data presented does indeed provide molecular insights into the mechanism by which Elongator modulates microtubule dynamics.

It's all a question of acceptance of the word “molecular”. In our view, a molecular mechanism can be understood as defining how protein:protein molecular interactions contribute to a biological outcome, even without structural details. In this study, we demonstrate that Elp456 binds to the tubulin tails (Figure 3, Figure 6), Elp456 binds to Elp123 (Figure 5), Elp123 associates with microtubule ends (Figures 1 and 2), all of which are necessary (Figure 4) for Elongator's role in modulating microtubule dynamics. Our additional data (Figure S7C, D) show how these interactions influence the critical concentration of tubulin, supporting elongation.

Thus, overall, we think that our results do provide insights into the molecular interactions by which Elongator influences microtubule dynamics, albeit without solving atomic-level structures.

Despite my criticism about these overstatements, I am very positive about the manuscript being published in EMBO Journal after some major reorganization of the manuscript. In short, the authors add strong new data, like the analysis of how "Elongator can discriminate between microtubules and tRNA", the analyses of Elongator proteins from different species and the newly presented in vivo data. For instance, the newly introduced data on the comparison between MT- and tRNA binding is indeed really interesting. The assays in presence of 100uM tRNAs or within the cell lysate do provide additional insights into the interplay between the two functions of Elongator. If I understand correctly, the data now clearly shows that Elongator (more precisely Elp456) is able to bind tRNAs and $\alpha\beta$ -tubulin heterodimers at the same time, but when Elongator binds to MTs it is not able to bind tRNAs anymore. However, the incorporation of high-quality data also make the fact obvious to me, that certain other pieces of data presented in this manuscript do not pass/fulfil the same quality criteria.

Hence, I would suggest to remove the complete cryo-EM analyses from the study, as it does not add anything, except for vague speculations - as mentioned by the authors themselves "Note that during this 2D analysis step we cannot exclude the extra-signal belonging to tightly-packed neighbouring Elp456 molecules." - in my opinion this is a very likely scenario. The mass spec data with AF3 models provides similar level of insights. Furthermore, I would suggest excluding all data on the "solo" mutant that only adds confusion. The mutants is not ideal and even if the authors are reasoning that it is better than a mutant that full abolishes binding, the interpretation of these data leaves too many possibilities. A manuscript that focuses on less, but more solid, data would definitely be a great follow-up on the publications that described the initial observation in flies. In my opinion, a focus on the following topics would be enough to justify publication in EMBO Journal

- *Elongator complex tracks the growing ends of microtubules.*
- *Elp123 binds but does not stabilize microtubules.*
- *Elongator complex binds to tubulin through Elp456 (with the necessary controls described below).*
- *Binding to both microtubules and free tubulin is essential for microtubule stabilization.*
- *Elongator is a tubulin polymerase selective for poly-glutamylated monomers.*

Instead of showing more and more data, and writing more and more text to explain weak data, I would suggest to reevaluate the quality of specific experimental approaches.

I would highly recommend the publication of simplified manuscript in EMBO J.

We thank the reviewer for their detailed feedback and agree that the manuscript would benefit from simplification. As suggested, we will remove the cryo-EM analysis and the ΔpolyN mutant to streamline the results as they are mostly negative results.

However, we think that retaining the “solo” mutant data is essential to understanding the (molecular) mechanism by which Elongator stabilizes microtubules. This mutant critically allows us to demonstrate that the interaction between the two subcomplexes, namely Elp123 and Elp456 is key and necessary for this stabilization.

While we acknowledge the reviewer’s concern that the “solo” mutant does not entirely abolish the interaction, we wish to emphasize that there exists a concentration range (25-50 nM) where the “solo” mutant shows no effect, whereas the wild-type Elp456 robustly stabilizes microtubules (previous Figure 5C-E). This concentration range is relevant, as it aligns closely with the estimated *in vivo* levels of the least abundant subcomplex protein, Elp6, measured at approximately 50 nM (OpenCell, Cho et al., 2022, Science).

In line with the reviewer’s suggestion, we have extensively reorganized the figures and rewritten the manuscript to enhance clarity and streamline the flow in the presentation of the results. Additionally, we have decided to move the “solo” data from the main figures into the appendix figures (Appendix Figure S4) to improve readability and support a clearer narrative.

In addition, I have a few technical issues and questions that should be addressed and resolved before publication of the work.

- *Figure 3B - the specificity of the assay should be tested with other proteins as negative controls. In addition, I did not find a co-migration assay for Elp123 - only an IP assay. The*

interaction with $\alpha 1\beta 1$ -tubulin heterodimers needs to be compared with the same assay format. Otherwise, the behavior/specificity of the two subcomplexes cannot be concluded.

We agree with the reviewer and have added more controls to the co-migration data.

Specifically, we have added a co-migration assay of Elp456 with BSA as a negative control, showing that indeed Elp456 and BSA do not co-migrate (Appendix Figure S3F). Moreover, we have also performed a co-migration assay of Elp123 and tubulin, confirming that Elp123 does not interact with $\alpha 1\beta 1$ -tubulin heterodimers (Appendix Figure S3G). As Elp123 and tubulin do not interact, we deemed an extra control between Elp123 and BSA irrelevant.

We hope that these new controls will clear any remaining doubts from this reviewer about the interactions between Elp456, Elp123 and tubulin heterodimers.

We have added several references to these panels throughout the text.

- *Figure S7C - the provided data lack the respective controls and the basically non-existing labelling of the figure makes hard to comprehend.*

We have added the controls, improved the labelling, and included a new quantifications to help with readability and to highlight the robustness of the observations (Figure EV3A, B).

- *The authors themselves must know that the following statement from the response is not very satisfying and typically the amounts needed for MS are far lower than for any other method - "We previously attempted to analyse the PTM content of both fractions by mass-spectrometry, but we found that the amount of Elongator is too low to provide consistent data." Maybe this data can still be included or at least the authors can comment in the manuscript that no differences in PTMs were detectable.*

We appreciate the reviewer's insights and apologize for any confusion our wording may have caused.

While we agree that protein amounts required for MS are typically on the lower end of the spectrum for general protein identification, our aim here would not be to merely identify proteins but to detect specific PTMs in Elongator. More specifically, we sought to determine any differences in PTM content between the microtubule-bound and unbound fractions of Elongator. Achieving this **quantitatively** and with **high confidence** is challenging in heterogenous samples, as peptide generation and identification vary significantly across experiments. In other words, it is one thing to detect a given PTM in an experiment, but is it a physiologically relevant one? Validating these findings would require further *in vivo* investigation with point mutants lacking specific modifications, which we think is way beyond the scope of this manuscript.

We are cautious with our statements, and for this reason we refrain from speculating on the role of Elongator's PTMs in microtubule binding in the manuscript. We hope the reviewer agrees with this prudent approach.

- *Please show the knock-down efficiency of the used RNAi experiments - making sure that the effects are indeed related to the depletion of the proteins and not to other effects (e.g. changes in the cell cycle distribution).*

The efficiency of the RNAi was shown in the revised manuscript in Supplementary Figure S9H-I (now Fig EV5C, D). We have added a specific reference to it in the text ("**see Fig EV5C, D for RNAi deficiency**").

- *Label for Figure 5E is missing*

We thank the reviewer for spotting this error, which has now been corrected.

Dear Emmanuel and Vicente,

Thank you for submitting your manuscript to The EMBO Journal. I have now looked through your point-by-point response and I find it very reasonable. Therefore, there now remain only a few editorial points and formatting steps that need to be completed before I can extend official acceptance of the manuscript:

1. Please check if the email provided for the author Mariya Genova (mariya.genova@curie.fr) is correct.
2. Please check that the funding information is correct and identical both in the manuscript and our online system. Currently, France Alzheimer grant 2023 and Institut Curie 3-i PhD Program (IC-3i) are missing from our online system.
3. In the Author Checklist file, please fill in the row 57 (Laboratory Animals or Model Organisms).
4. Please add a "Disclosure and competing interests statement" (further info: <https://www.embopress.org/page/journal/14602075/authorguide#conflictsofinterest>).
5. CRedit has replaced the traditional author contributions section because it offers a systematic, machine-readable author contributions format that allows for more effective research assessment. Please remove the Authors Contributions from the manuscript and use the free text boxes beneath each contributing author's name in our online submission system to add specific details on the author's contribution. More information is available in our guide to authors.
6. Please update references according to The EMBO Journal style - where there are more than 10 authors on a paper, the first 10 should be listed, followed by 'et al.' Please see further information here: <https://www.embopress.org/page/journal/14602075/authorguide#referencesformat>
7. In the Appendix, please add page numbers and a brief table of contents.
8. In the Data Availability section, please provide resolvable links to the datasets PXD058480, S-BIAD1471, S-BIAD1472 and S-BIAD1473.
9. In our standard source data check, we have noted unexplained numerical duplications in the source data for figure 5E. I have attached the corresponding file with the detected duplications labelled in colour. Please take a look and correct if needed. A brief explanation would be very helpful.
10. Several of the source data image files in our system appear devoid of signal, e.g., for figures 1D-E, 2A, 3A, 4A, 4D, 5E-F, EV1B-C, EV2A (apart from the third row), EV2C, EV5E, EV5G. Please check if there have been any problems with the export.
11. Our data editors have flagged the following issues in figure legends that need correcting:
 - Please provide the exact p values in the legends of figures 4B, 5D, E-G; EV2 B; EV3 B, EV4 E, EV5 F.
 - Please indicate the statistical test used for data analysis in the legends of figures 5E-G, EV5 F, H.
 - Please provide information on the number and nature of replicates in the legends of figures 4C, EV4 F, EV5 F, H.
 - Please define the error bars in the legends of figures 5D, E-G; EV2 B; EV3 B, EV4 F, EV5 F, H.
 - Please define the scale bar for figures EV1 B, C; EV3 A, D, E; EV4 C.
 - Please note that scale bar and its definition are missing for figure EV3 F.
12. Papers published in The EMBO Journal are accompanied online by a 'Synopsis' to enhance discoverability of the manuscript. It consists of A) a short (1-2 sentences) summary of the findings and their significance, B) 3-4 bullet points highlighting key results and C) a synopsis image that is 550x300-600 pixels large (width x height, jpeg or png format). You can either show a model or key data in the synopsis image. Please note that the image size is rather small and that text needs to be readable at the final size. Please send us this information together with the revised manuscript.

With best wishes,

leva

leva Gailite, PhD
Senior Scientific Editor
The EMBO Journal
Meyerhofstrasse 1
D-69117 Heidelberg
Tel: +4962218891309
i.gailite@embojournal.org

We realize that it is difficult to revise to a specific deadline. In the interest of protecting the conceptual advance provided by the work, we recommend a revision within 3 months (10th Mar 2025). Please discuss the revision progress ahead of this time with the editor if you require more time to complete the revisions.

The authors addressed the remaining editorial issues.

Dear Vicente and Emmanuel,

Thank you for addressing the final editorial points. I am now pleased to inform you that your manuscript has been accepted for publication in the EMBO Journal. Congratulations on a great study!

Before we forward your manuscript to our publishers, we would like to propose some minor edits in the manuscript abstract and synopsis (please see below and the attached file). I have also written a short blurb that will accompany the title of your manuscript in our online system. Please let me know if any corrections or adjustments are needed:

Blurb:

Two subcomplexes of the tRNA modifier, Elongator, bind to microtubule ends and tubulin dimers and selectively promote growth of microtubules enriched for polyglutamylated tubulin.

Synopsis:

Elongator, an established tRNA-modifying complex, can also stabilize microtubules during asymmetric cell division. This study shows that Elongator subcomplexes achieve this by binding to microtubule ends as well as tubulin dimers, and enriching microtubules with polyglutamylated tubulin.

- Dual binding mechanism: Elongator binds microtubule ends and free $\alpha\beta$ -tubulin heterodimers via its Elp123 and Elp456 subcomplexes, respectively.
- Microtubule stabilization: Coupled activity of Elp123 and Elp456 reduces the tubulin concentration required for microtubule elongation, increasing microtubule growth speed and reducing their catastrophe rate.
- Selective tubulin recognition: Elp456 binds stronger to polyglutamylated tubulin, resulting in preferential incorporation of polyglutamylated tubulin into Elongator-assembled microtubules.
- Tubulin code rewriting: By enriching microtubules with polyglutamylated tubulin, Elongator modifies the tubulin code in vitro, in cells and in vivo in Drosophila sensory organ precursor cells.

If you have any questions, please do not hesitate to contact the Editorial Office. Thank you for your contribution to The EMBO Journal and congratulations on a nice study!

Best wishes,

Ieva
